# Scaling Autoregressive Models for Content-Rich Text-to-Image Generation

**Jiahui Yu**[*]   **Yuanzhong Xu**[†]   **Jing Yu Koh**[†]   **Thang Luong**[†]   **Gunjan Baid**[†]
**Zirui Wang**[†]      **Vijay Vasudevan**[†]      **Alexander Ku**[†]
**Yinfei Yang   Burcu Karagol Ayan   Ben Hutchinson**
**Wei Han   Zarana Parekh   Xin Li   Han Zhang**
**Jason Baldridge**[†]      **Yonghui Wu**[*]
{jiahuiyu, yuanzx, jykoh, thangluong, gunjanbaid, ziruiw, vrv, alexku,
jasonbaldridge, yonghui}@google.com[§]

[*] **Equal contribution.**      [†] **Core contribution.**

**Google Research**

**Reviewed on OpenReview:** `https://openreview.net/forum?id=AFDcYJKhND`

## Abstract

We present the Pathways (Dean, 2021) Autoregressive Text-to-Image (Parti) model, which generates high-fidelity photorealistic images and supports content-rich synthesis involving complex compositions and world knowledge. Parti treats text-to-image generation as a sequence-to-sequence modeling problem, akin to machine translation, with sequences of image tokens as the target outputs rather than text tokens in another language. This strategy can naturally tap into the rich body of prior work on large language models, which have seen continued advances in capabilities and performance through scaling data and model sizes. Our approach is simple: First, Parti uses a Transformer-based image tokenizer, ViT-VQGAN, to encode images as sequences of discrete tokens. Second, we achieve consistent quality improvements by scaling the encoder-decoder Transformer model up to 20B parameters, with a new state-of-the-art zero-shot FID score of 7.23 and finetuned FID score of 3.22 on MS-COCO. Our detailed analysis on Localized Narratives as well as PartiPrompts (P2), a new holistic benchmark of over 1600 English prompts, demonstrate the effectiveness of Parti across a wide variety of categories and difficulty aspects. We also explore and highlight limitations of our models in order to define and exemplify key areas of focus for further improvements.

## 1 Introduction

People are generally able to conjure rich and detailed scenes through descriptions expressed in written or spoken language. Supporting the ability to generate images based on such descriptions can potentially unlock creative applications in many areas of life, including the arts, design, and multimedia content creation. Recent research on text-to-image generation, *e.g.*, DALL-E (Ramesh et al., 2021) and CogView (Ding et al., 2021), has made significant progress in generating high-fidelity images and demonstrating generalization capabilities to unseen combinations of objects and concepts. Both treat the task as a form of language modeling, from textual descriptions into visual words, and use modern sequence-to-sequence architectures like Transformers (Vaswani et al., 2017) to learn the relationship between language inputs and visual outputs. A key component of these approaches is the conversion of each image into a sequence of discrete units through the use of an image tokenizer such as dVAE (Rolfe, 2017) or VQ-VAE (Van Den Oord et al., 2017). Visual tokenization essentially unifies the view of text and images so that both can be treated simply as sequences of discrete tokens—and thus amenable to sequence-to-sequence models. To that end, DALL-E and CogView

---

[§]Correspondence to {jiahuiyu, jasonbaldridge, yonghui}@google.com.

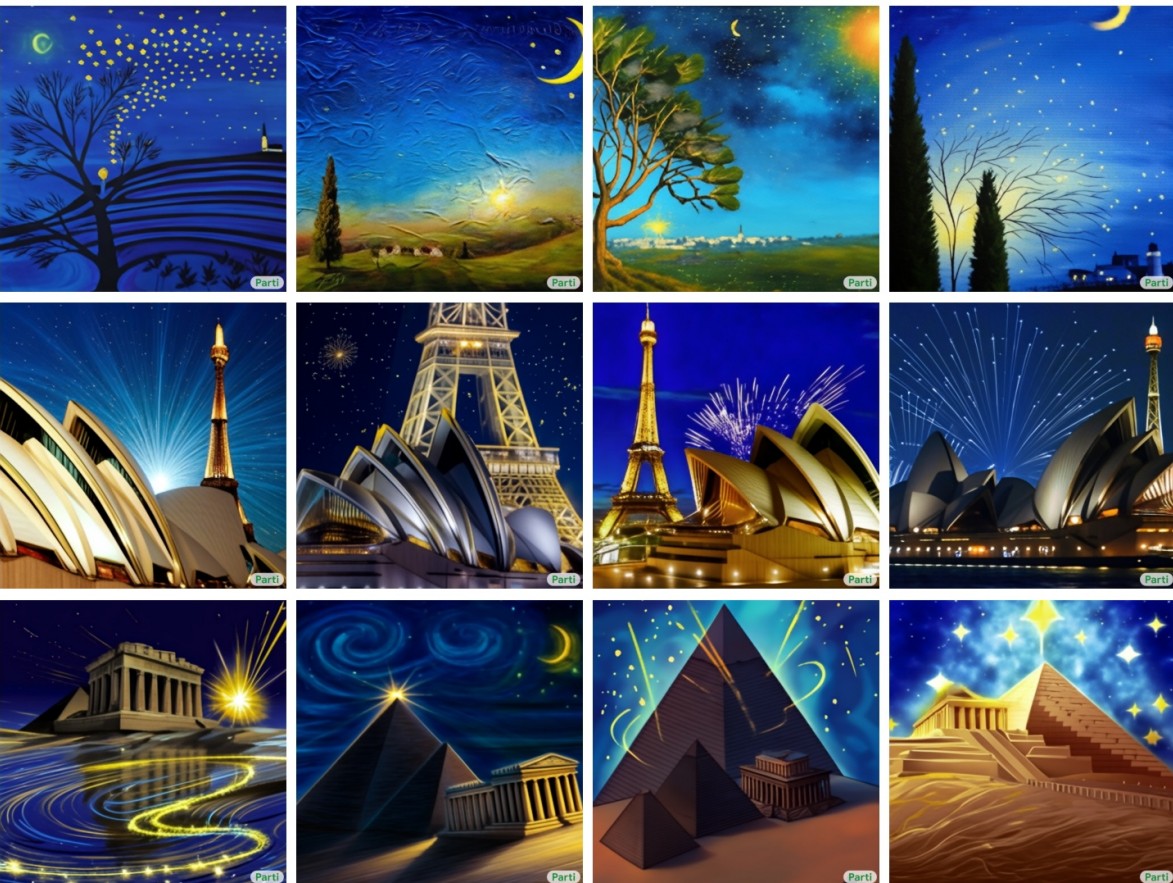

Figure 1: Example images generated by Parti. **Top row**: "*Oil-on-canvas painting of a blue night sky with roiling energy. A fuzzy and bright yellow crescent moon shining at the top. Below the exploding yellow stars and radiating swirls of blue, a distant village sits quietly on the right. Connecting earth and sky is a flame-like cypress tree with curling and swaying branches on the left. A church spire rises as a beacon over rolling blue hills.*" (a 67-word description of the Starry Night by Vincent van Gogh). **Middle row**: "*A close-up high-contrast photo of Sydney Opera House sitting next to Eiffel tower, under a blue night sky of roiling energy, exploding yellow stars, and radiating swirls of blue*". **Last row**: Similar to the middle row, but with "*anime illustration*" and different landmarks (*the Great Pyramid and the Parthenon*).

employed decoder-only language models, similar to GPT (Radford et al., 2018), to learn from a large collection of potentially noisy text-image pairs (Changpinyo et al., 2021; Jia et al., 2021a). Make-A-Scene (Gafni et al., 2022) further expands on this two-stage modeling approach to support both text and scene-guided image generation.

A different line of research with considerable momentum involves diffusion-based text-to-image models, such as GLIDE (Nichol et al., 2022) and concurrent works DALL-E 2 (Ramesh et al., 2022) (*a.k.a.*, unCLIP) and Imagen (Saharia et al., 2022). These models eschew the use of discrete image tokens in favor of diffusion models (Ho et al., 2020; Dhariwal & Nichol, 2021) to directly generate images. These models improve zero-shot Fréchet Inception Distance (FID) scores on MS-COCO (Lin et al., 2014) and produce images of markedly higher-quality and greater aesthetic appeal compared to previous work. Even so, autoregressive models for text-to-image generation remain appealing given extensive prior work on scaling large language models (Brown et al., 2020; Cohen et al., 2022; Du et al., 2022; Chowdhery et al., 2022a) and advances in discretizing other modalities–such as images and audio–so that inputs in those modalities can be treated as language-like tokens. This work presents the Pathways Autoregressive Text-to-Image (*Parti*) model, which generates high-quality images from text descriptions, including photo-realistic ones, paintings, drawings,

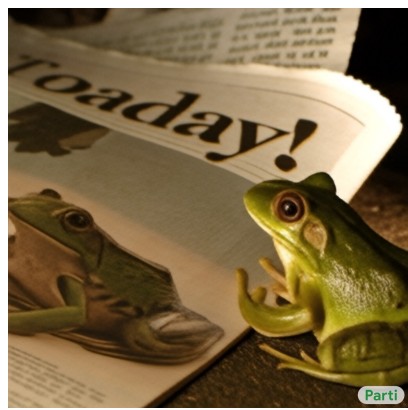

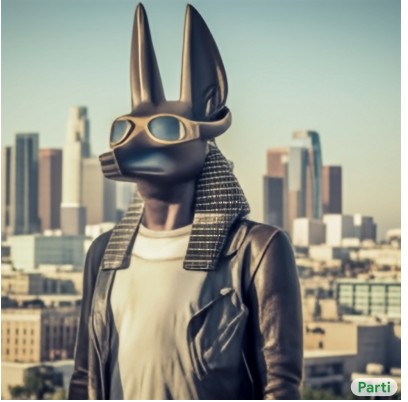

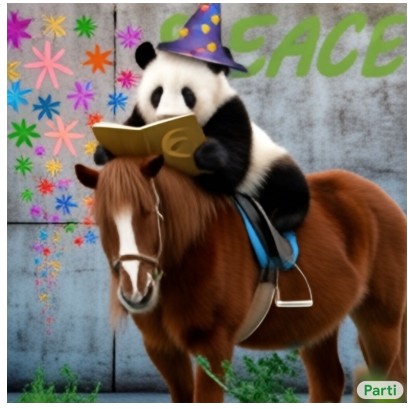

**A**. *A photo of a frog reading the newspaper named "Toaday" written on it. There is a frog printed on the newspaper too.*

**B**. *A portrait of a statue of the Egyptian god Anubis wearing aviator goggles, white t-shirt and leather jacket. The city of Los Angeles is in the background. Hi-res DSLR photograph.*

**C**. *A high-contrast photo of a panda riding a horse. The panda is wearing a wizard hat and is reading a book. The horse is standing on a street against a gray concrete wall. Colorful flowers and the word "PEACE" are painted on the wall. Green grass grows from cracks in the street. DSLR photograph. daytime lighting.*

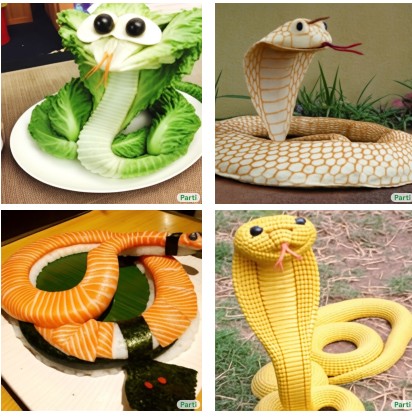

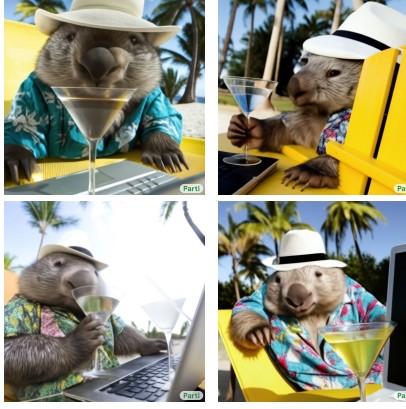

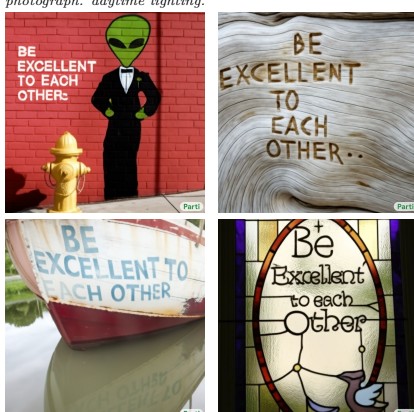

**D**. *A giant cobra snake made from X. X ∈ { "salad", "pancakes", "sushi", "corn"}*

**E**. *A wombat sits in a yellow beach chair, while sipping a martini that is on his laptop keyboard. The wombat is wearing a white panama hat and a floral Hawaiian shirt. Out-of-focus palm trees in the background. DSLR photograph. Wide-angle view.*

**F**. *The saying "BE EXCELLENT TO EACH OTHER" ..., (a) brick wall and alien (b) driftwood. (c) old wooden boat with reflection. (d) stained glass. (See text for full prompts.)*

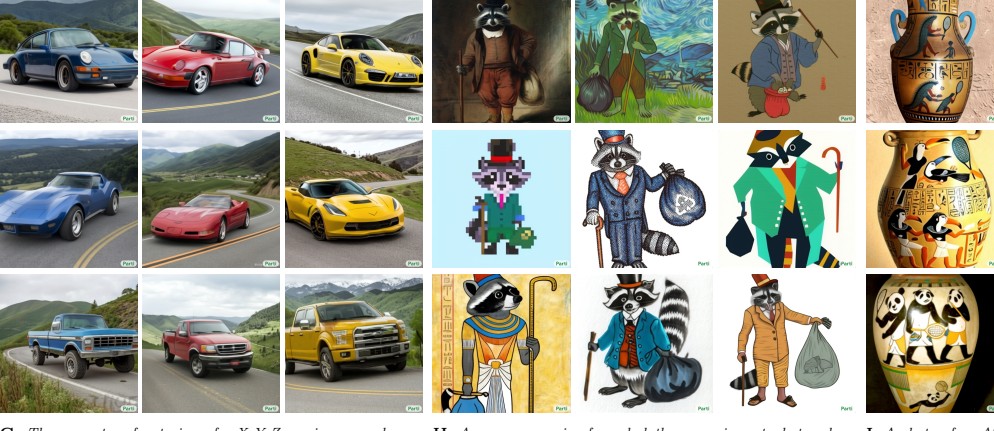

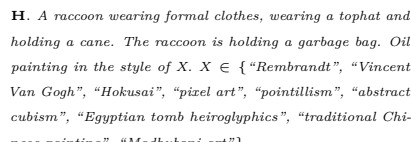

**G**. *Three-quarters front view of a X Y Z coming around a curve in a mountain road and looking over a green valley on a cloudy day. DSLR photograph. X ∈ { blue, red, yellow}, Y ∈ {1977, 1997, 2017}, Z ∈ {Porsche 911, Corvette, Ford F-150}*

**H**. *A raccoon wearing formal clothes, wearing a tophat and holding a cane. The raccoon is holding a garbage bag. Oil painting in the style of X. X ∈ { "Rembrandt", "Vincent Van Gogh", "Hokusai", "pixel art", "pointillism", "abstract cubism", "Egyptian tomb heiroglyphics", "traditional Chinese painting", "Madhubani art"}*

**I**. *A photo of an Athenian vase with a painting of X playing Y in the style of Egyptian hieroglyphics. X ∈ { "pandas", "toucans", "pangolins"}, Y ∈ { "tennis", "soccer", "basketball"}*

Figure 2: Selected Parti images. See Section 6.1 for discussion.

and more (see Fig. 1 & 2). We show that with a ViT-VQGAN (Yu et al., 2022a) image tokenizer, scaling autoregressive models is an effective way to improve text-to-image generation, enabling such models to accurately integrate and visually convey world knowledge.

Parti is a sequence-to-sequence model based on the Transformer (Vaswani et al., 2017), an architecture critical to performance on many tasks, including machine translation (Vaswani et al., 2017), speech recognition (Zhang et al., 2020; Gulati et al., 2020), conversational modeling (Adiwardana et al., 2020), image captioning (Yu et al., 2022b), and many others. Parti takes text tokens as inputs to an encoder and autoregressively predicts discrete image tokens with a decoder (see Figure 3). The image tokens are produced by the Transformer-based ViT-VQGAN image tokenizer (Yu et al., 2022a), which produces higher-fidelity reconstructed outputs and has better codebook utilization compared with dVAE (Rolfe, 2017), VQ-VAE (Van Den Oord et al., 2017), and VQGAN (Esser et al., 2021). Parti is conceptually simple: all of its components – encoder, decoder and image tokenizer – are based on standard Transformers (Vaswani et al., 2017). This simplicity makes it straightforward to scale our models using standard techniques and existing infrastructure (Shoeybi et al., 2019; Du et al., 2022; Chowdhery et al., 2022a; Xu et al., 2021). To explore the limits of this two-stage text-to-image framework, we scale the parameter size of Parti models up to 20B, and observe consistent quality improvements in terms of both text-image alignment and image quality. The 20B Parti model achieves new state-of-the-art zero-shot FID score of 7.23 and finetuned FID score of 3.22 on MS-COCO.

While most recent work has focused exclusively on the MS-COCO benchmark, we also show that strong zero-shot and finetuned results can be achieved on the Localized Narratives dataset (Pont-Tuset et al., 2020), which has descriptions that are four times longer than MS-COCO's on average. These results demonstrate the strong generalization capability of Parti to longer descriptions, allowing us to pile on considerable complexity in our explorations with the model (see examples in Figure 2 and the Appendix, and the discussion of *growing a cherry tree* in Section 6.2). Nevertheless, existing captioning / descriptioning datasets are limited to photographs and descriptions of their contents, but much of the appeal of text-to-image models is that they can produce novel outputs for fantastical prompts. Given this, we introduce *PartiPrompts* (P2), a rich set of over 1600 (English) prompts curated to measure model capabilities across a variety of categories and controlled dimensions of difficulty.[1] Each prompt in P2 is associated with both a broad category (out of 12 categories, ranging from abstract to animals, vehicles, and world knowledge) and a challenge dimension (out of 11 aspects, ranging from basic, to quantity, words & symbols, linguistics, and complex). Our detailed analyses and human evaluations on MS-COCO, Localized Narratives and P2, along with extensive discussion of Parti's limitations (Section 6.3) give a comprehensive picture of the strengths and weaknesses of Parti models—and establish autoregressive models as strong contenders for high-quality, broadly capable, open-domain text-to-image generation models.

Our main contributions include: (1) We demonstrate that autoregressive models can achieve *state-of-the-art* performance: 7.23 zero-shot and 3.22 finetuned FID on MS-COCO, and 15.97 zero-shot and 8.39 finetuned FID on Localized Narratives; (2) *Scale matters*: our largest Parti model (20B) is most capable at high-fidelity photo-realistic image generation and supports content-rich synthesis, particularly those involving complex compositions and world knowledge; and (3) We also introduce a holistic benchmark, the *PartiPrompts* (P2), propose a novel concept of *Growing a Cherry Tree*, establish a new precedent regarding identifying limitations of text-to-image generation models, and provide a detailed breakdown, with exemplars, for error types we observed.

## 2 Parti Model

Similar to DALL-E (Ramesh et al., 2021), CogView (Ding et al., 2021), and Make-A-Scene (Gafni et al., 2022), Parti is a two-stage model, composed of an image tokenizer and an autoregressive model, as highlighted in Figure 3. The first stage involves training a tokenizer that turns an image into a sequence of discrete visual tokens for training and reconstructs an image at inference time. The second stage trains an autoregressive sequence-to-sequence model that generates image tokens from text tokens. We describe details for these two stages below, together with other techniques for building high-performing autoregressive text-to-image models, such as text encoder pretraining, classifier-free guidance, and reranking.

---

[1]Imagen (Saharia et al., 2022) introduces 200 prompts in DrawBench for similar purposes. See Section 4.3 for discussion.

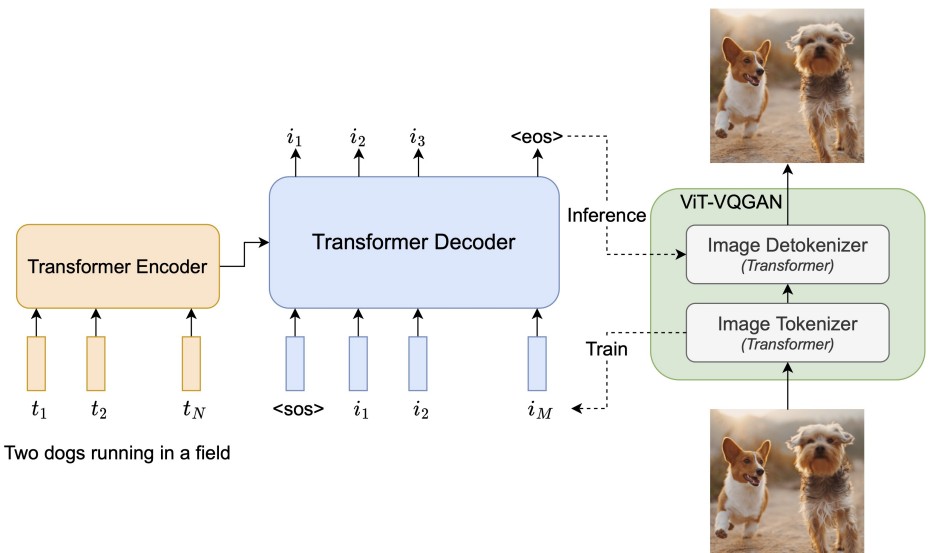

Figure 3: Overview of Parti sequence-to-sequence autoregressive model (left) for text-to-image generation with ViT-VQGAN as the image tokenizer (Yu et al., 2022a) (right).

## 2.1 Image Tokenizer

Autoregressive text-to-image models must linearize 2D images into 1D sequences of patch representations. In the limit, these are just pixels, as with iGPT (Chen et al., 2020), but this requires modeling very long sequences even for small images (*e.g.*, a $256 \times 256 \times 3$ RGB image leads to 196,608 rasterized values). Worse, it is based on a very low-level representation of the inputs rather than a richer one informed by the position of a pixel in the context of the image. Previous work (Van Den Oord et al., 2017; Ramesh et al., 2021; Yu et al., 2022a; Gafni et al., 2022) addressed this problem by using a discrete variational auto-encoder to learn quantized representations of image patches over a collection of raw images. Instead of learning representations that can take any value in the latent space, a visual codebook is learned that maps a patch embedding to its nearest codebook entry, which is a learned and indexable location in the latent space. These entries can be thought of as visual word *types*, and the appearance of any of these words in a patch in a given image is thus an image *token*.

To be most useful for the second stage model, the image tokenizer needs to learn an effective visual codebook that supports balanced usage of its entries across a broad range of images. It also must support reconstruction of a sequence of visual tokens as a high-quality output image. We use ViT-VQGAN (Yu et al., 2022a) with techniques including $\ell_2$-normalization codes and factorized codes, which contribute to training stability, reconstruction quality and codebook usage. A ViT-VQGAN image tokenizer is trained with the same losses and hyper-parameters as (Yu et al., 2022a) on images of our training data (see Section 4.1). We first train a ViT-VQGAN-Small configuration (8 blocks, 8 heads, model dimension 512, and hidden dimension 2048 as shown in Table 2 of (Yu et al., 2022a), with about 30M total parameters), and learn 8192 image token classes for the codebook. We note that the second stage autoregressive encoder-decoder *training* only relies on the encoder and the codebook of a learned image tokenizer. To further improve visual acuity of the reconstructed images after second-stage encoder-decoder training, we freeze the tokenizer's encoder and codebook, and finetune a larger-size tokenizer decoder (32 blocks, 16 heads, model dimension 1280, and hidden dimension 5120, with about 600M total parameters). We use $256 \times 256$ resolution for the image tokenizer's input and output.

We notice visual pixelation patterns in some of the output images of ViT-VQGAN when zooming in (see Appendix H), and further find ill-conditioned weight matrices of the output projection layer before the sigmoid activation function. As a fix, we remove the final sigmoid activation layer and the logit-laplace loss, exposing

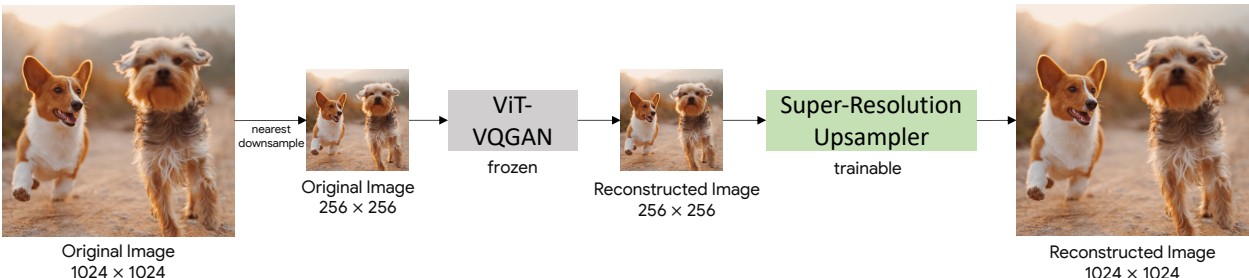

Figure 4: A learned super-resolution module to upsample $256 \times 256$ images to higher-resolution $1024 \times 1024$ ones based on a frozen ViT-VQGAN image tokenizer. The super-resolution module takes $256 \times 256$ images as inputs without conditioning on text inputs.

| Model | Encoder Layers | Decoder Layers | Model Dims | MLP Dims | Heads | Total Params |
|---|---|---|---|---|---|---|
| Parti-350M | 12 | 12 | 1024 | 4096 | 16 | 350M |
| Parti-750M | 12 | 36 | 1024 | 4096 | 16 | 750M |
| Parti-3B | 12 | 36 | 2048 | 8192 | 32 | 3B |
| Parti | 16 | 64 | 4096 | 16384 | 64 | 20B |

Table 1: Size variants of Parti. Both encoder and decoder are based on Transformers (Vaswani et al., 2017). The self-attention layer in decoder transformer is causally masked. Parameters of ViT-VQGAN image tokenization are not included in the total parameter count and can be found in Section 2.1.

the raw values as RGB pixel values (in range $[0, 1]$). Conveniently, this fix can be hot-swappable into an already trained image tokenizer by finetuning the decoder.

Finally, while images of resolution $256 \times 256$ capture most of the contents, structures and textures, higher-resolution images have greater visual impact. To this end, we employ a simple super-resolution module on top of the image tokenizer, shown in Figure 4. Stacked convolutional layers with residual connections are used as the super-resolution network module following WDSR (Yu et al., 2018) (12 residual blocks with 128 channels). It is learned with the same losses of ViT-VQGAN (perceptual loss, StyleGAN loss and $\ell_2$ loss with same loss weighting in (Yu et al., 2022a)), mapping from reconstructed images to higher-resolution reconstructed images. The super-resolution module has about 15M parameters for the $512 \times 512$ version and about 30M parameters for the $1024 \times 1024$ version. We note that diffusion models could also be used here as iterative refinement super-resolution modules, as also demonstrated in DALL-E 2 (Ramesh et al., 2022) and Imagen (Saharia et al., 2022), either with or without conditioning on text inputs.

## 2.2 Encoder-Decoder for Text-to-Image Generation

As shown in Figure 3, a standard encoder-decoder Transformer model is trained at the second stage, by treating text-to-image as a sequence-to-sequence modeling problem. The model takes text as input and is trained using next-token prediction of rasterized image latent codes generated from the first stage image tokenizer. For text encoding, we build a sentence-piece model (Sennrich et al., 2016; Kudo & Richardson, 2018) of vocabulary size 16,000 on a sampled text corpus from the training data (Section 4.1). Image tokens are produced by a learned ViT-VQGAN image tokenizer (see Section 2.1). At inference time, the model samples image tokens autoregressively, which are later decoded into pixels using the ViT-VQGAN decoder.

We use a maximum length of text tokens of 128, and the length of image tokens are fixed to 1024 (*i.e.*, $32 \times 32$ latent codes from a $256 \times 256$ input image). As an example, the 67-word description of the Starry Night prompt given in Figure 1 has a total length of 92 text tokens. All decoder transformers use conv-shaped masked sparse attention (Child et al., 2019) with implementation following DALL-E (Ramesh et al., 2021) (detailed information can be found in Appendix B.1 Figure 11 in (Ramesh et al., 2021)). We train four size

variants ranging from 350 million to 20 billion parameters, as detailed in Table 1. Specifically, we configure the Transformers following previous practice of those in scaling language models with default expansion ratio of $4\times$ in MLP dimensions. We double the number of heads when the model dimension is doubled. In the current scaling variants, our configuration prefers a larger decoder for modeling image tokens and as a result the decoder has more layers (*e.g.*, $3\times$ in the 3B model and $4\times$ in the 20B model).

Most of the existing two-stage text-to-image generation models, including DALL-E (Ramesh et al., 2021), CogView (Ding et al., 2021) and Make-A-Scene (Gafni et al., 2022), are decoder-only models. We found that at the model scale of 350-million to 750-million parameters, the encoder-decoder variants of Parti outperformed decoder-only ones, both in terms of training loss and text-to-image generation quality in our early exploration. We thus chose to focus on scaling the encoder-decoder models.

## 2.3 Text Encoder Pretraining

The encoder-decoder architecture also decouples text encoding from image-token generation, so it is straightforward to explore warm-starting the model with a pretrained text encoder. Intuitively, a text encoder with representations based on generic language training should be more capable at handling visually-grounded prompts. We pretrain the text encoder on two datasets: the Colossal Clean Crawled Corpus (C4) (Raffel et al., 2020) with BERT (Devlin et al., 2019) pretraining objective, and our image-text data (see Section 4.1) with a contrastive learning objective (image encoder from the contrastive pretraining is not used). After pretraining, we continue training both encoder and decoder for text-to-image generation with softmax cross-entropy loss on a vocabulary of 8192 discrete image tokens.

The text encoder after pretraining performs comparably to BERT (Devlin et al., 2019) on GLUE (see Appendix G, Table 9); however, the text encoder degrades after the full encoder-decoder training process on text-to-image generation. We leave this observation as a future research topic on the difference and unification of generic language representation and visually-grounded language representation. Still, the text-encoder pretraining *marginally* helps text-to-image generation loss with 3B-parameter Parti models, so pretraining is used by default in our 20B model. We provide detailed training loss, GLUE evaluation of text encoders, and some qualitative comparison in Appendix G.

## 2.4 Classifier-Free Guidance and Reranking

Classifier-free guidance (Ho & Salimans, 2021) (CF-guidance in short) is critical in the context of improving the sample quality of diffusion models (Nichol et al., 2022; Ramesh et al., 2022; Saharia et al., 2022) without pretrained classifiers. In this setup, a generative model $G$ is trained to be able to perform unconditional generation $G(\mathbf{z})$ (where $\mathbf{z}$ represents random noise) and conditional generation $G(\mathbf{z}, \mathbf{c})$ (where $\mathbf{c}$ represents some condition, such as language descriptions). It is implemented as randomly dropping out the conditional vector (masking out or switching to a learned embedding) with some probability. During the inference process, sampling of an output $I$ is done by using a linear combination of the unconditional and conditional predictions:

$$I = G(\mathbf{z}) + \lambda(G(\mathbf{z}, \mathbf{c}) - G(\mathbf{z})), \tag{1}$$

where $\lambda$ is a hyperparameter representing the weight of classifier-free guidance. Intuitively, it decreases the unconditional likelihood of the sample while increasing the conditional likelihood, which can be viewed as encouraging alignment between the generated sample and the text condition.

Classifier-free guidance has been similarly applied in the context of autoregressive models for text-to-image generation (Crowson, 2021; Gafni et al., 2022) to great effect. Make-A-Scene (Gafni et al., 2022) finetunes the model by randomly replacing the text prompts with padded tokens. During inference, tokens are sampled from a linear combination of logits sampled from an unconditional model and a conditional model on a text prompt. We also apply CF-guidance in Parti, and find it has a significant improvement on the output image-text alignment, especially on challenging text prompts. For the unconditional inputs, we simply set text token ids to zeros and text token paddings as ones.

With batch-sampled images per text prompt, contrastive reranking is used in DALL-E (Ramesh et al., 2021) which produces image-text alignment scores after the generation. We apply contrastive reranking in

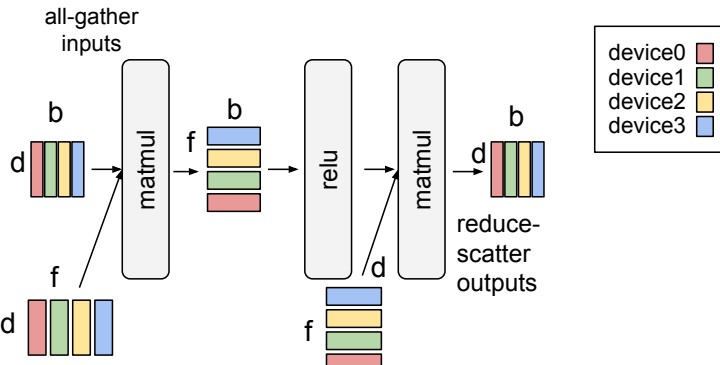

Figure 5: 4-way in-layer model parallelism with fully partitioned activations used to scale the 3B model training. The figure shows a simplified Transformer feed-forward layer (with the sequence dimension omitted); each color represents data on one device. We also use 128-way data parallelism.

our work and find it is complementary to classifier-free guidance. Compared with the 512 images used in DALL-E (Ramesh et al., 2021), we sample just 16 images per text prompt for the experiments reported in this paper. We rerank each output set based on the alignment score of image and text embedding of a Contrastive Captioners model (CoCa) (Yu et al., 2022b). A CoCa base-size model (Table 1 in (Yu et al., 2022b)) is trained on the same dataset with details in Section 4.1. We note that reranking over a small set of batch-sampled images is computationally cheap in the text-to-image sampling process, and produces helpful image-text alignment scores among diverse image outputs.

## 3   Scaling

We implement our models in Lingvo (Shen et al., 2019) and scale with GSPMD (Xu et al., 2021) on CloudTPUv4 hardware for both training and inference. GSPMD is an XLA compiler-based model partitioning system that allows us to treat a cluster of TPUs as a single virtual device and use *sharding annotations* on a few tensors to instruct the compiler to automatically distribute data and compute on thousands of devices.

**Training.** We train both 350M and 750M models simply with data parallelism. For the 3B model, we use 4-way in-layer model parallelism (see Figure 5), and 128-way data parallelism. Partitioning a single dimension in each tensor is sufficient to scale a 3B model. The model weights are partitioned on the feed-forward hidden dimension and the number of attention heads dimension; the internal activation tensors of the feed-forward and attention layers are also partitioned on the hidden and heads dimensions. One difference from Megatron-LM (Shoeybi et al., 2019) is we fully partition the output activations of feed-forward and attention layers on a different dimension, with the details illustrated as the *finalized 2d sharding* in the GSPMD work (Xu et al., 2021). This strategy will result in `ReduceScatter` and `AllGather` communication patterns instead of `AllReduce`, which significantly reduce peak activation memory.

The 20B model has 16 encoder layers, and 64 decoder layers (see Table 1). The size of the weights per layer is moderate (as opposed to being very wide), which makes pipeline parallelism (Huang et al., 2018) a good option for scaling. We use a generic pipelining wrapper layer allowing us to specify a single-stage program, which will later be automatically transformed into a multi-stage pipelining program; the wrapper layer uses vectorization and shifting buffers to reduce pipelining into a tensor partitioning problem (see Section 3.3 of (Xu et al., 2021)). Thus, all lower-level infrastructure can be reused for pipelining. There are two additional benefits in adopting GSPMD pipelining: 1) it allows us to conveniently configure pipelines within model sub-components, simplifying the overall complexity for encoder-decoder models, and 2) since pipelining is implemented as tensor partitioning on vectorized programs, we can reuse the same set of devices for other types of parallelism outside the transformer layers.

We configure the model to have separate encoder and decoder pipelines, each with 16 stages. We also use 64-way data parallelism in addition to pipelining. However this makes per-core batch size small, exposing

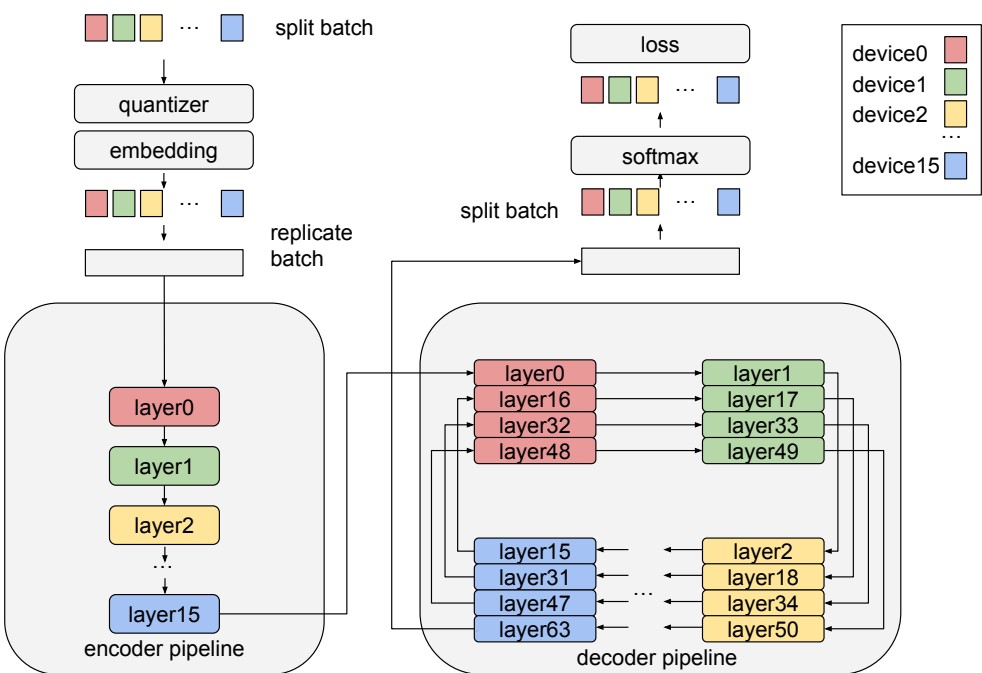

Figure 6: An illustration of 16-stage GSPMD pipelines used to scale the 20B model training. The figure shows how the 16 devices are used for data parallelism in the quantizer, embedding and softmax layers, but repurposed for pipelining in the encoder and decoder layers. Each color represents data or layer assigned to one device. The decoder uses 4-round circular schedule to further reduce the pipeline bubble ratio. On top of this, we use additional 64-way data parallelism for all layers.

an additional challenge of excessive pipeline stalls due to inter-stage data dependency (known as *bubbles* in pipeline parallelism (Huang et al., 2018)). To reduce the ratio of bubbles, we adapt the *circular schedule* as described in (Xu et al., 2021) in the decoder pipeline (a similar technique was also proposed in (Narayanan et al., 2021)), where the 4 layers in each stage are executed in a round-robin order. Outside the encoder and decoder, we use the same set of devices to do data parallelism instead of pipelining for the embedding, softmax, and image tokenizer layers. Figure 6 illustrates the overall distributed training strategy.

During training, Adafactor (Shazeer & Stern, 2018) optimizer is used to save memory with $\beta_1 = 0.9$, $\beta_2 = 0.96$ and decoupled weight decay value of $4.5 \times 10^{-2}$. The first moments of optimizer slot variables are also quantized from float32 to int8. We use default dropout ratio 0.1 for all models in both encoder and decoder. A deterministic version of dropout layer as well as a vectorized version of Adafactor optimizer are used in the 20B model to enable training pipelined models. Data types are cast to bfloat16 for attention projection and feed-forward transformers layers, while all layer norms and model output are kept as float32. We use a default learning rate of 4.5e-5 and exponential learning rate schedule with 5,000 warm-up steps. Exponential decaying starts at training steps 85,000 with a total of 450,000 steps and final ratio of 0.025. We use a global batch size of 8192 during training. We do not use exponential moving average of the model weights to save device memory. Conv-shaped sparse attention is used in the decoder transformers, similar to DALL-E (Ramesh et al., 2021) (Appendix B.1. Architecture, Fig. 11). We additionally clip gradient norm to a value of 4.0 to stabilize the training, especially at the beginning. At the output of both the encoder and decoder, we apply an additional layer normalization.

**Inference.** Our primary goal for inference optimization is to speed up small-batch image generation. We choose in-layer model parallelism for both the 3B and 20B models. As opposed to training, we do not fully partition the output activations for feed-forward and attention layers for inference; this is because 1) each step of the autoregressive decoding produces much smaller tensors and (at the time of writing) `AllReduce`

| Dataset | Train | Val | AvgWords | Caption | Image |
|---|---|---|---|---|---|
| MS-COCO (2014) (Lin et al., 2014) | 82K | 40K | 10.5 | *"A bowl of broccoli and apples with a utensil."* | |
| Localized Narratives (COCO subset) (Pont-Tuset et al., 2020) | 134K | 8K | 42.1 | *"In this picture, we see a bowl containing the chopped apples and broccoli. In the background, we see a white table on which seeds or grains, broccoli, piece of fruit, water glass and plates are placed. This table is covered with a white and blue color cloth. This picture is blurred in the background."* | |

Table 2: Evaluation data statistics and examples. Images from the COCO portion of Localized Narratives come from the MS-COCO (2017) set; Localized Narratives descriptions are four times the length of captions in MS-COCO on average. The example above highlights the massive difference in detail between MS-COCO and Localized Narratives for the *same* image.

performs better on small data, 2) activation memory is not a concern during inference, which does not have a backward pass.

## 4 Training and Evaluation Datasets

### 4.1 Training Datasets

We train on a combination of image-text datasets for all Parti models. The data includes the publicly available LAION-400M dataset (Schuhmann et al., 2021); FIT400M, a filtered subset of the full 1.8 billion examples used to train the ALIGN model (Jia et al., 2021a); JFT-4B dataset (Zhai et al., 2022), which has images with text annotation labels. For textual descriptions of JFT, we randomly switch between the original labels as text (concatenated if an image has multiple labels) or machine-generated captions from a SimVLM-Huge model (Wang et al., 2022). We discuss the limitations of the data in Section 8. For all image inputs, we follow the DALL-E dVAE input processing (Section A.2. Training in (Ramesh et al., 2021)) for image tokenizer training and the DALL-E Transformer input processing (Section B.2. Training in (Ramesh et al., 2021)) for encoder-decoder training.

### 4.2 Evaluation Datasets

We evaluate our models on MS-COCO (2014) (Lin et al., 2014) and Localized Narratives (Pont-Tuset et al., 2020), summarized in Table 2. MS-COCO is the current standard dataset for measuring both zero-shot and finetuned text-to-image generation performance, which makes it a consistent point of comparison with prior work. However, MS-COCO captions are short, high-level characterizations of their corresponding images. For a more comprehensive evaluation, we also use the COCO portion of Localized Narratives (LN-COCO), which provides longer, detailed descriptions of images corresponding to the MS-COCO (2017) dataset, and compare Parti's performance on LN-COCO against (Koh et al., 2021; Zhang et al., 2021). These long-form descriptions are typically quite different from the descriptions used to train large text-to-image generation models. This provides a measure of generalization to out-of-domain distributions, as well as the finetuning capability of these models. Regardless of the community's current focus on zero-shot performance, the ability to finetune effectively is also important for adapting an open-domain text-to-image generation model to work with a specific application area or domain.

### 4.3 PartiPrompts

Existing benchmarks like MS-COCO (Lin et al., 2014) and Localized Narratives (Pont-Tuset et al., 2020) are clearly useful for measuring the progress of text-to-image synthesis systems, but the descriptions available in them are generally limited to everyday scenes and objects found in natural images. This limits their

| Category | Examples |
|---|---|
| ABSTRACT | *commonsense; happiness; hope; insight; 300; 101; golden ratio; Fibonacci number* |
| ANIMALS | *a Tyrannosaurus Rex; 7 dogs sitting around a poker table, two of which are turning away; a white rabbit in blue jogging clothes doubled over in pain while a turtle wearing a red tank top dashes confidently through the finish line* |
| ARTS | *a dutch baroque painting of a horse in a field of flowers; a painting of a fox in the style of starry night; A raccoon wearing formal clothes, wearing a top hat and holding a cane. The raccoon is holding a garbage bag. Oil painting in the style of Vincent Van Gogh.* |
| ILLUSTRA-TIONS | *concurrent lines; a diagram of brain function; concentric squares fading from yellow on the outside to deep orange on the inside; a drawing of a series of musical notes wrapped around the Earth* |
| OUTDOOR SCENES | *a grand piano next to the net of a tennis court; a white country home with a wrap-around porch; beautiful fireworks in the sky with red, white and blue; a peaceful lakeside landscape with migrating herd of sauropods* |
| PEOPLE | *the Beatles crossing Abbey road; a woman with long hair next to a luminescent bird; a tall man stooping down to enter a low red sports car; a politician wearing a soccer jersey and holding a volleyball while giving a speech on a stage* |
| VEHICLES | *an F1; a boat in the canals of venice; a car with tires that have yellow rims; a blue semi-truck and its trailer jumping over a row of motorcycles. there are metal ramps on either side of the motorcycles.* |
| WORLD KNOWL-EDGE | *U.S. 101; the skyline of New York City; An aerial view of Ha Long Bay without any boats; view of the Great Wall from its base; A close-up high-contrast photo of Sydney Opera House sitting next to Eiffel tower, under a blue night sky of roiling energy, exploding yellow stars, and radiating swirls of blue.* |

Table 3: Sample categories in the PartiPrompts (P2) benchmark. Examples (separated by semicolons) range from abstract concepts such as "golden ratio" to concrete ones such as "the skyline of New York City". For full descriptions of all categories, see Appendix D.

| Challenge | Examples |
|---|---|
| BASIC | *a rabbit; U.S. 101; a margarita; lily pads; brain coral; The Alamo; a family* |
| COMPLEX | *the Sydney Opera House with the Eiffel tower sitting on the right, and Mount Everest rising above; a white rabbit in blue jogging clothes doubled over in pain while a turtle wearing a red tank top dashes confidently through the finish line; Oil-on-canvas painting of a blue night sky with roiling energy. A fuzzy and bright yellow crescent moon shining at the top. Below the exploding yellow stars and radiating swirls of blue, a distant village sits quietly on the right. Connecting earth and sky is a flame-like cypress tree with curling and swaying branches on the left. A church spire rises as a beacon over rolling blue hills.* |
| IMAGINATION | *a four-eyed horse; a toaster shaking hands with a microwave; a peaceful lakeside landscape with migrating herd of sauropods; a flower with large red petals growing on the moon's surface; A rusty spaceship blasts off in the foreground. A city with tall skyscrapers is in the distance, with a mountain and ocean in the background. A dark moon is in the sky. realistic high-contrast anime illustration.* |
| LINGUISTIC STRUCTURES | *Incomprehensibilities; Pneumonoultramicroscopicsilicovolcanoconiosis; The horse raced past the barn fell; One morning I chased an elephant in my pajamas; The dog chased the cat, which ran up a tree. It waited at the top; The dog chased the cat, which ran up a tree. It waited at the bottom.* |
| PERSPECTIVE | *the back of a violin; an extreme close-up view of a capybara sitting in a field; tall buildings seen through a window with rain on it; view from below of a tall white ladder with just one rung leaning up against a yellow brick wall* |
| QUANTITY | *an owl family; 7 dogs sitting around a poker table, two of which are turning away; a basketball game between a team of four cats and a team of three dogs* |
| WRITING & SYMBOLS | *a grumpy porcupine handing a check for $10,000 to a smiling peacock; a group of cats in a meeting. there is a whiteboard with "stack more layers" written on it; The saying "BE EXCELLENT TO EACH OTHER" written on a red brick wall with a graffiti image of a green alien wearing a tuxedo. A yellow fire hydrant is on a sidewalk in the foreground.* |

Table 4: Sample challenge aspects in the P2 benchmark. Examples (separated by semicolons) range from basic to complex ones such as a full description of the Starry Night "Oil-on-canvas painting of a blue night sky ... rolling blue hills". For full descriptions of all challenge aspects, see Appendix D.

representation of a broad spectrum of prompts – in particular, they lack prompts that allow us to better probe model capabilities on open-domain text-to-image generation. For example, MS-COCO captions are brief characterizations of high level participants and actions in images; these typically cover common scenarios and are oriented toward objects. Localized Narratives has highly-detailed descriptions, but also emphasizes natural scenes and objects. Recently, the work by (Park et al., 2021) focuses on the text-to-image generation task, but is limited to only two scenarios, unseen object-color (*e.g.*, "blue petal") and object-shape (*e.g.*, "long beak"). Motivated by these shortcomings, we present PartiPrompts (P2), a set of 1600 diverse English prompts that allow us to more comprehensively evaluate and test the limits of text-to-image synthesis models.

Each prompt in the P2 benchmark is associated with two labels: (1) *Category*, indicating a broad group that a prompt belongs to, and (2) *Challenge*, highlighting an aspect which makes a prompt difficult. Table 3 provides a few samples of categories (out of 12 options) used in P2, ranging from abstract concepts such as the "golden ratio" to concrete world-knowledge ones such as "the skyline of New York City". Similarly, Table 4 lists a sample of challenge aspects (out of 11), ranging from basic ones such as "a rabbit" to complex ones such as a full description of the painting *Starry Night* ("Oil-on-canvas painting of a blue night sky ... A church rises as a beacon against rolling blue hills."). For example, the prompt "a peaceful lakeside landscape with migrating herd of sauropods" is categorized as OUTDOOR SCENES, while its challenge aspect is IMAGINATION. Similarly, the prompt "7 dogs sitting around a poker table, two of which are turning away" has ANIMALS as category and QUANTITY as challenge aspect. These two views of a prompt allows us to analyze a model's capabilities from both aspects–the overall content generated and the subtle details captured.

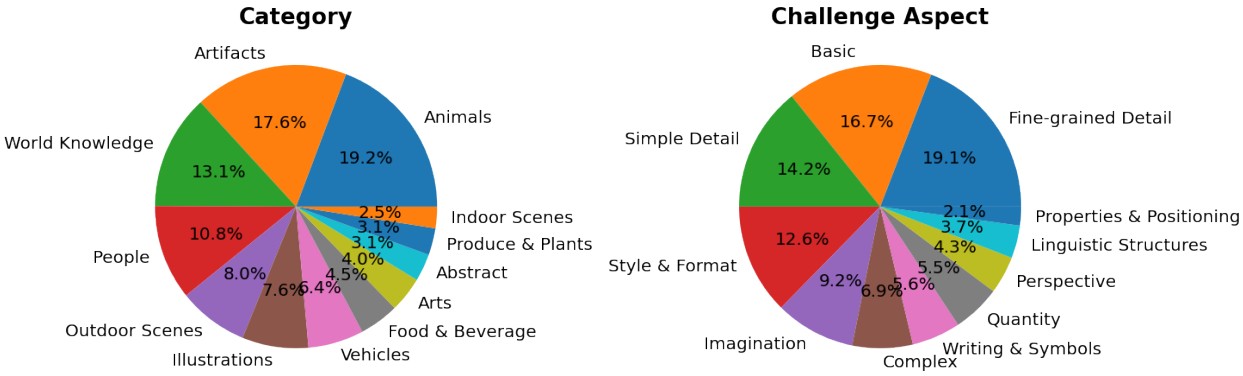

Figure 7: A summary of the PartiPrompts (P2) set of 1600 descriptions, spanning across many *category labels* (left) and *challenge aspects* (right).

We created PartiPrompts both by thinking of novel prompts and by manually curating and sampling prompts from recent papers (Ramesh et al., 2021; Ding et al., 2021; Gu et al., 2022; Nichol et al., 2022; Ramesh et al., 2022) (which accounts for about 7% of prompts in P2). While it is possible to assign multiple categories and challenge aspects to a prompt, we chose to reduce the complexity of model analysis by manually deciding on a single primary category and challenge aspect for each prompt. For example, when there is a proper noun, we will prefer to label the prompt as WORLD KNOWLEDGE (e.g., "a painting of street in Paris") over other categories, e.g., ARTS. We also prioritize categories that have fewer examples, *e.g.*, ARTS, over those with plenty, such as ANIMALS, as in the case of the prompt "A raccoon wearing formal clothes, wearing a tophat and holding a cane. The raccoon is holding a garbage bag. Oil painting in the style of Vincent Van Gogh." The PEOPLE category always takes priority, *e.g.*, "a team playing baseball at the beach" is labeled as PEOPLE instead of OUTDOOR SCENES; we do this to ease future work on fairness and bias interested in using PartiPrompts, as it can be easily split to include or exclude prompts involving people. Figure 7 highlights the distribution of category labels and challenge aspects over the 1600 prompts. One can group these prompts into levels of difficulties by the challenge aspects: *Standard* includes BASIC and SIMPLE DETAIL (about 1/3 of the prompts); *Intermediate* includes FINE-GRAINED DETAIL and STYLE & FORMAT (also about 1/3 of the prompts); and *Challenging* includes the remaining 7 challenge aspects such as IMAGINATION, QUANTITY, COMPLEX, and LINGUISTIC STRUCTURES.

It is also worth mentioning DrawBench, a concurrently developed benchmark of 200 prompts introduced in (Saharia et al., 2022). It has eleven labels that mix both categories (*e.g.*, "DALL-E") and challenging aspects (*e.g.*, "Counting"). PartiPrompts, in contrast, teases apart these two dimensions, with 12 categories and 11 challenging aspects, allowing for richer categorizations of prompts and more fined-grained analyses, together with 8× more prompts. Both benchmarks contain prompts that present strong challenges for current best models–including DALL-E 2, Imagen and Parti–and hopefully will inspire further benchmarks to increase the level of difficulty as future models continually improve.

## 5  Experiments

We conduct automatic evaluations on both MS-COCO and Localized Narratives to compare with previous work. On MS-COCO and PartiPrompts, we also obtain human side-by-side evaluations for Parti 20B to compare with a strong retrieval baseline, as well as the XMC-GAN model (Zhang et al., 2021), which has the best FID out of all publicly available models at the time of writing. We also conduct human evaluation on two Parti models with parameters 3B and 20B on PartiPrompts, and provide a detailed breakdown on categories. By default, Parti samples 16 images per text prompt and uses a CoCa model to rank the outputs (see Section 2.4).

### 5.1  Retrieval Baseline

Perhaps the most compelling use for text-to-image generation models is creating novel images for situations that have never been depicted. Strong models should thus be more effective than an approach that simply retrieves candidate images from a large dataset. We implement a retrieval baseline as follows. For every training data example, we compute image embeddings from an EfficientNet-L2 ALIGN-based model (Jia et al., 2021b). Then, given a text prompt, we identify the nearest training image measured by the alignment between the text prompt embedding from the ALIGN-based model and the image embeddings. This can be done at the scale of our data by using efficient similarity search libraries such as ScaNN (Guo et al., 2020). These retrieved examples (from the training set) are then provided as the output of the baseline, to be evaluated by images actually generated by our models. We manually visualize the retrieved images given the text prompts and observe that this retrieval approach represents a high-quality baseline, especially for common text descriptions.

To compare with Parti generated images, we report retrieval baseline results under two settings that we characterize as *zero-shot* and *finetuned* to align with the model evaluation terminology. For MS-COCO, retrieval over our training data is "zero-shot", while retrieval over MS-COCO's train split is "finetuned" – corresponding to out-of-dataset and in-dataset retrieval, respectively. We compare Parti generated images with retrieved images using both automated measures and human evaluation for both image realism and image-text alignment.

### 5.2  Evaluation Metrics

We evaluate using two primary axes: (1) generated image quality, and (2) alignment of the generated image with the input text. We report both automated quantitative metrics and human evaluation results. In addition, we show example model outputs for qualitative assessment and comparison.

**Automatic image quality.** Similar to prior work in text-to-image generation, we use the Fréchet Inception Distance (FID) (Heusel et al., 2017) as the primary automated metric for measuring image quality.[2] FID is computed by running generated and real images through the Inception v3 (Szegedy et al., 2016) model, and extracting features from the last pooling layer of the model. The Inception features of the generated and real images are used to fit two separate multi-variate Gaussians. Finally, the FID score is computed by measuring the Fréchet distance between the two multivariate Gaussian distributions. Following (Xu et al., 2018; Zhang et al., 2021; Ramesh et al., 2021), we use 30,000 generated and real image samples for evaluation on MS-COCO (2014) using the same DALL-E input preprocessing (Section B.2. Training in (Ramesh et al.,

---

[2]We use `https://github.com/mseitzer/pytorch-fid` for computing FID scores.

| Approach | Model Type | MS-COCO FID (↓) | | LN-COCO FID (↓) | |
|---|---|---|---|---|---|
| | | Zero-shot | Finetuned | Zero-shot | Finetuned |
| Random Train Images (Gafni et al., 2022) | - | 2.47 | | - | |
| Retrieval Baseline | - | 17.97 | 6.82 | 33.59 | 16.48 |
| TReCS (Koh et al., 2021) | GAN | - | - | - | 48.70 |
| XMC-GAN (Zhang et al., 2021) | GAN | - | 9.33 | - | 14.12 |
| DALL-E (Ramesh et al., 2021) | Autoregressive | ∼28 | - | - | - |
| CogView (Ding et al., 2021) | Autoregressive | 27.1 | - | - | - |
| CogView2 (Ding et al., 2022) | Autoregressive | 24.0 | 17.7 | - | - |
| GLIDE (Nichol et al., 2022) | Diffusion | 12.24 | - | - | - |
| Make-A-Scene (Gafni et al., 2022) | Autoregressive | 11.84 | 7.55 | - | - |
| DALL-E 2 (Ramesh et al., 2022) | Diffusion | 10.39 | - | - | - |
| Imagen (Saharia et al., 2022) | Diffusion | **7.27** | - | - | - |
| Parti | Autoregressive | **7.23** | **3.22** | **15.97** | **8.39** |

Table 5: Comparison with previous work on the MS-COCO (2014) (Lin et al., 2014) and Localized Narratives (COCO split) (Pont-Tuset et al., 2020) validation sets. When available, we report results for both zero-shot and finetuned models. Retrieval models either perform retrieval over our training set ("zero-shot"), or the respective MS-COCO and LN-COCO training sets ("finetuned"). Parti samples 16 images per text prompt and uses a CoCa model to rank the outputs (Section 2.4). Similar to DALL-E 2 (Ramesh et al., 2022), we use guidance scale 1.2 for all above results. We report zero-shot FID score of other model sizes in Figure 9.

2021)) with 256×256 image resolution. The validation set of the Localized Narratives COCO split contains only 5,000 unique images, so we follow (Zhang et al., 2021) in oversampling the captions to acquire 30,000 generated images.

**Automatic image-text alignment.** Following DALL-Eval (Cho et al., 2022), we also measure text-image fit through automated captioning evaluation (or captioner evaluation): an image output by the model is captioned with a pretrained VL-T5 model (Cho et al., 2021) and then the similarity of the input prompt and the generated caption is assessed via BLEU (Papineni et al., 2002), CIDEr (Vedantam et al., 2015), METEOR (Denkowski & Lavie, 2014) and SPICE (Anderson et al., 2016).

**Human side-by-side.** We follow previous work (Zhang et al., 2021; Ramesh et al., 2021) in doing side-by-side evaluations in which human annotators are presented with two outputs for the same prompt and are asked to choose which output is a higher quality image (generally, better with respect to image realism) and which is a better match to the input prompt (image-text alignment). The models are anonymized and the pairs are randomly ordered (left *vs.* right) for each presentation to an annotator, and each pair is judged by five independent annotators. We graphically show the gradual breakdown of results for each model in terms of the number of examples where it obtains 0, 1, 2, 3, 4 or 5 votes. In addition, we highlight the percentage of examples where each model has obtained the majority (three or more votes), as a summary of the comparison. See Appendix E for a screenshot of our annotator interface.

### 5.3 Main Results

Table 5 presents our main results of automated image quality evaluation. Parti achieves a comparable zero-shot FID score 7.23 compared with diffusion-based model Imagen (Saharia et al., 2022). When finetuned, Parti achieves state-of-the-art FID score of 3.22, a dramatic improvement over previous best finetuned FID 7.55 from an autoregressive model, Make-a-Scene (Gafni et al., 2022). It is also better than the in-dataset retrieval baseline, with an FID score 6.82. We note that the retrieval baseline is worse than using 30,000 random samples from MS-COCO real training set images – which obtains an FID of 2.47. The root cause is that the retrieval model often selects the same images for similar types of prompts, leading to duplicates in the retrieved images for evaluation. For example, there are only 17,782 unique retrieved MS-COCO train images for 30,000 validation text prompts, leading to worse diversity and poorer FID score as compared to the 30,000 random samples from the train set. We also show qualitative comparisons (Appendix C, Figure 24) of non-cherry-picked Parti sampled images along with outputs of other approaches (Ramesh et al., 2021; Nichol et al., 2022; Gafni et al., 2022; Ramesh et al., 2022) on MS-COCO prompts. Parti demonstrates

| Approach | BLEU (↑) | METEOR (↑) | CIDEr (↑) | SPICE (↑) |
|---|---|---|---|---|
| Random Train Images | 4.4 | 9.2 | 4.8 | 2.0 |
| Retrieval Baseline | 24.7 | 23.9 | 84.1 | 16.6 |
| Ground Truth (upper bound) | 32.5 | 27.5 | 108.3 | 20.4 |
| DALL-E[Small5] | 9.3 | 12.9 | 20.2 | 5.6 |
| ruDALL-E-XL[6] | 13.9 | 16.0 | 38.7 | 8.7 |
| minDALL-E (Kim et al., 2021) | 16.6 | 17.6 | 48.0 | 10.5 |
| X-LXMERT (Cho et al., 2020) | 18.5 | 19.1 | 55.8 | 12.1 |
| Parti | **26.4** | **23.9** | **83.9** | **16.5** |

Table 6: Comparison with prior work on captioner evaluation on the MS-COCO 5K test set (Karpathy & Fei-Fei, 2017) with baselines from DALL-Eval (Cho et al., 2022). Ground Truth represents the theoretical upper bound on this evaluation with captions generated using MS-COCO images as inputs to the VL-T5 model (Cho et al., 2021). Parti samples 16 images per text prompt and uses a CoCa (Yu et al., 2022b) model to rank the outputs (Section 2.4).

strong generalization without finetuning on specific domains like MS-COCO, and it achieves a high degree of image realism – often very close to that of real images.

For LN-COCO, Parti achieves a finetuned FID score of 8.29, which is a massive improvement over XMC-GAN's finetuned result of 14.12 and the retrieval baseline's 16.48. Moreover, Parti achieves a zero-shot FID score of 15.97, which nearly matches XMC-GAN's finetuned score (trained on LN-COCO set). We visualize and compare side-by-side with XMC-GAN and find the zero-shot images produced by Parti are qualitatively much better in realism and image-text fit compared to images produced by XMC-GAN, which we offer as a cautionary tale that researchers should not rely solely on FID for comparison of text-to-image generation models.

## 5.4 More Results on MS-COCO

**Automatic image-text alignment evaluation.** Table 6 provides results of Parti on the captioner evaluation (Cho et al., 2022) as an automatic image-text alignment measure. Parti outperforms other models on this measure, and it closes much of the gap to the scores obtained for captions generated from the ground truth images. The retrieval baseline performs comparable to Parti. Unlike FID scores, random train images perform considerably worse on the captioner evaluation, as expected. The captioner evaluation complements FID score evaluation as an automatic image-text alignment measurement for text-to-image generation models; however, we also note these results are limited by the captioner model's (Cho et al., 2021) ability to discriminate between outputs from different approaches.

**Human evaluations.** For MS-COCO, we compare our *zero-shot* generation results against the *finetuned* XMC-GAN (Zhang et al., 2021) model, which has best the FID out of all publicly available models with available images of the same MS-COCO prompts, at the time of writing. For each prompt, the output from Parti and XMC-GAN are anonymized and shown to 1,000 independent human evaluators. The results are summarized and shown in Figure 8. Even though Parti is not trained on MS-COCO captions or images, our results are overwhelmingly preferred by human annotators over XMC-GAN outputs: 91.7% preference score for image realism and 90.5% for image-text match.[5] When compared against the retrieval model on MS-COCO, Parti is evaluated as slightly worse on image realism (45.2% compared to 54.8%) but evaluated as better on image-text match (55.2% compared to 44.8%). This shows that in nearly half of the comparisons between Parti's output and *real* images, the former *generated* images were judged as more realistic by

---

[5] https://github.com/lucidrains/DALLE-pytorch

[6] https://rudalle.ru

[5] We analyzed the cases in which XMC-GAN is rated as more realistic compared to Parti, and found that most of these examples were due to Parti producing illustrations or cartoons, rather than photo-realistic images. While these were generally well aligned with the given prompts, the evaluation likely disadvantages Parti since MS-COCO is entirely focused on photographs and descriptions of them.

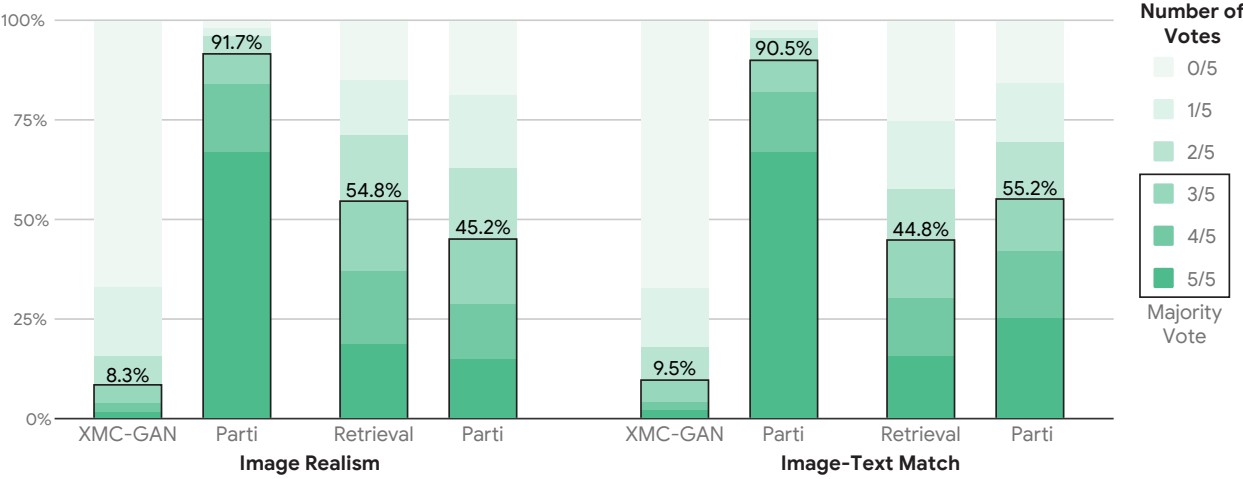

Figure 8: Human evaluation results over 1,000 randomly sampled prompts from the MS-COCO (2014) validation set. Each prompt is rated by 5 independent human evaluators. The zero-shot Parti models are used in all comparisons. Our model significantly outperforms XMC-GAN (Zhang et al., 2021), despite the latter being finetuned on MS-COCO. When compared against the retrieval model (retrieval over about 4B training images), Parti is better on image-text match, but worse on image realism (as retrieved images are real images).

people—a strong statement of the visual quality of images produced by the model. The fact that Parti outputs are preferred for image-text match shows that generation is an important means of producing accurate visual depictions of even the mostly quotidian scenes described in MS-COCO captions.

| MS-COCO (zero-shot) | |
|---|---|
| Parameters | FID ↓ |
| 350M | 14.10 |
| 750M | 10.71 |
| 3B | 8.10 |
| 20B | 7.23 |

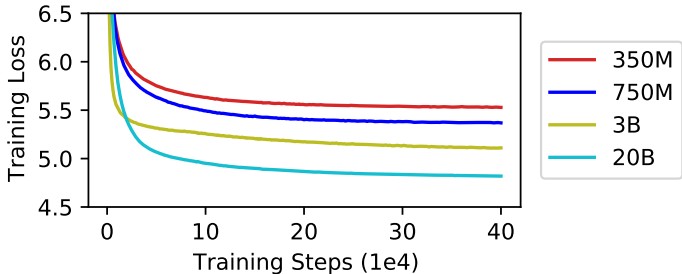

Figure 9: Effects of scaling Parti models of different sizes. We show zero-shot FID scores *(left)* on MS-COCO (2014) and the training loss curves of the corresponding models *(right)*.

**Comparison of Model Scaling.** We compare four different model sizes of Parti, with parameter counts ranging from 350M, 750M to 3B and 20B, as shown in Table 1. All four models are trained on the same mixture of datasets with the same image tokenizer and CoCa reranking model described in Section 2.4. Figure 9 summarizes the corresponding zero-shot FID scores on MS-COCO (2014). Parti models are trained with next token prediction loss for text-to-image generation, using softmax cross-entropy loss over a 8192-vocab image codebook. The loss is averaged by 1024 (the total output length) image tokens per example. We observe better training loss as well as zero-shot FID on MS-COCO when we scale up the model. Specifically, a significant quality jump is achieved by scaling model from 750M to 3B; furthermore, the 20B model outperforms 3B model in more challenging prompts (*e.g.*, text rendering). We highlight qualitatively how these models perform visually in Figures 10 and 13, using challenging prompts from the P2 benchmark (see Section 5.5).

**Parti-350M**        **Parti-750M**        **Parti-3B**        **Parti-20B**

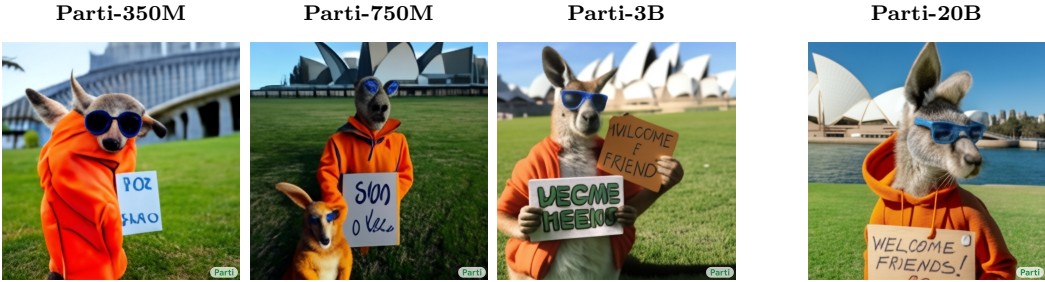

A portrait photo of a kangaroo wearing an orange hoodie and blue sunglasses standing on the grass in front of the Sydney Opera House holding a sign on the chest that says Welcome Friends!

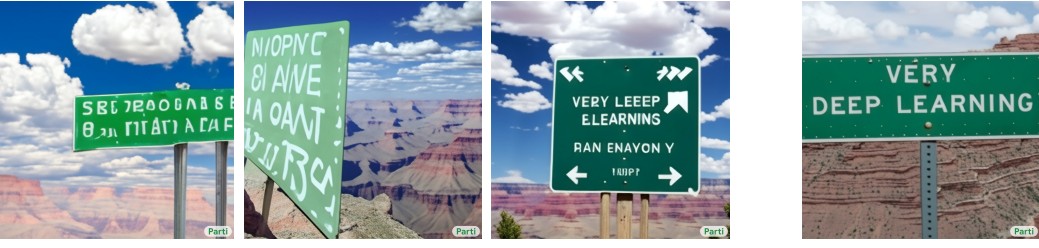

A green sign that says "Very Deep Learning" and is at the edge of the Grand Canyon. Puffy white clouds are in the sky.

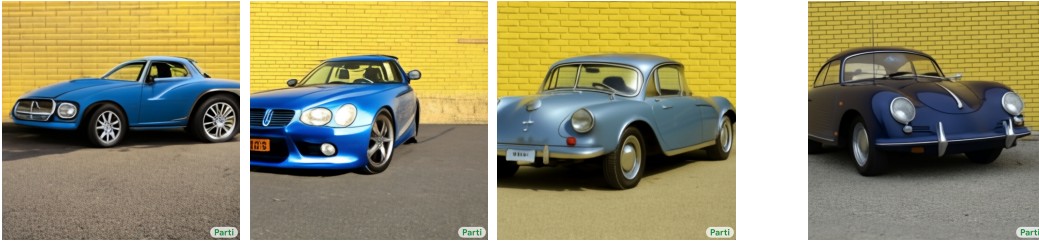

A blue Porsche 356 parked in front of a yellow brick wall.

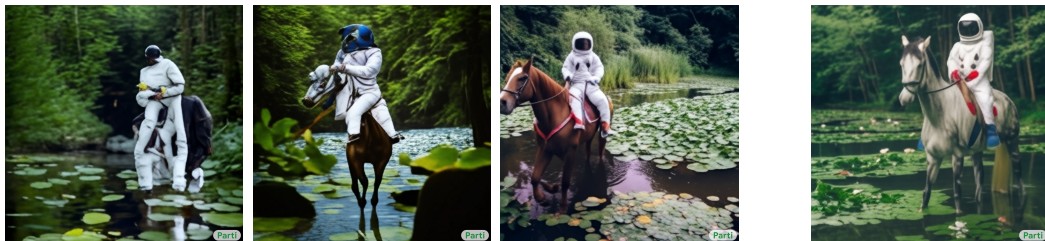

A photo of an astronaut riding a horse in the forest. There is a river in front of them with water lilies.

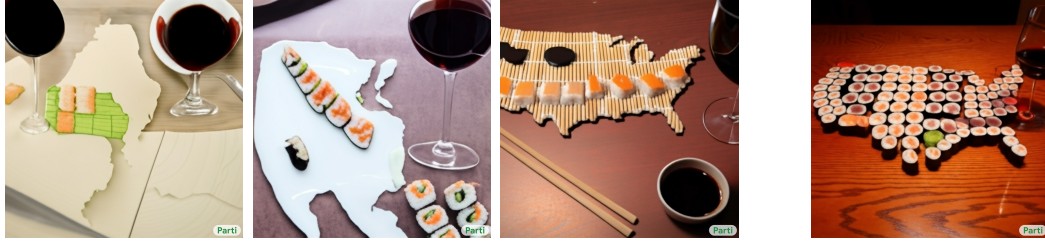

A map of the United States made out of sushi. It is on a table next to a glass of red wine.

Figure 10: Qualitative comparison of top-1 images sampled from Parti models of increasing sizes (350M, 750M, 3B, 20B). All Parti models sample 16 images per text prompt and rerank using the same CoCa model described in Section 2.4. We use prompts in the P2 benchmark (Section 4.3) to test WORLD KNOWLEDGE, FINE-GRAINED DETAIL, and WRITING & SYMBOLS.

## 5.5    Results on PartiPrompts

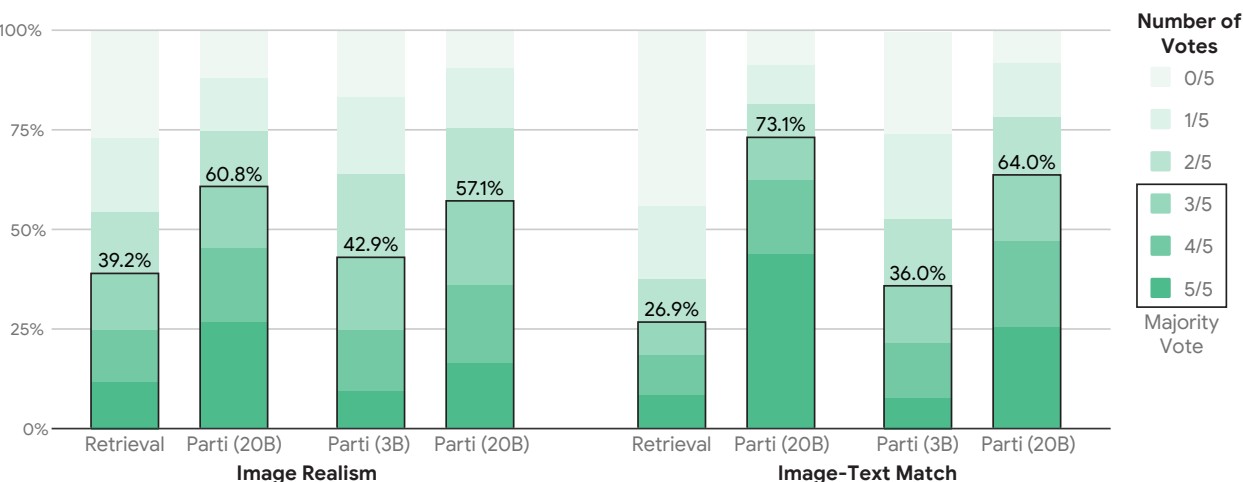

Figure 11: Human evaluation results on PartiPrompts.

**Human Evaluations.** In addition to MS-COCO, we also conduct human evaluations on the P2 benchmark, comparing our 20B model against the 3B variant and the Retrieval baseline. Figure 11 shows that the 20B model is clearly preferred by annotators over the retrieval baseline both in terms of image realism (63.2%) and image-text match (75.9%). These results offer a complementary view to the comparison in Figure 8: on the much more challenging P2 benchmark (Section 4.3), the retrieval baseline was unable produce matching outputs for many prompts. The 3B model closes the gap but the 20B is still preferred in terms of image realism (56.8%) and image-text match (62.7%).

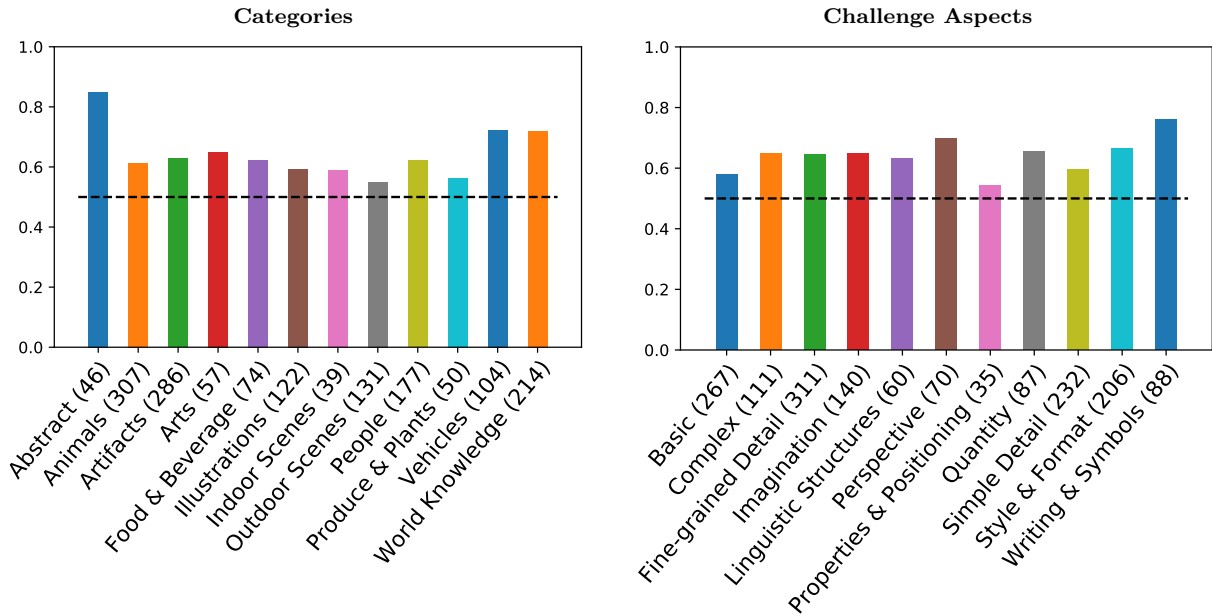

Figure 12: Breakdown of human preferences of the Parti 20B model over the 3B model in terms of P2 categories (*left*) and challenge aspects (*right*). Each aspect is shown along with the number of prompts associated with it (e.g., the *Abstract* category has 46 prompts).

| Parti-350M | Parti-750M | Parti-3B | Parti-20B |

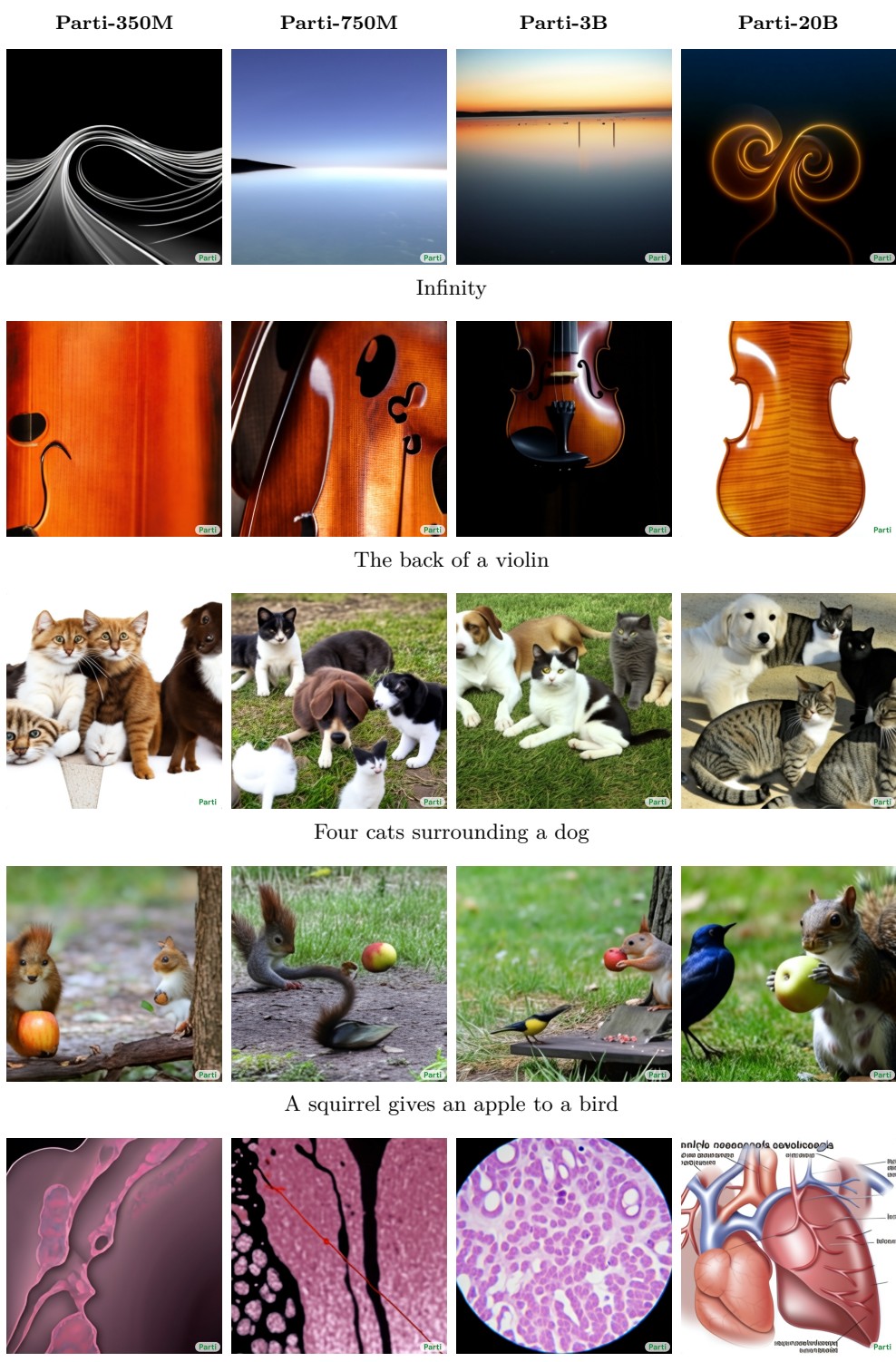

Infinity

The back of a violin

Four cats surrounding a dog

A squirrel gives an apple to a bird

Pneumonoultramicroscopicsilicovolcanoconiosis

Figure 13: Qualitative comparison of scaling Parti models, similar to Figure 10. We show that simple prompts from the P2 benchmark (Section 4.3) can also be quite challenging. These examples test concepts such as ABSTRACT, PERSPECTIVE, QUANTITY, and LINGUISTIC STRUCTURE.

To better understand the improvements of the 20B over the 3B models, Figure 12 further breaks down human preferences of the 20B model in terms of *image-text match* across P2 categories (left) and challenge aspects (right). In terms of categories, the 20B model is clearly preferred over most categories, especially ABSTRACT,

World Knowledge, Vehicles, and Arts. The 20B and 3B models are on par for Produce & Plants. In terms of challenge aspects, the 20B model are better over all dimensions, especially Writing & Symbols, Perspective, and Imagination. See Appendix F for full breakdown on image realism and image-text match for both the Retrieval baseline and the 3B model.

**Qualitative comparison.** To understand qualitatively the effect of scaling, we present in Figure 10 and 13 non-cherry-picked top-1 images sampled from Parti models of increasing sizes (350M, 750M, 3B, 20B). All model variants use the same image tokenizer and CoCa reranking model, described in Section 2.4, with sampling of 16 images per text prompt. We use prompts from the P2 benchmark to test the models' capabilities across a range of categories and challenging aspects.

Figure 10 clearly shows improved quality as we scale up model size. The 3B model is, sometimes, as good as the 20B one in terms of the visual quality and image-text alignment for Fine-grained Detail prompts such as "astronaut riding a horse" with "water lilies". The "blue Porsche 356" over "yellow brick wall" is a strong test of world knowledge: only the 20B model gets it right. The 3B model produces a visually clean car, but it is one which never existed – it seems to merge features of multiple two-seater sports cars from the 1960s. When it comes to more challenging prompts such as those that test World Knowledge and Writing & Symbols, the 20B model is able to produce better compositions, such as the "kangaroo wearing an orange hoodie and blue sunglasses" over the "Sydney Opera House" and the sushi-made "map of United States" (interestingly with wasabi in the map from the 20B model), as well as precise text outputs, such as "Welcome Friends!" and "Very Deep Learning", compared to those of the 3B model.

Figure 13 examines models from a different perspective by demonstrating that short prompts in the P2 can also be quite challenging. The 20B model shows its strong visual abilities when generating Abstract concepts, *e.g.*, the "infinity" sign, and atypical Perspective, *e.g.*, "the back of a violin." While both the 3B and 20B models generate animals rather well, the 20B model shines with more photorealistic outputs, *e.g.*, for the prompt "a squirrel gives an apple to a bird", and correct Quantity in the case of "Four cats surrounding a dog". Lastly, for the Linguistic Structure example of "Pneumonoultramicroscopicsilicovolcanoconiosis", considered to be the longest English word (related to a lung disease), the 20B model generates a reasonable illustration of a lung.

# 6 Discussion

In this section, we discuss our selected examples, then give a walk through of working with a complex prompt, and finally provide a break-down (with examples) of limitations of Parti.

## 6.1 Selected Examples

In Figures 1 and 2 (along with the additional examples in Figures 16, 17 and 19 in the Appendix), we hope to concretely convey some of the strengths of Parti, including its ability to handle complex prompts, multiple visual styles, words-on-image, world knowledge, and more. The top row of Figure 1 shows the model accommodating a very long and complex description of van Gogh's painting The Starry Night—the outputs all come from the same batch and show considerably visual diversity. The other rows show that the model can co-locate famous landmarks in a common scene and adapt styles.

The top row of Figure 2 shows single images for tricky or complex prompts: (A) is short but includes writing of an intentionally misspelled word "toaday" (toad-ay) and an image-within-image specification that is successfully executed; (B) has a complex prompt that requires knowledge of both Anubis and the Los Angeles skyline; and (C) has extensive visual complexity covering multiple entities and their details and (literal) writing on the wall, with photo-realism. (D) shows simple but effective concept combination. (E) provides four outputs from the same batch, showing both diversity and quality of multiple outputs for a complex prompt. (F) demonstrates text rendering of a reasonably long expression along with other complex details, including following the contours of driftwood, fading of the writing and its reflection in the water, and integration of words into stained glass.[6] (G) shows that the model can reproduce world knowledge related to

---

[6]The prompts in Figure 2, panel F are: *The saying "BE EXCELLENT TO EACH OTHER" X*, where X is:

precise visual details for multiple variants of three vehicles that have existed and changed over many decades, while also incorporating additional scene details and color specifications and producing photo-realistic outputs. (H) shows a detailed description in the style of various artists and art movements. (I) shows the adaptation of animals and sports to the style of Egyptian hieroglyphics and additionally placed on Athenian vases (which featured a different art style), including conforming the painting to the contours of the vases.

### 6.2 Growing a Cherry Tree

Like other recent work on text-to-image generation, this paper includes novel images and the complex prompts given to the model to produce them, as discussed in the previous subsection. Naturally, the most challenging and impressive examples are *selected* (that is, *cherry picked*), as noted in the captions. As such, they do not typically represent, for example, a single shot interaction in which the model directly produces such an image as its most highly ranked output. As noted in section 8, we are unable to release our model directly to the public, so in this section we hope to provide a brief window into the process of increasing descriptive and visual complexity with Parti, including how, along the way, things go right or do not just work immediately.

A key concept we would like to introduce in this endeavor is that of *growing the cherry tree* — a concept that we believe will be useful in this space going forward. To put a fine point on it, many of the prompts and resulting images that are seen in Figure 2 and others in the Appendix (noted as "selected") are not only cherry picked: they are the result of exploring and probing the model's capabilities – the product of an interactive process in which a prompter tests a prompt idea, evaluates the outputs holistically, modifies the prompt, and repeats the process. Sometimes, all the outputs are great, and the prompter wants to push the model further. Other times, none of the outputs are ideal, so strategies to change or rephrase the prompt are employed. In this way, one develops a prompt in increments while interacting with the model, in order to produce a cherry tree that offers up some great outputs that can ultimately be plucked. While many–perhaps most–of the prompts an average user would come up with would be quite a bit simpler, this is how we find the breaking points and identify the next opportunities and challenges to focus on. We also expect that designers, artists and other creatives would similarly want to push on these limits; anecdotally, this is how some artists have described working with current text-to-image models that are available to the public.

As a concrete example, consider the process of developing a complex prompt as depicted in Figure 14. This shows a branching and merging process of creating prompt variations, along with two outputs for each prompt. In each box with a prompt, the best of the top eight 20B Parti outputs (as ranked from all outputs) is given on the left and the worst of the top eight is given on the right.

- We start with two core entities, the sloth and the van, in Row 1.

- Row 2 adds specific details for each; overall, the model accommodates this well, but notice that box 2(b) has a sloth with two books and an odd bow tie as its worst outcome of eight.

- Row 3 shows the first major problems: Box 3(a) has two left arms on the sloth and Box 3(b) has a sloth but is completely missing the van mentioned in the prompt.

- Row 4 shows where things start to get particularly tricky: 4(a) is the full combination of the sloth and van with all details, though only by simple concatenation of the respective prompts. The best (left) output manages to get things technically correct, but the van is out of the picture; however, the worst (right) output is a confused mess. 4(b) attempts to improve matters by relating the sloth and van via positioning (*The sloth stands a few feet in front of a shiny VW van*), but this ends up producing cartoonish outputs, and even the best output puts the cityscape in the background rather than on the van. 4(c) shows one fix, which is to switch to flowers on the van; with this, the model can produce a strong, photorealistic output that corresponds well with the prompt.

  - *written on a red brick wall with a graffiti image of a green alien wearing a tuxedo. A yellow fire hydrant is on a sidewalk in the foreground.*
  - *written with carved letters on driftwood.*
  - *written in faded paint on the hull of an old wooden boat and reflected in the water. Wide-angle lens.*
  - *written in a stained glass window.*

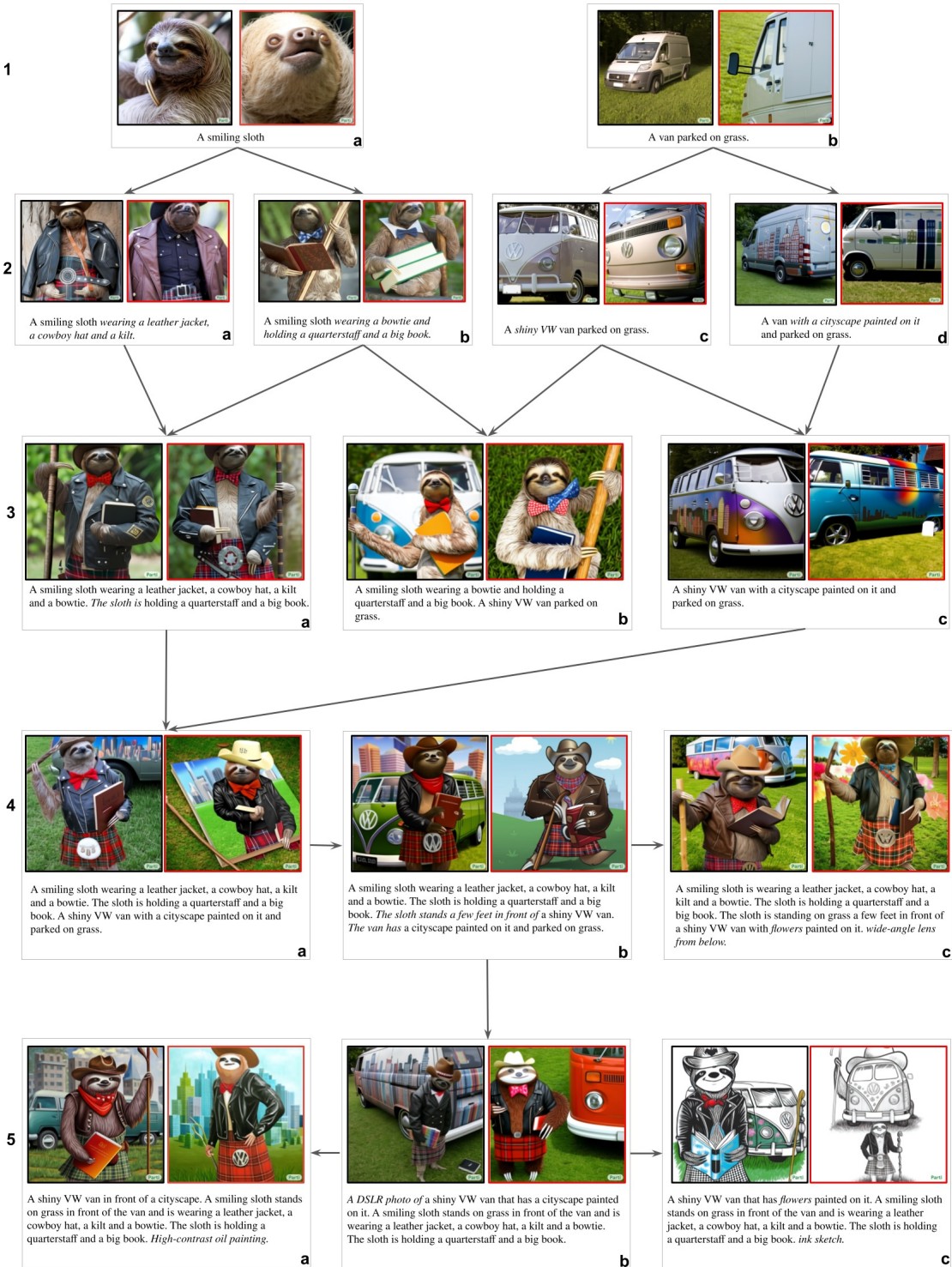

Figure 14: A depiction of several steps in the process of adding details or rephrasing to build a complex prompt that the model responds to well. For each prompt, we show the best output of the top eight ranked images (on the left) and the worst (on the right), using the 20B Parti model. Note that for rows 1-4, the additional text "*dslr photograph. daytime lighting.*" is appended to the given prompt. See Section 6.2 for discussion.

- A second fix is given in 5(b), where the van and photo are promoted to the front of the prompt (rephrasing); alas, the sloth is missing its quarterstaff in the best output, and the worst output is cartoonish and lacking in key details. 5(a) shows that some problems of perspective and representation are resolved by going to a non-photo format (but the city is not on the van, yet again). 5(c) shows that a format change (to *ink sketch*) and swapping flowers for the cityscape again rescue the composition for the best output; however, the worst output is again a confused mess with a van that has a cowboy hat and a sloth arm holding a quarterstaff.

We hope this diagram and its description give a sense of how a model like this responds as one adds detail and rephrases prompts. In a sense, this is a form of *model whispering* as one stretches such models to their limits. That said, it is often remarkable how much descriptive complexity and diversity the model can readily accommodate. In the next section, we point to specific areas in which the Parti model still systematically runs into difficulty and are thus key areas for improvement.

### 6.3 Limitations

There are a number of situations that Parti currently handles poorly or inconsistently, or which lead to interesting patterns in outputs – even producing some bloopers (which can at times be delightful). The likelihood of all of these errors increases with prompt complexity. Figure 15 provides example prompts and images, along with mention of specific failure modes they exemplify. Note that the examples are selected *non-cherries* that are often low in the ranked outputs, and that in many cases (though not all) the model produces a highly ranked and high-quality output for the prompt. [7] We list the failure modes and discuss them here. Unless otherwise specified, all references are to Figure 15, and are given in terms of panel (capital letter) and image (a-d).

**Color bleeding.** When color is provided for one object in a description or is very strongly associated with the object itself, but *left unspecified for others*, it often spreads to the under-specified objects. Examples include baseballs being made yellow when in the presence of tennis balls (A(b,c)), or a crown being given the color of a shirt (D(a,d)).

**Feature blending.** Similarly, when two described objects have some similarities, they can become fused as one object or the attributes of another are incorporated. Examples included baseballs with tennis ball fuzz (A(b,c)), hybrids of the Great Pyramid and Mount Everest instead of co-placement (B(c,d)), and the melding of the VW symbol into the kilt's sporran (Fig. 14, boxes 4b, 4c, 5a).

**Omission, hallucination, or duplication of details.** Especially in complex scenes, the model will at times either omit some mentioned details, duplicate them or hallucinate things that are not mentioned. Examples include missing baseballs in A(d), the missing space shuttle in D(a,c), the missing horse carriage and statue in I(b), and the inclusion (hallucination) of glasses in H(d).

**Displaced positioning or interactions.** Objects are at times put in the wrong position (especially with increased prompt complexity). Examples include the beetle not grappling with the airplane in C(a,c,d), the space shuttle and drawing of a shuttle in D(b,d), the grass and crack in H(a), and the position of the Earth in I(b).

**Counting.** Parti can reliably produce up to seven objects of the same type (when not also specifying other objects as in panel A or mixing other details). Beyond that, it is mostly imprecise. When there are counts of

---

[7]Some prompts could not fit in the figure. They are:

**H(a,b)** *A robot painted as graffiti on a brick wall. The words "Fly an airplane" are written on the wall. A sidewalk is in front of the wall, and grass is growing out of cracks in the concrete.*

**I(a,b)** *Horses pulling a carriage on the moon's surface, with the Statue of Liberty and Great Pyramid in the background. The Planet Earth can be seen in the sky. DSLR photo.*

**I(c)** *A shiny robot wearing a race car suit and black visor stands proudly in front of an F1 race car. The sun is setting on a cityscape in the background. comic book illustration.*

**I(d)** *the saying "BE EXCELLENT TO EACH OTHER" on a rough wall with a graffiti image of a green alien wearing a tuxedo.*

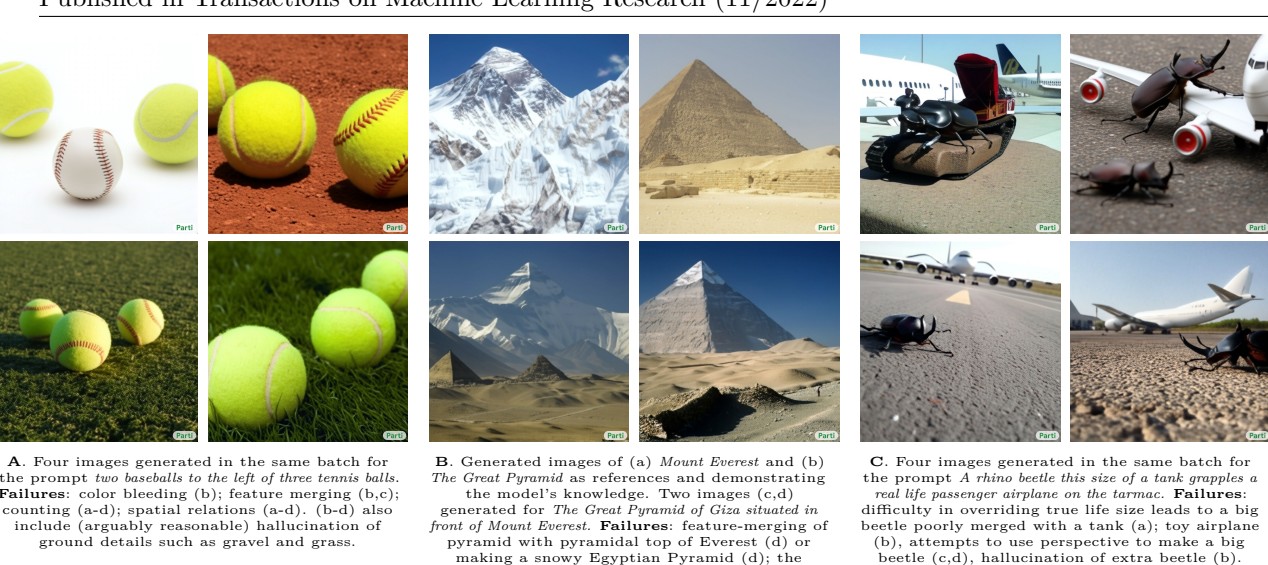

**A**. Four images generated in the same batch for the prompt *two baseballs to the left of three tennis balls.* **Failures**: color bleeding (b); feature merging (b,c); counting (a-d); spatial relations (a-d). (b-d) also include (arguably reasonable) hallucination of ground details such as gravel and grass.

**B**. Generated images of (a) *Mount Everest* and (b) *The Great Pyramid* as references and demonstrating the model's knowledge. Two images (c,d) generated for *The Great Pyramid of Giza situated in front of Mount Everest.* **Failures**: feature-merging of pyramid with pyramidal top of Everest (d) or making a snowy Egyptian Pyramid (d); the pyramid depicted is Khafre's, not Khufu's (Great) Pyramid (d).

**C**. Four images generated in the same batch for the prompt *A rhino beetle this size of a tank grapples a real life passenger airplane on the tarmac.* **Failures**: difficulty in overriding true life size leads to a big beetle poorly merged with a tank (a); toy airplane (b), attempts to use perspective to make a big beetle (c,d), hallucination of extra beetle (b).

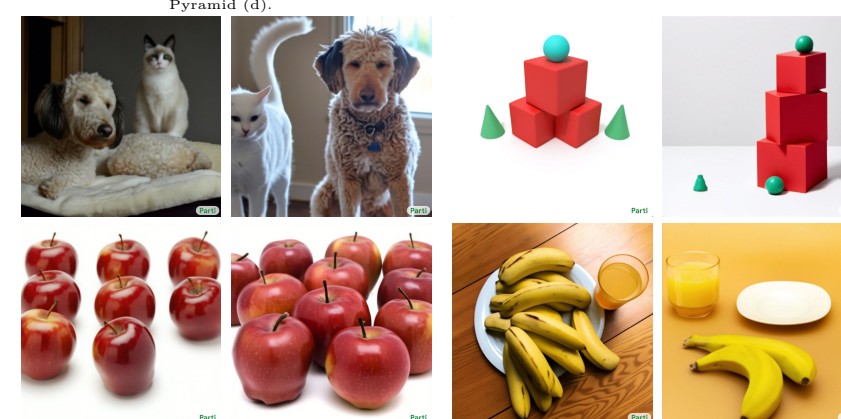

**D**. Four images generated in the same batch for the prompt *A portrait of a statue of Anubis with a crown and wearing a yellow t-shirt that has a space shuttle drawn on it. A white brick wall is in the background.* **Failures**: color bleeding (a,d); incorrect visual aspect (a,b,d); (unspecified) media blending (c); displaced positioning (b,d); missing details (a,c).

**E**. (a,b) Two images generated in the same batch for *a cream colored labradoodle next to a white cat with black-tipped ears.* (c,d) Two images generated in the same batch for *ten red apples.* **Failures**: hard to disentangle specific features assigned to multiple entities in the same description (a,b); incorrect count of 8 (a) and 11 (b). (Note that some correctly had ten apples.)

**F**. (a, b) Two images in the same batch for the prompt *a stack of three red cubes with a blue sphere on the right and two green cones on the left.* (c, d) Two images in the same batch for the prompt *a plate that has no bananas on it. there is a glass without orange juice next to it.* **Failures**: Incorrect relative positioning of objects (a,b,d). Incorrect coloring-to-attribute association (b). Hallucination (of objects specifically mentioned as absent) (c, d).

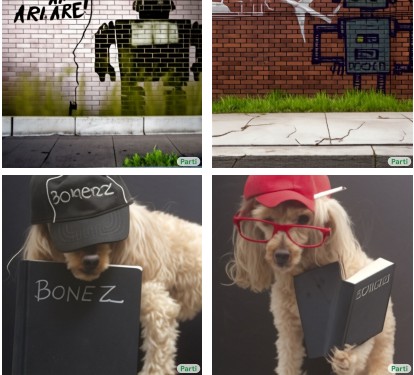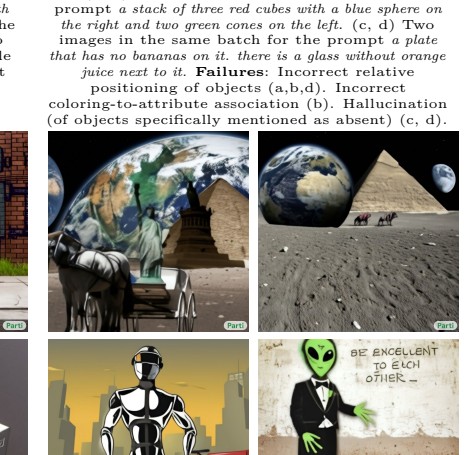

**G**. (a,b) Two images in the same batch for *A horse sitting on an astronaut's shoulders. DSLR photo.* (c,d) Two images in the same batch for *Zoomed out view of a giraffe and a zebra in the middle of a field covered with colorful flowers* **Failures**: hallucination (a); difficulty overriding strong priors (b); entity overriding (another horse instead of an astronaut) (b); entity duplication (zebras) (c,d).

**H**. (a,b) Two images generated in the same batch for *A robot painted as graffiti on a brick wall. The words "Fly an airplane" are written on the wall. A sidewalk is in front of the wall, and grass is growing out of cracks in the concrete.* (c,d) Two images for *a poodle wearing a baseball cap holding a dictionary in hand and writing bonez on a chalkboard* **Failures**: Errors/omissions in rendering text (all); use-mention confusion for words (b); incorrect visual aspect (grass)(a); displaced positioning (cracks) (a); hallucination (d).

**I**. (*See main text for full prompts*) (a,b) Two images generated for a complex prompt of horses pulling a carriage, the statue of liberty, the moon, and the Earth, and images for (c) a robot race car driver and (d) alien graffiti with writing. **Failures**: Duplication of objects (a,b); relative scaling errors (a,b,c); physically impossible configurations (b,c,d); (unspecified) media blending (d).

Figure 15: Images from 20B Parti model showing errors and limitations. In the captions, (a), (b), (c), (d) refer to top left, top right, bottom left and bottom right, respectively. Note that many of the example images come from batches that included successful (and typically more highly ranked) images. See Section 6.3 for detailed discussion.

multiple types of entities, there is a near complete failure, as shown in A(a-d). Interestingly, the reranker appears to help with counting; *e.g.*for *five red apples* the top six images all have five apples, but thereafter the count varies from four to six. For *ten red apples* (see E(c,d)), the counts for the top-ranked eight images for one batch were *8, 10, 9, 9, 9, 8, 11, 6.*

**Spatial relations.** While the model often correctly depicts objects specified as above or below each other, it is still inconsistent for that and is usually random for *left vs. right.* These failures especially compound when it involves spatial relations between groups of objects (e.g., A(a) and F(a,b)).

**Negation and absence.** Parti tends to draw items that are mentioned, even when the prompt says a thing is missing. For example, F(c,d) shows outputs that include bananas and orange juice even though the prompt is *A plate that has no bananas on it. There is a glass without orange juice next to it.* F(d) has bananas off the plate, but this is just an incidentally interesting example and not an indicator of handling mention-of-absence well. We do find some examples with no bananas that are ranked very low, so there is also appears to be a compounding effect of both the generator and reranker in this case.

**Incorrect visual aspect and media blending.** Especially in scenes where mixed media types are involved, such as photo-realistic objects along with writing and paintings on walls, some items will jump from being depicted as an object to being depicted as a drawing, or vice versa. Examples include the drawn form of Anubis in D(a), the space shuttle as an object in D(b), and the grass as painted on the wall in H(a). We also see media blending, where an object's coherence is lost and it transitions from object to drawing, such as the blending of the drawn Anubis head with body in D(c) and the alien's top part being a painting that connects to non-drawn legs in I(d). While these are interesting visual effects, they were not specified in the prompt and thus indicate a lack of control over object coherence and appearance in such situations.

**Strong visual priors.** Certain configurations and visual features are so strongly correlated that it can be hard to nudge the model away from them, especially in the face of other complexities in the description. For example, Panel C shows a failed attempt to create a tank-sized rhinoceros beetle—for the most part, the model shrinks the scene (e.g. with a toy plane) or tries to use perspective to make the beetle appear visually larger. Its only successful attempt to make a massive beetle errs in including an actual tank. We also found that the model is very resistant to reverse a horse-and-rider situation: the only way to have the horse on the astronaut was to avoid *riding* and instead rephrase this as something akin to *a horse sitting on an astronaut's shoulders*; this produces some examples that give the requested configuration (G(a)), but most outputs contain errors, including the astronaut on the horse, the astronaut next to the horse, or even depicting the rider as a horse astronaut (G(b)). In some cases where a horse was put onto an astronaut, the model included yet another astronaut on the horse. As another example, when generating images of statues of Abraham Lincoln wearing clothing such as a polo shirt and a baseball cap, the model can do it reliably, but many of the outputs nevertheless have him wearing outfits from well-known statues and paintings of Lincoln.

**Strong linguistic priors.** Certain terms are highly associated with particular entities or word senses. For example, *a soccer ball flying over a Lincoln* shows a ball flying over a statue of Abraham Lincoln. The automotive Lincoln can be brought in by adding *car*, *SUV*, etc. Similarly, *a soccer ball flying over a bat* produces images of baseball bats; again, this can be fixed, as it were, by adding attributes of animal bats. It is even possible to explore the tensions between competing word senses in examples such as *the astronomer married a star*, where the presence of astronomer invokes the heavenly body sense of *star* but the act of marrying invokes (more plausibly) the sense of a movie star (Erk & Herbelot, 2021). When given *an illustration of an astronomer marrying a star*, Parti goes with the heavenly body depiction, even when making it *movie star.* Switching to a politician instead of astronomer leads to outputs showing a man and woman, but also including a star (iconic five-point representation) in the image. Even with more details such as *A photo of an astronomer in a tuxedo marrying a beautiful movie star. The star is wearing a white dress.*, the model produces some outputs with a wedding couple, but backed by an image of the cosmos. It seems likely that the overwhelming presence of stars as heavenly bodies or icons in the visual training data creates a very strong combined linguistic and visual prior that is hard for the model to shift from.

**Text rendering errors.** It is remarkable that text-to-image models like Parti and Imagen can render text in diverse and contextually appropriate ways on images (see the examples in Figure 2, Panel F, for example), even though there is no explicitly curated or created training data for learning this ability. Nevertheless, text

rendering is often hit-or-miss. It is common to have a few characters off even in simple prompts. For more complex prompts, the errors in rendering text increase with scene and descriptive complexity, as shown in Panel H. The ability to render text can also lead to situations where the model basically tries to render the entire prompt text in the image, e.g. as the title of a book. This usually happens for prompts that are not visually descriptive (e.g. *How to succeed in life.*), and it is necessary to explicitly call out formats like *oil painting* or *illustration* in order for these to produce a non-textual output.

**Use-mention errors.** The model can render text on images, but at times it produces an image (use) rather than text (mention) (e.g., the drawing of an airplane in H(b)), or vice versa (rendering *Space shuttle* on a t-shirt instead of drawing one – another output related to D, but not shown).

**Disentangling multiple entities.** The model is often able to pack quite a lot of detail into an image that contains a single entity, but is much more challenged when there are multiple key entities. This can be seen with the sloth and van example of Figure 14, where the model struggles at their combination (row 3, box b), and where subsequent addition of complexity leads to fewer good outputs. However, it is often even more difficult when the entities are of the same type, such as two animals, as demonstrated in E(a,b), where even fairly simple details are spread between the entities.

**Stylistic misses.** Parti can produce many styles reliably, such as pointillism and woodcut, but others like cubism and surrealism often miss the style at a deeper level, especially when applied to a complex scene. The styles of some specific painters can be modeled well, such as van Gogh and Rembrandt, but others are either lacking or very hit-or-miss, such as Michelangelo (which mostly look the same as adding *oil painting.*). Interestingly, scale interacts with specific painters: for example, applying *in the style of van Gogh* actually produces more diverse outputs consistent of style for the 3B model, while outputs are pretty much dominated by Starry Night with the 20B one.

**Impossible scenes.** Some outputs (usually ones ranked quite low) show entities that are inconsistently integrated, such as I(c), where the robot straddles the car in a nonsensical manner. Other cases involving mixed media lead to similarly bizarre outputs, including a drawing of Anubis's head on a photo-like body (D(c)) and a graffiti alien that extends to a depiction of real feet on the ground (I(d)). (Note that in both of these cases, these would be great outputs *if* the prompt explicitly specified that these effects were desired.) Other challenges come up in trying to compose complex fantastical scenes, such as the carriage, moon and earth outputs in I(a,b), where the lighting of the statue, moon and Earth are completely inconsistent (a) and the Earth appears sitting on the moon (b).

**Zoom and perspective.** Parti often produces outputs that are too zoomed in, e.g. showing just part of a vehicle or a subject. While it can respond to directives like *zoomed out*, *three-quarters view*, *wide-angle lens*, this still often results in cropping on the subject. To ensure broader view, it is often necessary instead to use other details, such as including shoes on a subject to get the feet or adding a description of a field and flowers to get an entire giraffe (as done in G(c,d)).

**Animal protagonists.** Due to concerns discussed in Section 8, we experimented with many animals acting as stand-ins for protagonists in descriptions. As it turns out, some animals are easier to work with than others. For example, mammals with paws that are more similar to human hands, such as bears, wombats, and raccoons more reliably result in better images than geckos, insects, and fish. This is likely due to the visual and morphological similarity they have with people as well as the fact that they are more commonly used as human-like protagonists in the wider visual world (*e.g.*, in cartoons). It is also often necessary to include "photo" or "photograph" in the prompt to push the model to make photo-like images, and even then it will often produce cartoon-like outputs—especially with increasing description complexity.

**Detailed or tricky visual effects.** It is very difficult to get the model to respond to prompts such as *a bear emerging from a puzzle* (and variations thereof) where one might want to have an Escher-like image of part of the bear appearing as quasi-real and the other being part of the puzzle. In general, controlling such fine-grained specifications seems beyond the current model, and is likely to be better served in an interactive editing setting.

**Common misconceptions.** Some visual world knowledge is incorrectly understood in the broader world, and this is partially reflected in the data and then the model. For example, it is commonly thought that the

central pyramid in the Giza Complex is the Great Pyramid, but in fact that is the pyramid of Khafre. The true Great Pyramid is Khufu's, which is to the north. Khafre's pyramid is often mistakenly attributed as the *great one* because it sits on slightly higher ground (but is actually shorter at 136.4 meters compared to Khufu's 146.6 meters) and because it sits prominently behind the Great Sphinx. This mistaken attribution is reflected in images associated with the terms *the Great Pyramid*, and it shows up in many of Parti's outputs for the Great Pyramid (including B(d)).

We hope these observations and breakdown of limitations and error types, and their correspondences in many of the PartiPrompts, will be useful both for contextualizing the strong capabilities we present elsewhere in the paper and for inspiring future work on improving text-to-image generation models in general. In this light, it is also worth recalling WordsEye (Coyne & Sproat, 2001), an automatic text-to-scene system built in 2001. It derived dependency structures from prompts, converted them to semantic structures, and then used those to select, position and scale the objects and participants in the described scene. Though it is no match for current text-to-image such as Parti on open-domain, broad capabilities, including world knowledge, it actually could *precisely* manage several of the above stated limitations, including counting, negation, relative scale, and positioning – such that it could produce a computer graphics visual (based on an explicit 3D representation) corresponding to complex prompts such as:

> John uses the crossbow. He rides the horse by the store.The store is under the large willow. The small allosaurus is in front of the horse. The dinosaur faces John. A gigantic teacup is in front of the store. The dinosaur is in front of the horse. The gigantic mushroom is in the teacup. The castle is to the right of the store.

WordsEye could also handle specifications of measurements such as lengths and heights, and make correct visual adjustments for the output image, *e.g.*, for prompts such as *The lawn mower is 5 feet tall. John pushes the lawn mower. The cat is 5 feet behind John. The cat is 10 feet tall.* With the advent of broadly capable – but often imprecise models – such as Dall-E 2, Imagen and Parti, this should inspire us to revisit ideas and capabilities of earlier systems such as WordsEye and aspire toward models that combine breadth, visual quality, and control.

## 7    Related Work

**Text-to-image generation.** The task of text-to-image generation tackles the problem of synthesizing realistic images from natural language descriptions. Successful models enable many creative applications. WordsEye (Coyne & Sproat, 2001) was a pioneering approach that was based on rule-based methods and explicit 3D representations. One of the earliest models based on deep learning (Reed et al., 2016a) proposed using conditional GANs for generating images of birds and flowers from language descriptions. Later works improved upon generation quality by introducing progressive refinement (Zhang et al., 2017; 2018a) and using cross-modal attention mechanisms (Xu et al., 2018; Zhang et al., 2021). Several other works propose using hierarchical models that generate images by explicitly modeling the location and semantics of objects (Reed et al., 2016b; Hong et al., 2018; Hinz et al., 2019; Koh et al., 2021).

Remarkable improvement has been achieved by treating text-to-image generation as a sequence modeling problem (Esser et al., 2021; Ding et al., 2021; Ramesh et al., 2021; Gafni et al., 2022; Dayma et al., 2021) trained on large-scale image-text pairs. A two-stage framework is usually exploited (Yu et al., 2022a; Chang et al., 2022) where in the first stage the images are tokenized into discrete latent variables. With image tokenization and de-tokenization, text-to-image generation is treated as a sequence-to-sequence problem amenable to language models with transformers, which provide opportunities of scaling such models by applying techniques and observations from large language models (Radford et al., 2018; Du et al., 2022; Chowdhery et al., 2022b; Hoffmann et al., 2022).

The most recent impressive results have been attained with diffusion models (Nichol et al., 2022; Ramesh et al., 2022; Saharia et al., 2022), where the models are learned to condition on the text encoder of the CLIP (Radford et al., 2021) image-text model, or frozen text encoder (like T5 (Raffel et al., 2020)) pretrained by language self-supervision. Diffusion models work by positing a process of iteratively adding noise to an

image and then learning to reverse that noise conditioned on text input or feature. When used with diffusion model cascading (Ho et al., 2022), these models have proven effective for generating high-fidelity images from text prompts and have achieved state-of-the-art zero-shot MS-COCO FID scores (Saharia et al., 2022).

**Image tokenizers.** Previous work has explored tokenizing images into discrete latent variables with a learned deep neural network. Early work like discrete Variational Auto-Encoders (dVAEs) (Rolfe, 2017) optimizes a probabilistic model with discrete latent variables to capture datasets composed of discrete classes. However, dVAEs often generate blurry pixels when applied to natural images. Recent work like VQGAN (Esser et al., 2021) (based on VQVAE (Van Den Oord et al., 2017)) further applies adversarial loss (Karras et al., 2020) and perceptual loss (Johnson et al., 2016; Zhang et al., 2018b) to synthesize images using convolutional neural networks with self-attention modules. ViT-VQGAN (Yu et al., 2022a) builds upon VQGAN, with improvements on both architecture and codebook learning. Transformers (Vaswani et al., 2017) are used to encode images into latent variables and decode them back to images. We use ViT-VQGAN (Yu et al., 2022a) with slight modifications (see Section 2.1) as our image tokenizer.

## 8 Broader Impacts

Beyond the model capabilities and evaluations presented above, there are broader issues to consider with large-scale models for text-to-image generation. Some of these issues pertain to the development process itself, including the use of large, mostly uncurated training datasets of images obtained from the web with little oversight (discussed also in (Saharia et al., 2022)), or conceptual vagueness around constructs in the task formulation (Hutchinson et al., 2022). Since large text-to-image models are foundation models (Bommasani et al., 2021) — enabling both a range of system applications as well as finetuning for specific image generation tasks — they act as a form of infrastructure which shapes our conceptions of what is both possible and desirable (Hutchinson et al., 2021; Denton et al., 2021). Predicting all possible uses and consequences of infrastructure is difficult if not impossible, and so responsible AI practices which emphasize transparently documenting and sharing information about datasets and models are crucial (Mitchell et al., 2019; Gebru et al., 2021; Pushkarna et al., 2022). Although applications are beyond scope of this paper, we discuss here some likely opportunities and risks that can be anticipated.

**Creativity and art.** The ability of machine learned models to produce novel, high-quality images using language descriptions opens up many new possibilities for people to create unique and aesthetically appealing images, including artistic ones. Like a paint brush, these models are a kind of tool that on their own do not produce art—instead people use these tools to develop concepts and push their creative vision forward. For artists, such models could provide new means of innovation and exploration, including opportunities to create on-the-fly generation of art that sets a theme or style while responding to viewer interactions, or to generate novel and unique visual interactions in video game environments. For non-artists, these affordances present a chance to explore their visual creativity through a natural language interface which does not require technical artistic ability. Text-to-image systems could also assist creativity for people with disabilities (cf. (El-Nouby et al., 2019; Sharma et al., 2018)), but we caution against doing so without also adopting participatory methods to increase the likelihood of actual needs being met and to avoid misconceptions about disability (Mankoff et al., 2010).

Assessing the design merit or artistic merit (or lack thereof) of a piece created using machine learned models requires a nuanced understanding of algorithmically based art over the years, the model itself, the people involved and the broader artistic milieu (Browne, 2022). The range of artistic outputs from a model is dependent on the training data, which may have cultural biases towards Western imagery, and which may prevent models from exhibiting radically new artistic styles the way human artists can (Srinivasan & Uchino, 2021).

**Visual (mis)communication.** The pre-ML history of text-to-image largely consists of assisting communication with non-literate groups including language learners (including children, *e.g.*, storybook illustrations), low-literacy social groups (*e.g.*, up until the late modern period, religious illustrations for low-literacy congregations), and speakers of other languages. Parti uses an architecture and strategy that is directly connected to the neural sequence-to-sequence models used for machine translation (Wu et al., 2016) and other communication aids such as sentence simplification (Alva-Manchego et al., 2020) and paraphrasing (Zhou & Bhat, 2021).

This potentially strengthens the temptation to use large text-to-image models to assist with communication. However, we caution against the use of text-to-image models as communication aids, including for education (cf. (El-Nouby et al., 2019)), until further research has examined questions of efficacy and utility, since text and image convey meaning in distinct ways and with distinct limitations. Cross-cultural considerations are of special concern, as little research has considered questions of accessibility of computer-generated images to members of non-Western cultures. Not only do visual styles differ cross-culturally, but also the form and appearance of instances of categories may radically differ across cultures (*e.g.*, wedding attire, food, etc (Shankar et al., 2017; De Vries et al., 2019)) in ways that might lead to miscommunication.

**Deepfakes and disinformation.** Given that the quality of model outputs is good enough to be confused for real photographs,[8] and also because output quality and realism is rapidly improving, there are obvious concerns around using such technology to create deepfakes. One way to mitigate this problem is to apply watermarks that people cannot perceive to every generated image (Luo et al., 2020), such that it is possible to verify whether any given image is generated by a particular model such as Parti. While this approach may mitigate risks of disinformation, harms may still occur when an individual's likeness is reproduced without their consent.

**Bias and safety.** Text-to-image generation models like GLIDE, DALL-E 2, Imagen, Make-a-Scene, CogView and Parti are all trained on large, often noisy, image-text datasets that are known to contain biases regarding people of different backgrounds. This is particularly highlighted in Birhane et al's (Birhane et al., 2021) analysis of the LAION-400M dataset (Schuhmann et al., 2021): their study of the dataset surfaced many problems with respect to stereotyping, pornography, violence and more. Other biases include stereotypical representations of people described as lawyers, flight attendants, homemakers, and so on. Models trained on such data without mitigation strategies thus risk reflecting and scaling up the underlying problems. Our primary training data is selected and highly filtered to minimize the presence of NSFW content; however, we incorporated LAION-400M during finetuning with classifier-free guidance – this improved model performance but also led to generation of NSFW images in some contexts. Other biases include those introduced by the use of examples that primarily have English texts and may be biased to certain areas of the world. In informal testing, we have noticed, for example, that prompts mentioning wedding clothes seem to produce images biased towards stereotypically female and Western attire.

**Intended uses.** Due to the impacts and limitations described above, and the need for further exploration of concerns, Parti is a research prototype. It is not intended for use in high-risk or sensitive domains, and is not intended to be used for generating images of people.

These considerations all contribute to our decision not to release our models, code or data at this time. Instead, we will focus in follow-on work on further, careful measurement of model biases, along with mitigation strategies such as prompt filtering, output filtering and model recalibration. We also believe that it may be possible to use text-to-image generation models as tools to understand biases in large image-text datasets at scale, by explicitly probing them for a suite of known types of bias and also trying to uncover other forms of hidden bias. We will also coordinate with artists to adapt capabilities of high performing text-to-image generation models toward their work, be it for purely creative ends or art-for-hire. This is all the more important given the intense interest among many research groups and the consequent fast pace of development of models and data for training them. Ideally, these models will augment—rather than replace—human creativity and productivity, such that we all can enjoy a world filled with new, varied and responsible aesthetic visual experiences.

## 9   Conclusion

In this work, we demonstrate that autoregressive models like Parti can produce diverse, high-quality images from textual prompts, and furthermore that they present distinct scaling advantages. In particular, Parti is able to represent a broad range of visual world knowledge, such as landmarks, specific years, makes and models of vehicles, pottery types, visual styles – and integrate these into novel settings and configurations. We also provide an extensive discussion of the limitations, including a breakdown of many kinds of model

---

[8]E.g., DALL-E 2 outputs: `https://www.mattbell.us/my-fake-dall-e-2-vacation-photos-passed-the-turing-test/`

errors and challenges, that we hope will be useful both for contextualizing what the model can do and for highlighting opportunities for future research. To this end, the PartiPrompts (P2) benchmark that we release with this work are intentionally crafted to induce many of these error types.

There are also opportunities to integrate scaled autoregressive models with diffusion models, starting with having an autoregressive model generate an initial low-resolution image and then iteratively refining and super-resolving images with diffusion modules (Gu et al., 2022; Ramesh et al., 2022; Saharia et al., 2022). It is also crucial to make progress on the many significant evaluations and Responsible AI needs for text-to-image generation models. To this end, we will conduct more experiments and comparisons with both autoregressive and diffusion models in order to understand their relative capabilities, to address key questions of fairness and bias in both classes of models and strategies for mitigating them, and to identify optimal opportunities for combining their strengths.

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

## A    More Selected Examples

In Figure 16, Figure 17, Figure 18 and Figure 19, we present more selected examples probing the model's capabilities to handle complex prompts, multiple visual styles, texts-on-image, world knowledge, and more.

## B    Image Samples for Cross Reference and Comparison

This section presents image samples with the same text prompts from related work including DALL-E (Ramesh et al., 2021) (Figure 20), GLIDE (Nichol et al., 2022) (Figure 21), unCLIP (Ramesh et al., 2022) (Figure 22) and Imagen (Saharia et al., 2022) (Figure 23) for cross reference and comparison. We exclude some text prompts or replace some sub-prompts with broader impact concerns that are discussed in Section 8. We also add some new (sub-)prompts (noted in the caption) that are not from the related work to make four images per row for typography.

## C    Qualitative comparison on MS-COCO

Figure 24 shows a qualitative comparison of non-cherry-picked Parti sampled images along with outputs of other approaches (Ramesh et al., 2021; Nichol et al., 2022; Gafni et al., 2022; Ramesh et al., 2022) on MS-COCO prompts. Parti demonstrates strong generalization without fine-tuning on specific domains like MS-COCO, and it achieves a high degree of image realism that is typically very close to that of real images.

## D    PartiPrompts

We list below the full set of categories (Table 7) and challenge aspects (Table 8) together with their descriptions and additional examples in the P2.

## E    Human Evaluation Procedure

This section describes the human evaluation procedure we conducted for evaluating model outputs. The website used for conducting these evaluations are shown in Figure 25. For each image, we request five independent annotators to select which of two (anonymized) models are preferred, for image realism and image-text match. Annotators were employed as contractors and were paid hourly wages that are competitive for their locale. They have standard rights as contractors. They are fluent non-native English speakers.

## F    Human Evaluations on PartiPrompts

### F.1    Effect of Model Scale

Figure 26 further breaks down the human preferences of the 20B model (over the 3B model) across P2 categories (left) and challenge aspects (right) for image realism. We observe that as compared to the human preference for image-text match (Figure 12), the 20B model wins out on image realism by a smaller margin. This suggests that model scaling mostly benefits language understanding and image-text match, as compared to improvements on image realism. This makes intuitive sense: image realism is largely dependent on the quality of the ViT-VQGAN quantizer, and thus scaling the transformer model would primarily improve the ability of Parti to composite and represent natural language concepts.

In terms of individual categories, the 20B model is preferred over the 3B for the majority of categories, most prominently ABSTRACT, WORLD KNOWLEDGE, VEHICLES, and ARTS. For other categories, the human preference is mostly on par for the 20B and 3B models, with the 3B model being slightly preferred for outdoor scenes. Interestingly, we observe that over the challenge aspects, the 20B model outperforms the 3B model on all challenge aspects, *except* for BASIC prompts. This suggests that the 3B model scale is sufficient to handle the basic prompts in PartiPrompts(which generally contain more simple prompts). Increasing model size to 20B does not help and may even slightly harm generation quality on these concepts. In contrast, the

| Category | Description | Additional Examples |
|---|---|---|
| ABSTRACT | *Descriptions that represent abstract concepts, including single words and simple numbers.* | *inspiration; derision; infinity; 42; 0; fairness; energy; gravity; intelligence; yin-yang; meaning of life; A city in 4-dimensional space-time* |
| ANIMALS | *Descriptions in which the primary participants are animals.* | *a Stegasaurus; brain coral; A donkey is playing tug-of-war against an octopus. The donkey holds the rope in its mouth. A cat is jumping over the rope; Dogs sitting around a poker table with beer bottles and chips. Their hands are holding cards.* |
| ARTIFACTS | *Descriptions of a usually simple object (such as a tool or ornament) showing human workmanship or modification as distinguished from a natural object* | *a violin; a t-shirt with Carpe Diem written on it; a doorknocker shaped like a lion's head; a lavender backpack with a triceratops stuffed animal head on top; a paranoid android freaking out and jumping into the air because it is surrounded by colorful Easter eggs* |
| ARTS | *Descriptions of existing paintings or intended to produce novel images in the format of a painting.* | *an abstract painting with blue, red and black; a super math wizard cat, richly textured oil painting; a sport car melting into a clock, surrealist painting in the style of Salvador Dali; Painting of a panic-stricken creature, simultaneously corpselike and reminiscent of a sperm or fetus, whose contours are echoed in the swirling lines of the blood-red sky* |
| FOOD & BEVERAGE | *Descriptions of things animals, especially human beings, eat or drink.* | *a margarita; milk pouring from a glass into a bowl; A bowl of Pho served with bean sprouts on top; a bottle of beer next to an ashtray with a half-smoked cigarrette; A photo of a hamburger fighting a hot dog in a boxing ring. The hot dog is tired and up against the ropes.* |
| ILLUSTRATIONS | *Descriptions of images that involve specific types of graphical representations, including geometrical objects, diagrams, and symbols.* | *a metallic blue sphere to the left of a yellow box made of felt; the cover of a book called 'Backpropaganda' by I.C. Gradients; A set of 2x2 emoji icons with happy, angry, surprised and sobbing faces. The emoji icons look like dogs. All of the dogs are wearing blue turtlenecks.* |
| INDOOR SCENES | *Descriptions about objects and participants that occur indoors.* | *a very fancy French restaurant; a room with two chairs and a painting of the Statue of Liberty; a small kitchen with a white goat in it; A single beam of light enter the room from the ceiling. The beam of light is illuminating an easel. On the easel there is a Rembrandt painting of a raccoon* |
| OUTDOOR SCENES | *Descriptions about objects and participants that occur outdoors.* | *a marina; a white bird in front of a dinosaur standing by some trees; a robot painted as graffiti on a brick wall. a sidewalk is in front of the wall, and grass is growing out of cracks in the concrete.* |
| PEOPLE | *Descriptions where the primary participants are human beings (but not specific individuals, living or dead).* | *a scientist; a family of four walking at the beach with waves covering their feet; Renaissance portrayals of the Virgin Mary, seated in a loggia. Behind her is a hazy and seemingly isolated landscape imagined by the artist and painted using sfumato.* |
| PRODUCE & PLANTS | *Descriptions focused on plants or their products (fruits, vegetables, seeds, etc).* | *lily pads; a banana without its peel; a pineapple surfing on a wave; a flower with large red petals growing on the moon's surface* |
| VEHICLES | *Descriptions where the focus is on man-made devices for transportion.* | *a rowboat; a friendly car; A sunken ship becomes the homeland of fish; a yellow dump truck filled with soccer balls driving in a coral reef. a blue whale looms in the background.* |
| WORLD KNOWLEDGE | *Descritpions focused on objects and places that exist in the real world.* | *A Big Ben clock towering over the city of London; the Sydney Opera House with the Eiffel tower sitting on the right, and Mount Everest rising above; A portrait of a metal statue of a pharaoh wearing steampunk glasses and a leather jacket over a white t-shirt that has a drawing of a space shuttle on it.* |

Table 7: Full descriptions of all *categories* in the PartiPrompts (P2) benchmark. We show additional categories and examples that were not included in Table 3.

| Challenge | Description | Additional Examples |
|---|---|---|
| BASIC | *Descriptions about a single subject or concept with little to no detail or embellishment.* | *a tornado; a hot air balloon; 300; The Starry Night; yin-yang; artificial intelligence* |
| COMPLEX | *Descriptions that include many fine-grained, interacting details or relationships between multiple participants.* | *a hot air balloon with a yin-yang symbol, with the moon visible in the daytime sky; A high-contrast photo of a panda riding a horse. The panda is wearing a wizard hat and is reading a book. The horse is standing on a street against a gray concrete wall. Colorful flowers and the word "PEACE" are painted on the wall. Green grass grows from cracks in the street.* |
| FINE-GRAINED DETAIL | *Descriptions that include very detailed specifications of attributes or actions of entities or objects in a scene.* | *a Triceratops charging down a hill; The Statue of Liberty on a cloudy day; a bamboo ladder propped up against an oak tree; beautiful fireworks in the sky with red, white and blue; a full moon peeking through clouds at night; a photo of a white bird in front of a dinosaur standing by some trees; an elder politician giving a campaign speech* |
| IMAGINATION | *Descriptions that include participants or interactions that are not, or are generally unlikely to be, found in the modern day world.* | *a smiling banana wearing a bandana; a grand piano next to the net of a tennis court; the statue of Liberty next to the Washington Monument; A photo of a light bulb in outer space traveling the galaxy with a sailing boat inside the light bulb* |
| LINGUISTIC STRUCTURES | *Long and/or abstract words or complex syntactic structures or semantic ambiguities.* | *supercalifragilisticexpialidocious; a laptop with no letters on its keyboard; To a squirrel, a dog gives an apple; A robot gives a wombat an orange and a lemur a banana; The trophy doesn't fit into the brown suitcase because it's too large; a brown mouse laughing at a gray cat because a 16 ton weight is about to fall on its head* |
| PERSPECTIVE | *Descriptions that specify particular viewpoints or positioning of the subjects in a scene.* | *Mars rises on the horizon; The frog found itself in the newspaper; view of a giraffe and a zebra in the middle of a field; three quarters view of a rusty old red pickup truck with white doors and a smashed windshield* |
| PROPERTIES & POSITIONING | *Descriptions that target precise assignment of properties to entities or objects (often in the context of multiple entities or objects), and/or the relative spatial arrangement of entities and objects with respect to one another or landmarks in the scene.* | *A green heart with shadow; a large yellow sphere behind a small purple pyramid; a brown trash bin to the left of a blue recyling bin; concentric squares fading from yellow on the outside to deep orange on the inside; a pen-and-ink crosshatched drawing of a sphere with dark square on it* |
| QUANTITY | *Decriptions that specify particular counts of occurences of subjects in a scene.* | *ten red apples; a pile of cash on a wooden table; a group of not more than five meerkats standing with the sun setting behind them* |
| SIMPLE DETAIL | *Descriptions that include only simple or high-level details.* | *a horse in a field of flowers; a soccer ball flying over a car; a subway train coming out of a tunnel; a pumpkin with a candle in it; an Egyptian statue in the desert* |
| STYLE & FORMAT | *Descriptions that specifically focus on the visual manner in which a subject or scene must be depicted.* | *a thumbnail image of a gingerbread man; a horse in a field in Minecraft style; the flag of the United Kingdom painted in rusty corrugated iron; Anime illustration of the Great Pyramid sitting next to the Parthenon under a blue night sky of roiling energy, exploding yellow stars, and chromatic blue swirls* |
| WRITING & SYMBOLS | *Descriptions that require words or symbols to be accurately represented in the context of the visual scene.* | *A green sign that says "Very Deep Learning" and is at the edge of the Grand Canyon; Portrait of a tiger wearing a train conductor's hat and holding a skateboard that has a yin-yang symbol on it. charcoal sketch; Two cups of coffee, one with latte art of a lovely princess. The other has latter art of a frog.* |

Table 8: Full descriptions of all *challenge aspects* in the P2. We show additional challenge aspects and examples that were not included in Table 4.

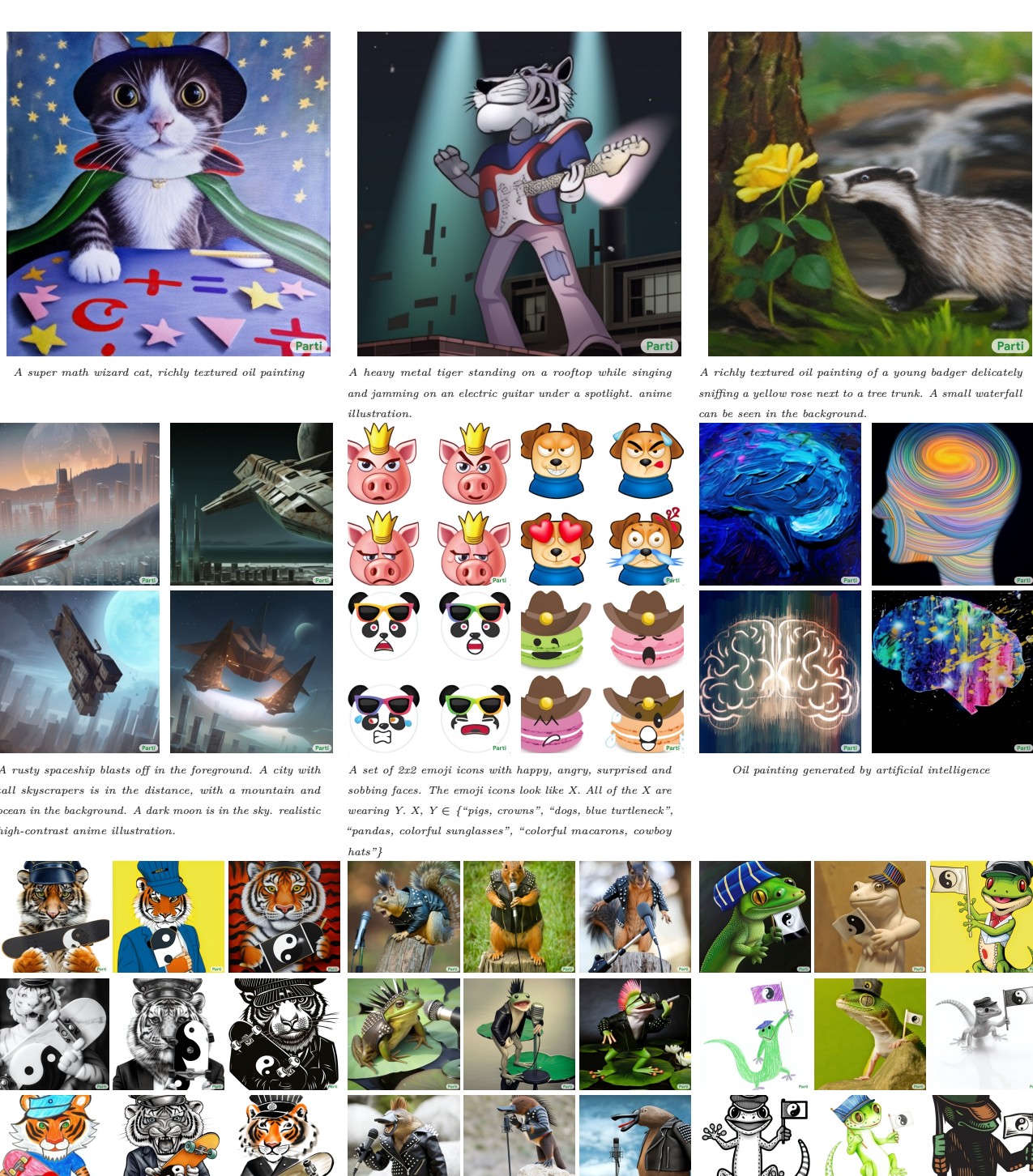

Figure 16: Selected Parti images.

20B model achieves a marked improvement in performance on more difficult PartiPromptsprompts, such as

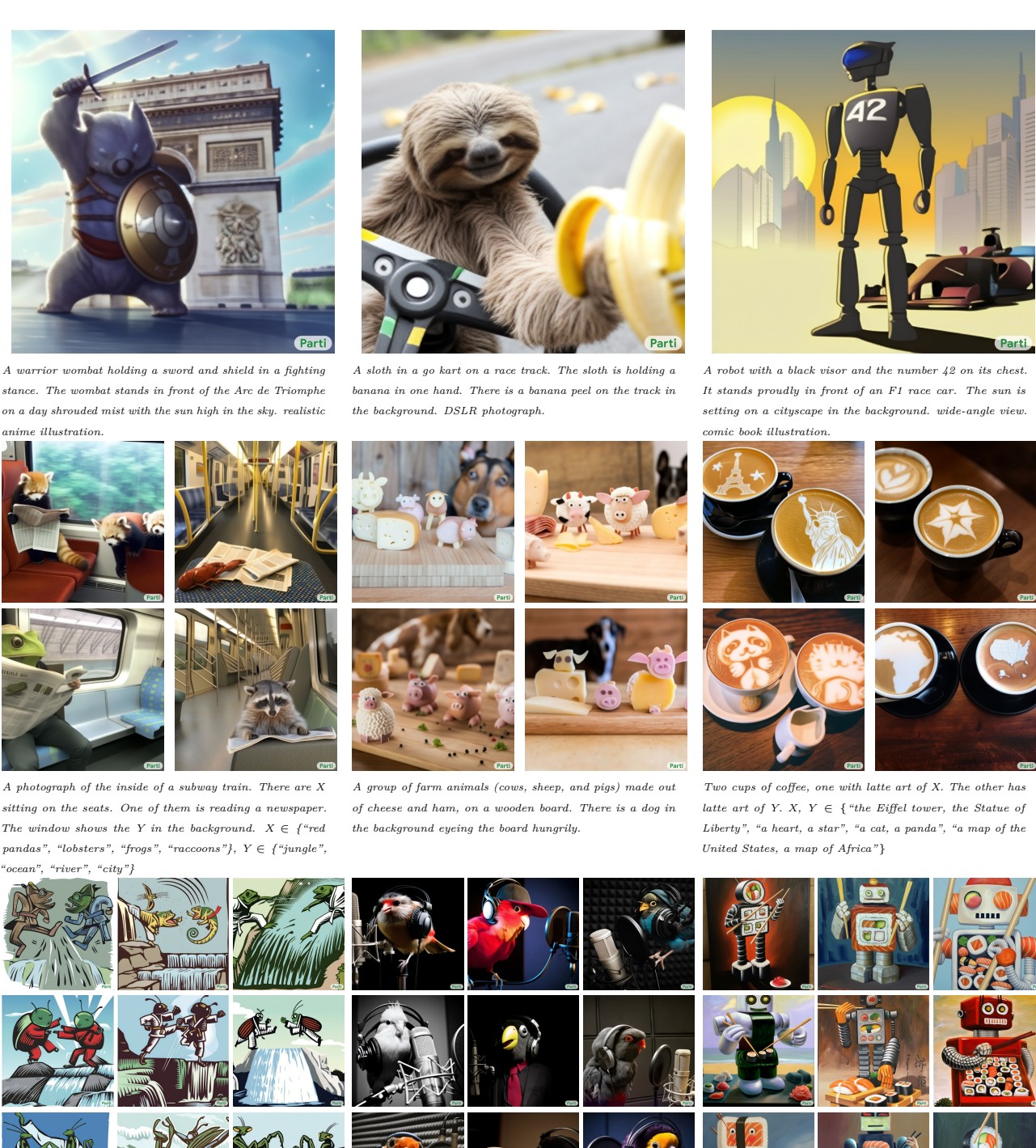

*A warrior wombat holding a sword and shield in a fighting stance. The wombat stands in front of the Arc de Triomphe on a day shrouded mist with the sun high in the sky. realistic anime illustration.*

*A sloth in a go kart on a race track. The sloth is holding a banana in one hand. There is a banana peel on the track in the background. DSLR photograph.*

*A robot with a black visor and the number 42 on its chest. It stands proudly in front of an F1 race car. The sun is setting on a cityscape in the background. wide-angle view. comic book illustration.*

*A photograph of the inside of a subway train. There are X sitting on the seats. One of them is reading a newspaper. The window shows the Y in the background. X ∈ {"red pandas", "lobsters", "frogs", "raccoons"}, Y ∈ {"jungle", "ocean", "river", "city"}*

*A group of farm animals (cows, sheep, and pigs) made out of cheese and ham, on a wooden board. There is a dog in the background eyeing the board hungrily.*

*Two cups of coffee, one with latte art of X. The other has latte art of Y. X, Y ∈ { "the Eiffel tower, the Statue of Liberty", "a heart, a star", "a cat, a panda", "a map of the United States, a map of Africa"}*

*A close-up of two X wearing karate uniforms and fighting, jumping over a waterfall. color woodcut illustration. X ∈ { "chameleons", "beetles", "matnis"}*

*A photograph of a bird wearing headphones and speaking into a high-end microphone in a recording studio.*

*Oil painting of a giant robot made of sushi, holding chopsticks.*

Figure 17: Selected Parti images.

those in the COMPLEX and IMAGINATION challenge dimensions. This aligns with our observations on the significant improvement of the 20B model on prompts from the ABSTRACT category.

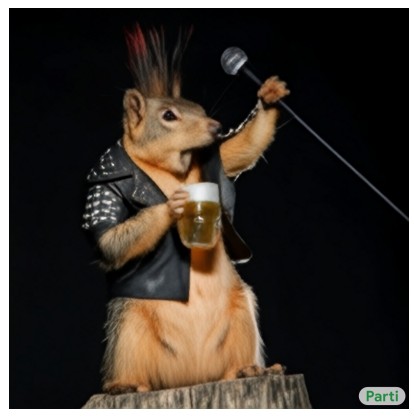
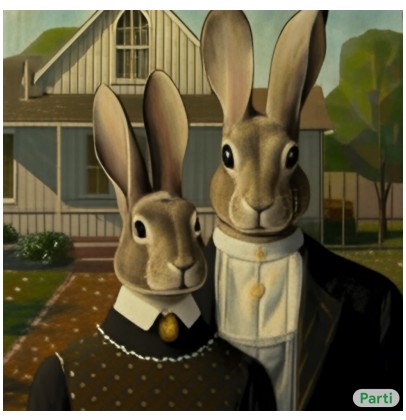
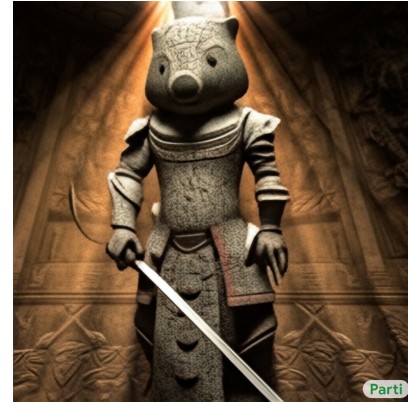

*A punk rock squirrel in a studded leather jacket shouting into a microphone while standing on a stump and holding a beer on dark stage. dslr photo.*

*An oil painting of two rabbits in the style of American Gothic, wearing the same clothes as in the original.*

*A soft beam of light shines down on an armored granite wombat warrior statue holding a broad sword. The statue stands an ornate pedestal in the cella of a temple. wide-angle lens. anime oil painting.*

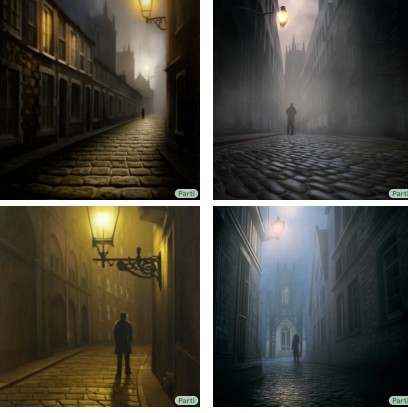
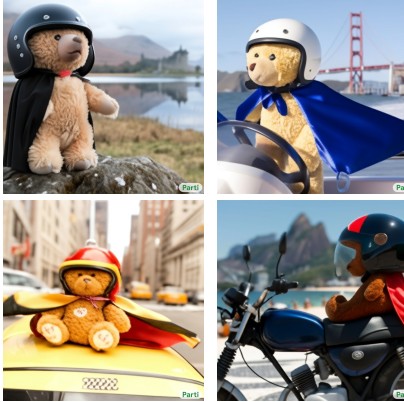
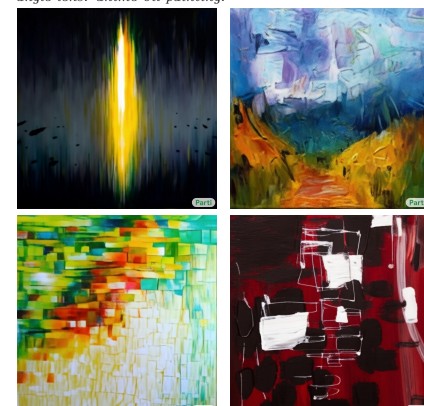

*A solitary figure shrouded in mists peers up from the cobble stone street at the imposing and dark gothic buildings surrounding it. an old-fashioned lamp shines nearby. oil painting.*

*A teddy bear wearing a motorcycle helmet and cape is X. dslr photo. X ∈ {standing in front of Loch Awe with Kilchurn Castle behind him, driving a speed boat near the Golden Gate Bridge, car surfing on a taxi cab in New York City, riding a motorcycle in Rio de Janeiro with Dois Irmãos in the background}*

*(a) the door of knowing, a portal brightly opening the way through darkness. abstract anime landscape oil painting.; (b) trying to find my way in a big confusing world, abstract oil painting; (c) light and happiness throughout and finding its way to every corner of the world, abstract oil painting; (d) an abstract oil painting in deep red and black with a thick patches of white*

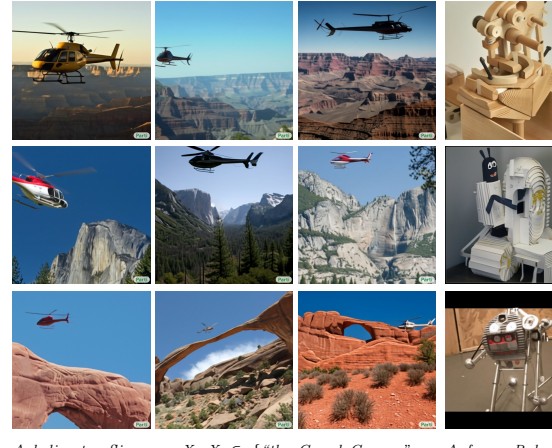
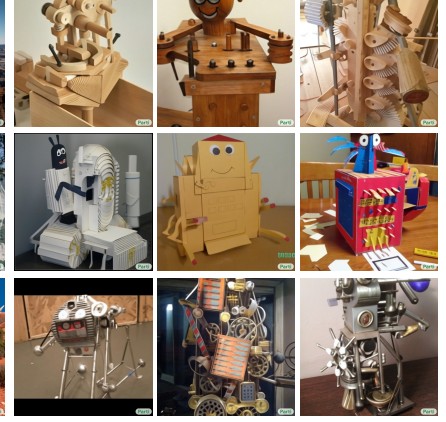
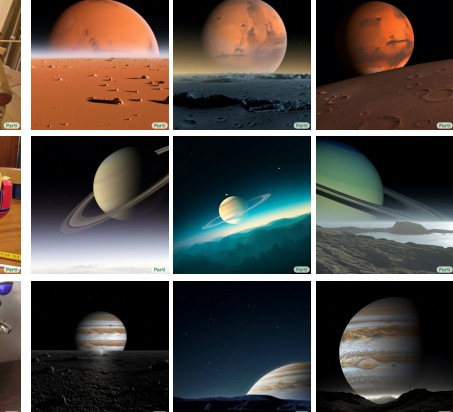

*A helicopter flies over X. X ∈ {"the Grand Canyon", "Yosemite", "the Arches National Park"}*

*A funny Rube Goldberg machine made out of X. X ∈ {"wood", "paper", "metal"}*

*X rises on the horizon. X ∈ {"Mars", "Saturn", "Jupiter"}*

Figure 18: Selected Parti images.

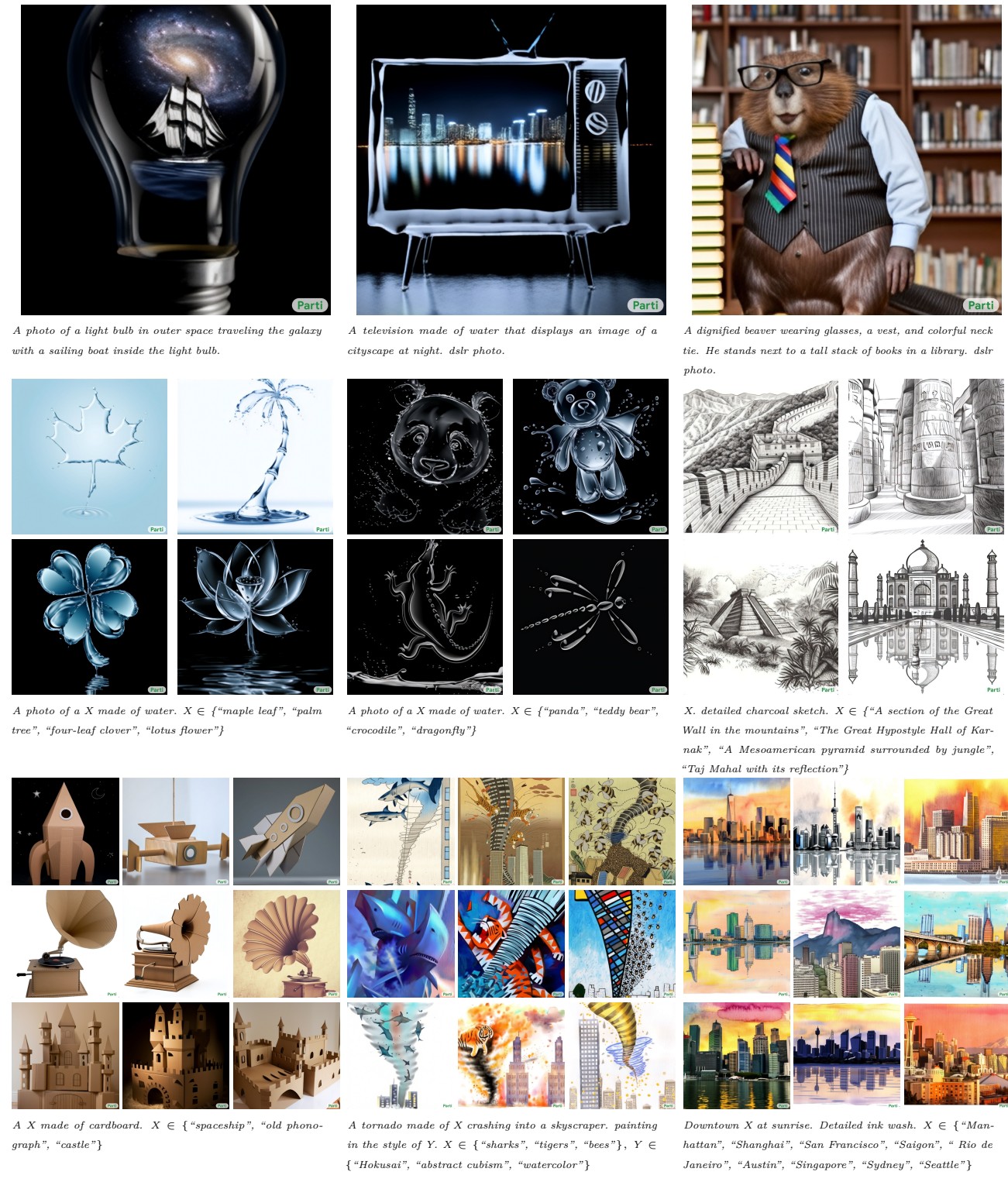

*A photo of a light bulb in outer space traveling the galaxy with a sailing boat inside the light bulb.*

*A television made of water that displays an image of a cityscape at night. dslr photo.*

*A dignified beaver wearing glasses, a vest, and colorful neck tie. He stands next to a tall stack of books in a library. dslr photo.*

*A photo of a X made of water. X ∈ {"maple leaf", "palm tree", "four-leaf clover", "lotus flower"}*

*A photo of a X made of water. X ∈ {"panda", "teddy bear", "crocodile", "dragonfly"}*

*X. detailed charcoal sketch. X ∈ {"A section of the Great Wall in the mountains", "The Great Hypostyle Hall of Karnak", "A Mesoamerican pyramid surrounded by jungle", "Taj Mahal with its reflection"}*

*A X made of cardboard. X ∈ {"spaceship", "old phonograph", "castle"}*

*A tornado made of X crashing into a skyscraper. painting in the style of Y. X ∈ {"sharks", "tigers", "bees"}, Y ∈ {"Hokusai", "abstract cubism", "watercolor"}*

*Downtown X at sunrise. Detailed ink wash. X ∈ {"Manhattan", "Shanghai", "San Francisco", "Saigon", "Rio de Janeiro", "Austin", "Singapore", "Sydney", "Seattle"}*

Figure 19: Selected Parti images.

## F.2  Comparison Against Retrieval Baseline

In addition to comparing our 20B and 3B models, we also run human evaluations of the Parti 20B model against the retrieval baseline. Figure 27 and 28 show human preference of the 20B model (over the retrieval

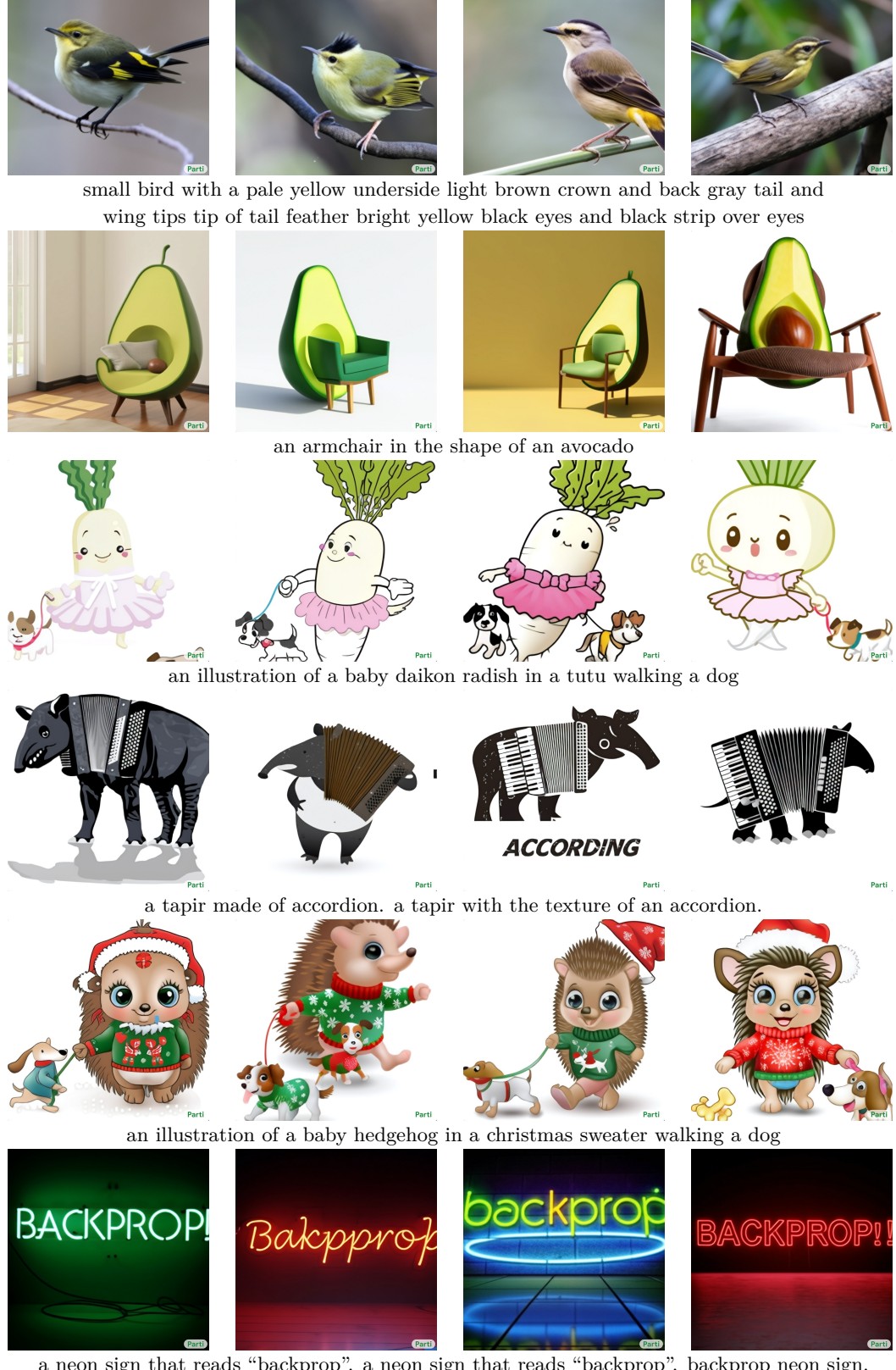

small bird with a pale yellow underside light brown crown and back gray tail and
wing tips tip of tail feather bright yellow black eyes and black strip over eyes

an armchair in the shape of an avocado

an illustration of a baby daikon radish in a tutu walking a dog

a tapir made of accordion. a tapir with the texture of an accordion.

an illustration of a baby hedgehog in a christmas sweater walking a dog

a neon sign that reads "backprop". a neon sign that reads "backprop". backprop neon sign.

Figure 20: Parti image samples with text prompts from DALL-E (Ramesh et al., 2021) for cross reference
and comparison.

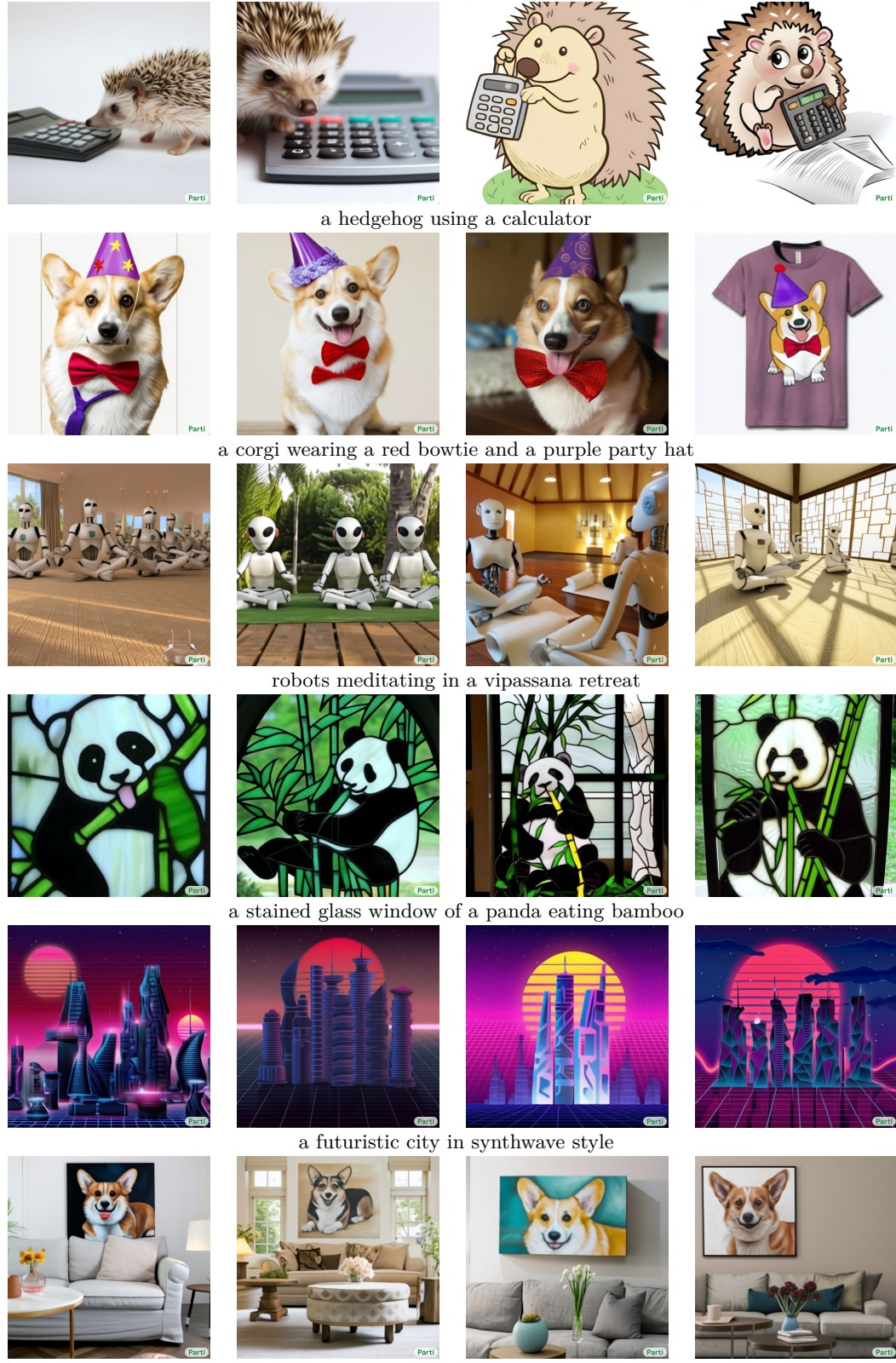

a hedgehog using a calculator

a corgi wearing a red bowtie and a purple party hat

robots meditating in a vipassana retreat

a stained glass window of a panda eating bamboo

a futuristic city in synthwave style

A cozy living room with a painting of a corgi on the wall above a couch and
a round coffee table in front of a couch and a vase of flowers on a coffee table

Figure 21: Parti image samples with text prompts from GLIDE (Nichol et al., 2022) for cross reference and comparison.

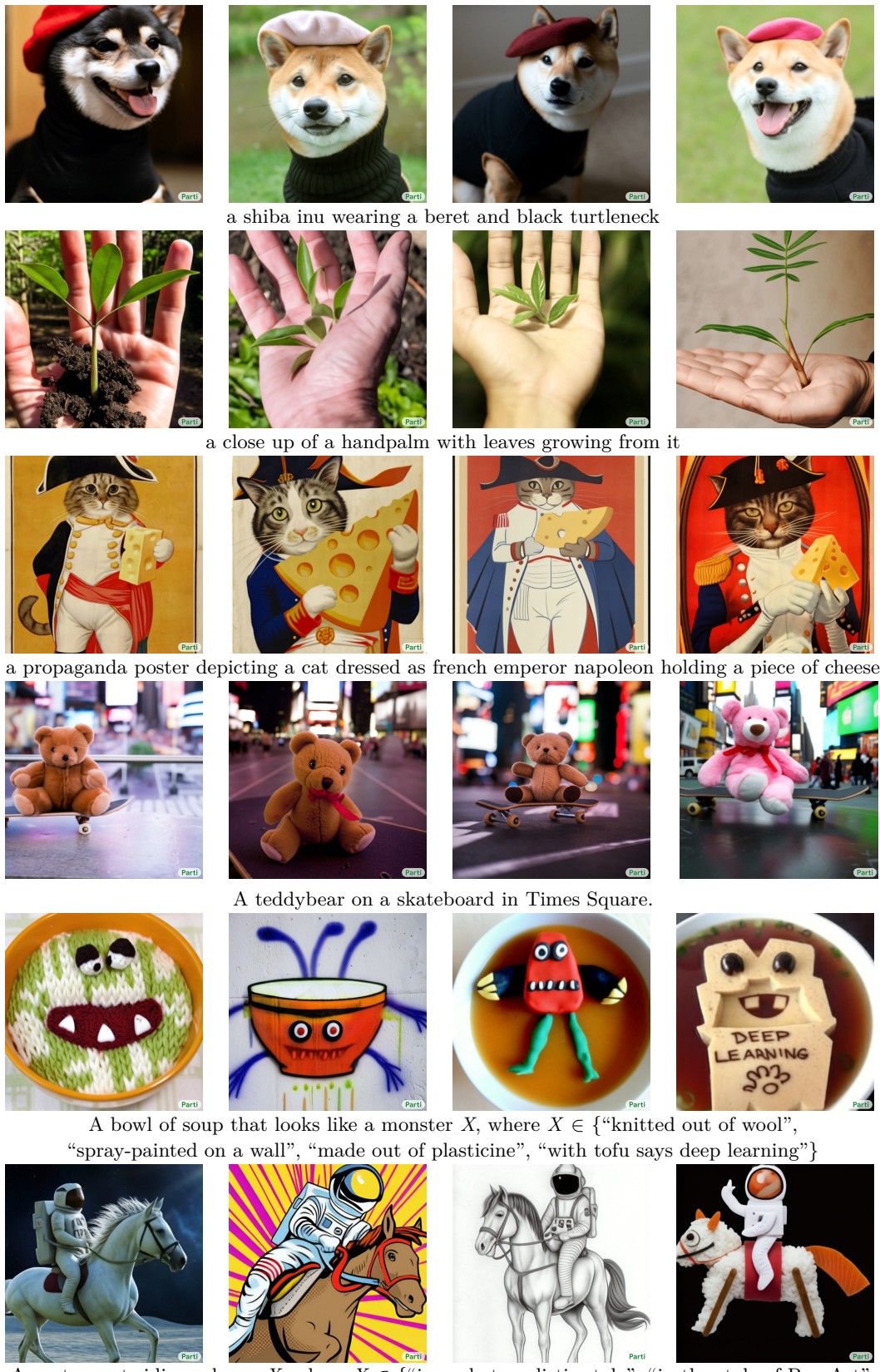

a shiba inu wearing a beret and black turtleneck

a close up of a handpalm with leaves growing from it

a propaganda poster depicting a cat dressed as french emperor napoleon holding a piece of cheese

A teddybear on a skateboard in Times Square.

A bowl of soup that looks like a monster $X$, where $X \in \{$"knitted out of wool",
"spray-painted on a wall", "made out of plasticine", "with tofu says deep learning"$\}$

An astronaut riding a horse $X$, where $X \in \{$"in a photorealistic style", "in the style of Pop Art",
"as a pencil drawing", "made out of sushi" (the last sub-prompt is not from DALL-E 2)$\}$

Figure 22: Parti image samples with text prompts from DALL-E 2 (Ramesh et al., 2022) (*a.k.a.*, unCLIP) for cross reference and comparison.

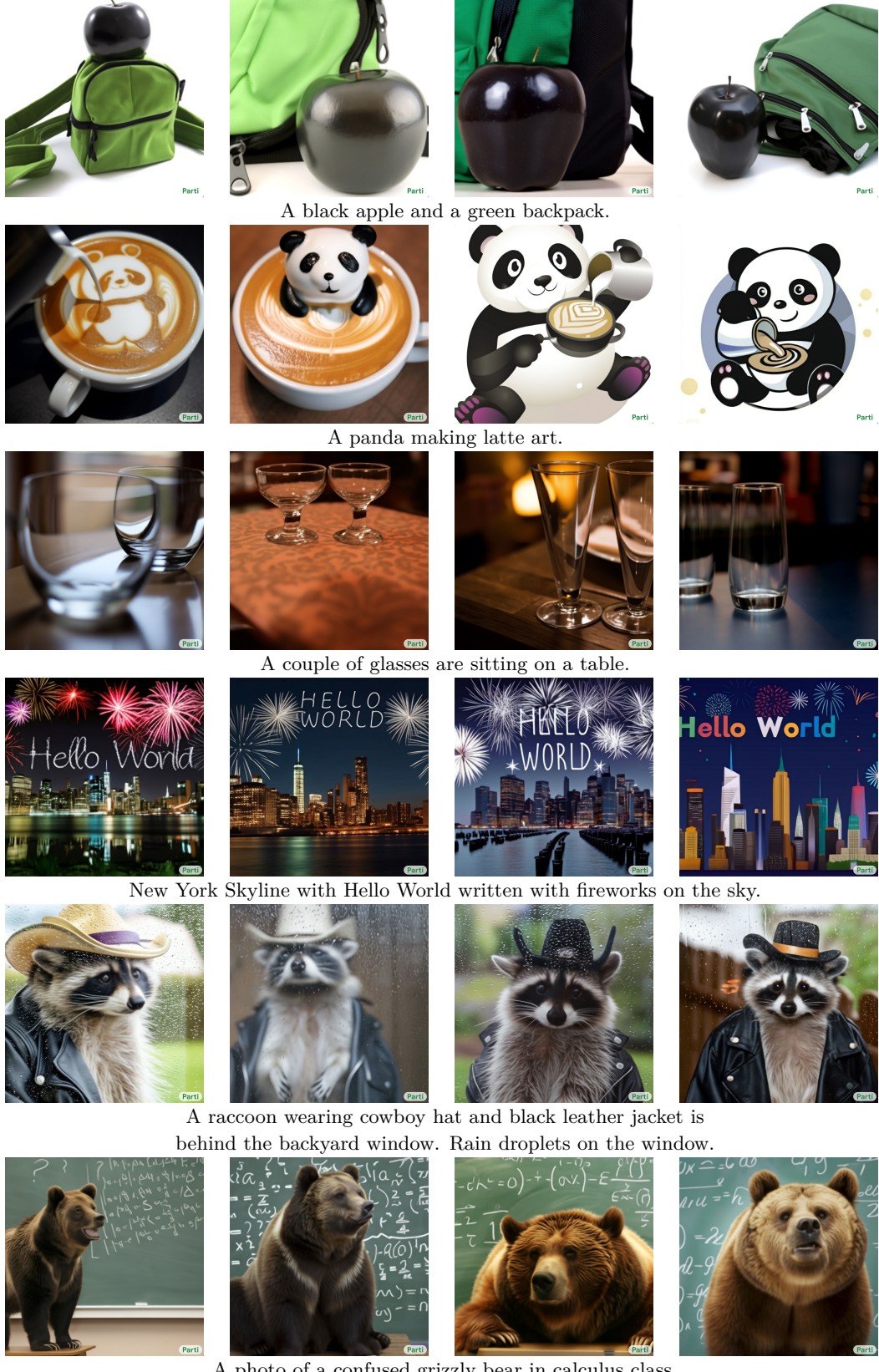

A black apple and a green backpack.

A panda making latte art.

A couple of glasses are sitting on a table.

New York Skyline with Hello World written with fireworks on the sky.

A raccoon wearing cowboy hat and black leather jacket is
behind the backyard window. Rain droplets on the window.

A photo of a confused grizzly bear in calculus class.

Figure 23: Parti image samples with text prompts from Imagen and DrawBench (Saharia et al., 2022) for
cross reference and comparison.

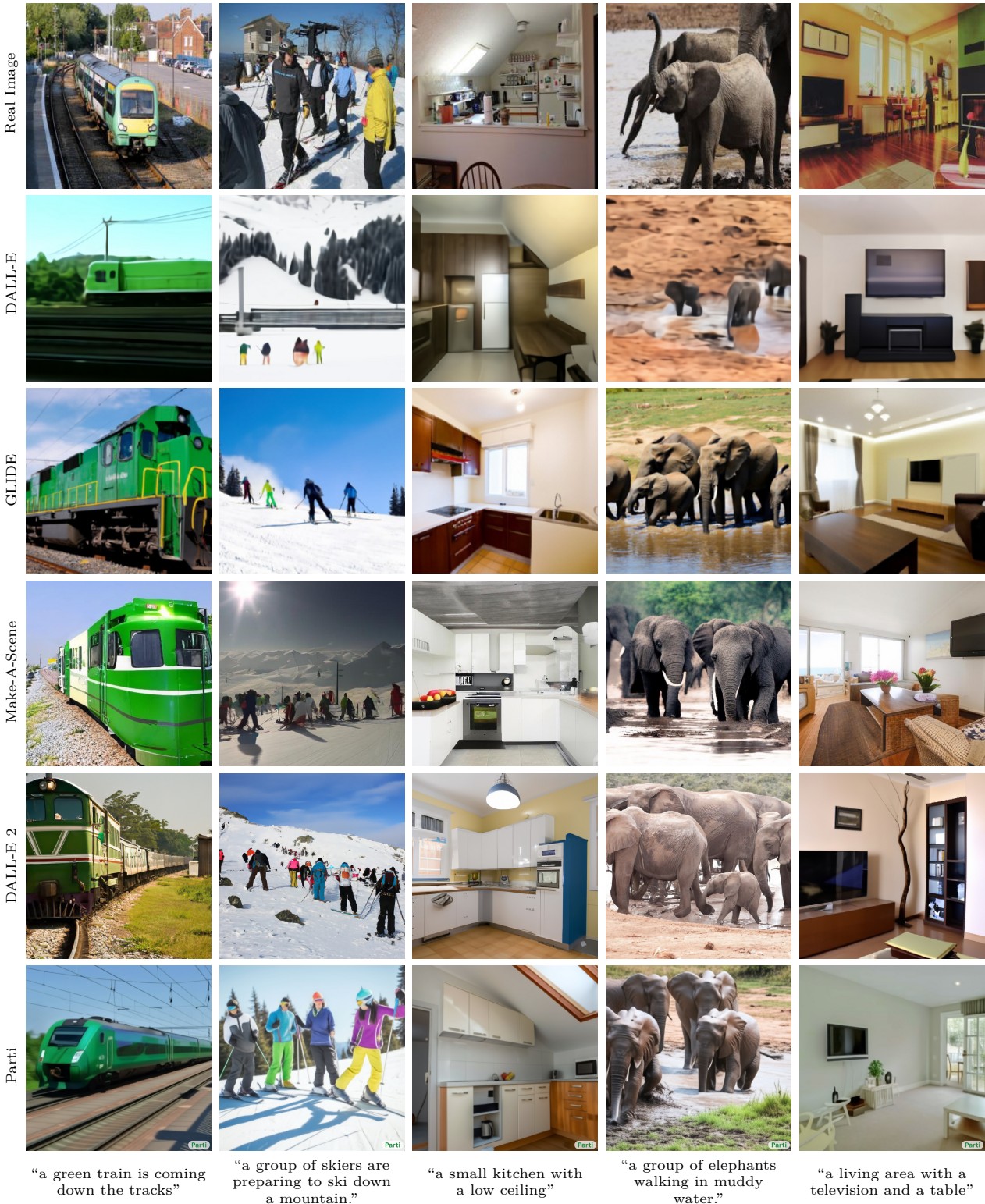

Figure 24: Qualitative comparison of zero-shot images sampled (not cherry picked) from Parti and other models on MS-COCO prompts. Parti model samples 16 images per text prompt and uses a CoCa model (Yu et al., 2022b) for re-ranking (Section 2.4).

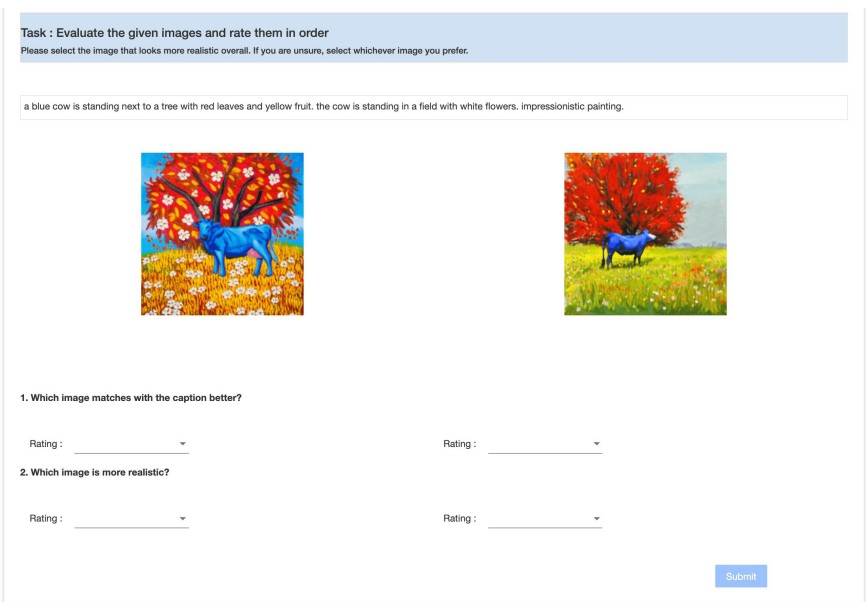

Figure 25: Human evaluation setup.

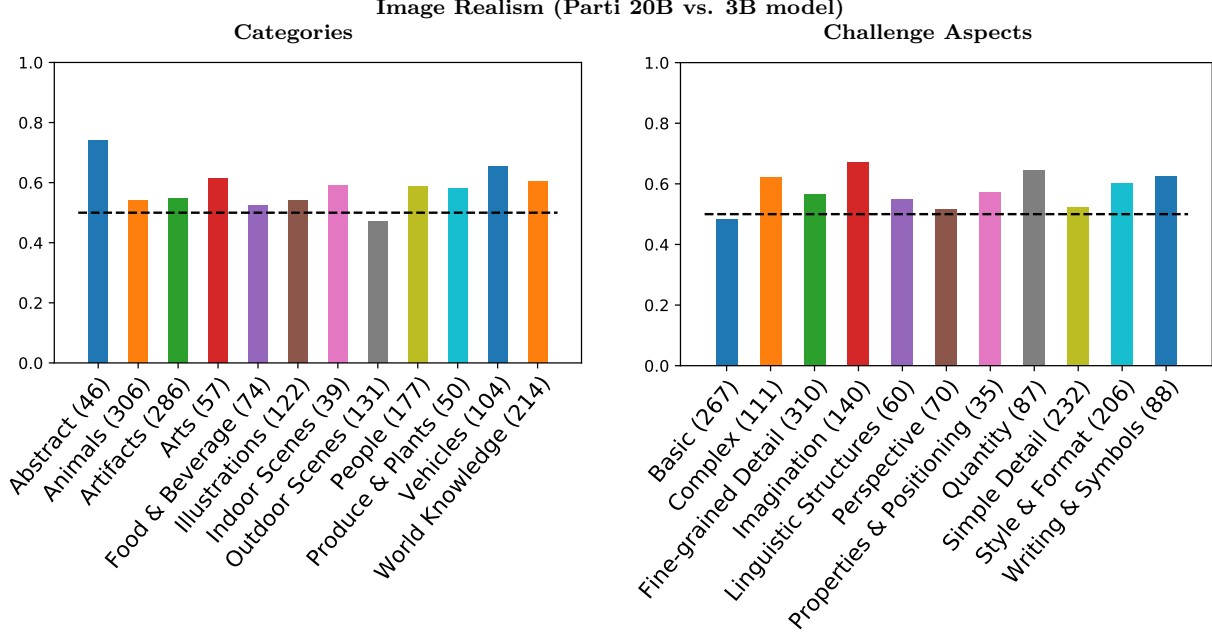

Figure 26: Breakdown of human preferences of the realism of images from the Parti 20B model over the 3B model in terms of P2 categories (*left*) and challenge aspects (*right*).

baseline) across P2. On image-text match (Figure 27), the Parti model significantly outperforms the retrieval baseline in most categories, except for ABSTRACT. This highlights the difficulty in generating good images for ABSTRACT prompts, which may be a potential area of improvement for future work. Along different challenge aspects, Parti outperforms the retrieval baseline in every category on image-text match, which we attribute to the improved ability of Parti to synthesize images for diverse and complex prompts. Many of the harder prompts in the P2 are 'adversarial' in nature, and were designed to test the limits of text-to-image synthesis models. These prompts are generally unlikely to have appeared in standard training datasets, and

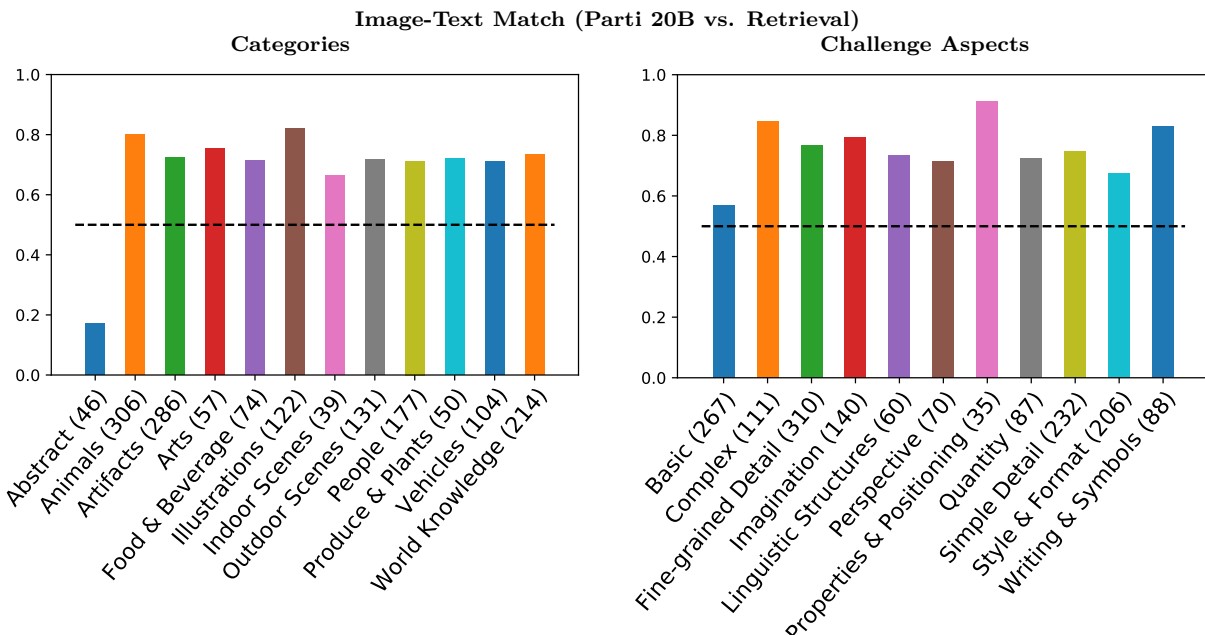

Figure 27: Breakdown of human preferences of the image-text match of images from the Parti 20B model over the retrieval baseline in terms of P2 categories (*left*) and challenge aspects (*right*).

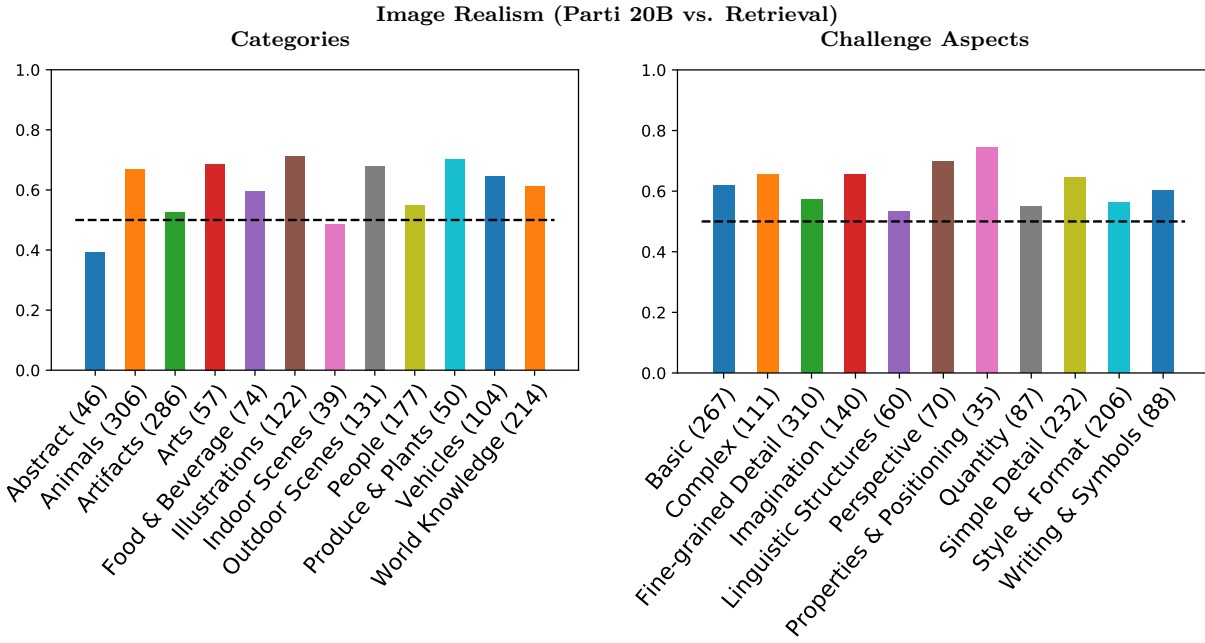

Figure 28: Breakdown of human preferences of the realism of images from the Parti 20B model over the retrieval baseline in terms of P2 categories (*left*) and challenge aspects (*right*).

consequently difficult for retrieval based models which rely on retrieving the closest match from our training corpus. The positive results along this evaluation highlight the strength of a generative model such as Parti, and its ability to handle diverse and creative prompts.

| Model | CoLa(↑) | SST2 (↑) | RTE (↑) | MRPC (↑) | QQP (↑) | QNLI (↑) |
|---|---|---|---|---|---|---|
| BERT | 54.6 | 92.5 | 62.5 | 87.6 | 87.4 | 91.0 |
| UNITER | 37.4 | 89.7 | 55.6 | 80.3 | 85.7 | 86.0 |
| SimVLM | 46.7 | 90.9 | 63.9 | 84.4 | 87.2 | 88.6 |
| CLIP | 25.4 | 88.2 | 55.2 | 65.0 | 53.9 | 50.5 |
| FLAVA | 50.7 | 90.9 | 57.8 | 86.9 | 87.2 | 87.3 |
| Encoder after encoder pretraining | 55.2 | 95.9 | 59.0 | 90.9 | 88.5 | 90.1 |
| Encoder after encoder-decoder training | | | | | | |
| (w/o encoder pretraining) | 15.7 | 82.5 | 49.5 | 81.5 | 76.8 | 66.6 |
| (w/ encoder pretraining) | 20.4 | 84.8 | 53.4 | 78.3 | 76.7 | 78.2 |

Table 9: Results of Parti text encoder on the GLUE benchmark after pretraining with joint BERT and CLIP objectives or (continued) training with text-to-image generation objective.

When evaluated on image realism (Figure 28), Parti fares well against the retrieval baseline, and is preferred by human annotators across the majority of the P2 categories (except ABSTRACT and INDOOR SCENES), as well as preferred along all challenge aspect dimensions. We note that this is a remarkable feat, considering that the retrieval model retrieves *real images* from a training corpus. These results highlight the ability of Parti to synthesize realistic images, which are deemed by human evaluators as close to photorealistic (or even more photorealistic, in some cases).

# G Encoder Pretraining

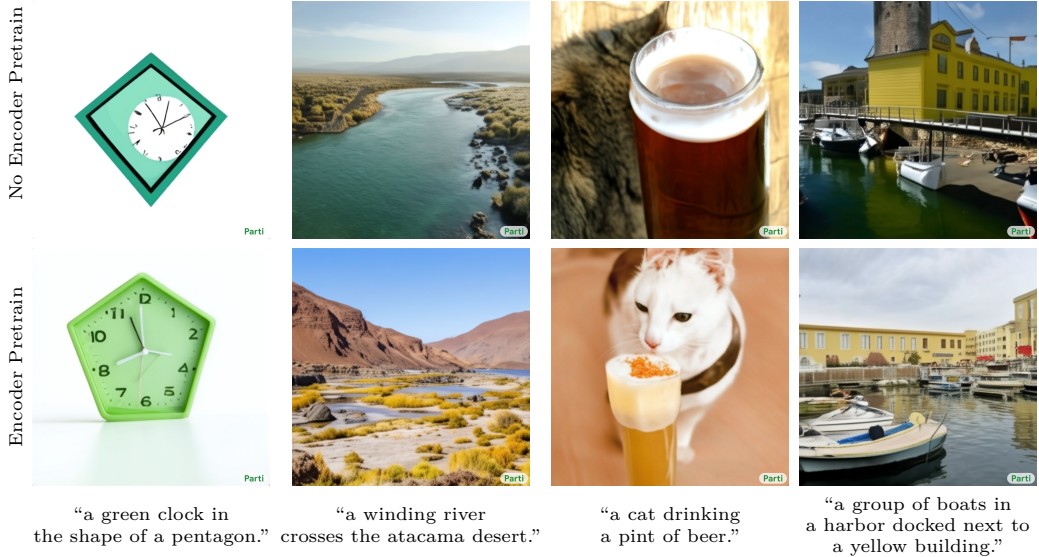

"a green clock in the shape of a pentagon."  "a winding river crosses the atacama desert."  "a cat drinking a pint of beer."  "a group of boats in a harbor docked next to a yellow building."

Figure 29: Ablation of text encoder pretraining. In some of the prompts, we observe text-pretrained model outperforms non-pretrained encoders as examples shown above. However on average, we observe *no significant* quality improvement by warming-up text encoder. Both models are with 3B parameters trained on the same mixture of datasets for ablation.

While it is straightforward to warm-start the model with a pretrained text encoder, we observe the text-encoder pretraining *very marginally* helps text-to-image generation loss with 3B-parameter Parti models. Qualitative examples are shown in Figure 29 and quantitative loss comparison is shown in Figure 30. We leave this observation as a future research topic on the difference and unification of generic language understanding and visually-grounded language understanding.

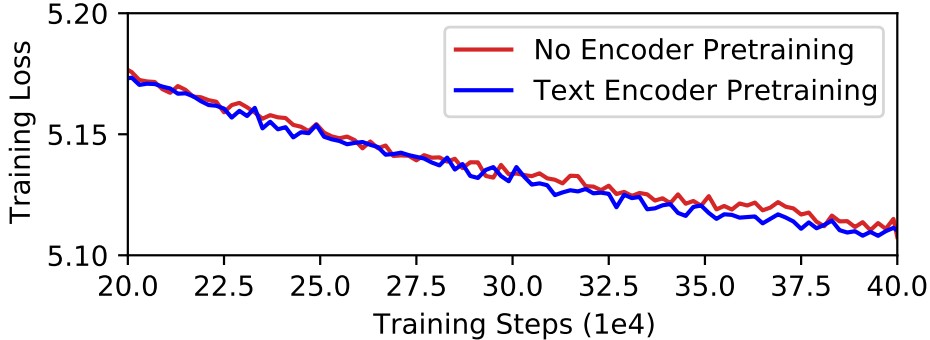

Figure 30: Ablation of text encoder pretraining. We plot text-to-image generation softmax cross-entropy training loss. The training of pretrained text encoder is only slightly better. Both models are with 3B parameters trained on the same mixture of datasets for ablation.

## H  Pixelation Patterns of ViT-VQGAN

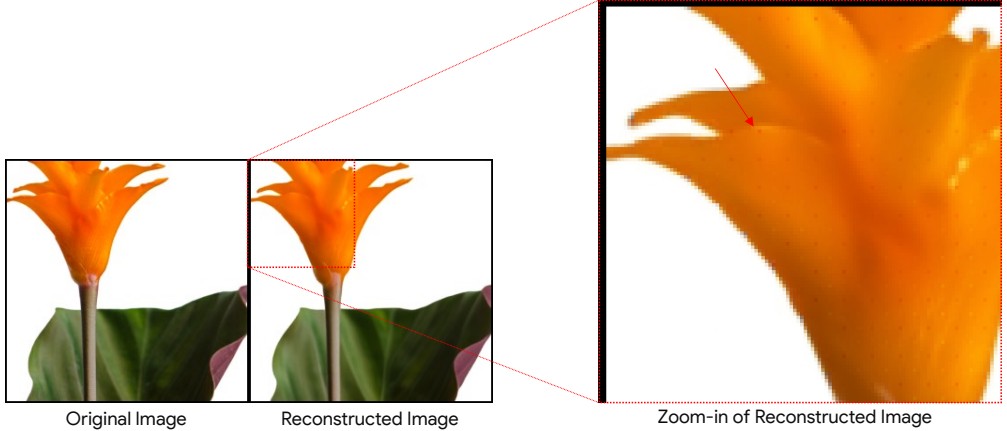

Figure 31: Example of pixelation patterns (saturated pixel values at some locations) in the outputs of ViT-VQGAN (Yu et al., 2022a) architecture when zooming in. It can be fixed by removing the final sigmoid activation layer and the logit-laplace loss, exposing the raw values as RGB pixel values (in range [0, 1]).

We notice visual pixelation patterns in some of the output images of ViT-VQGAN when zooming in (see Appendix H), and further find ill-conditioned weight matrices of the output projection layer before the sigmoid activation function. As a fix, we remove the final sigmoid activation layer and the logit-laplace loss, exposing the raw values as RGB pixel values (in range [0, 1]). Conveniently, this fix can be hot-swappable into an already trained image tokenizer by finetuning the decoder.

