# OpenReview forum: "Scaling Autoregressive Models for Content-Rich Text-to-Image Generation"
_TMLR — Accepted by TMLR_

### Review · Reviewer_Eo75 · 2022-09-16

**Summary Of Contributions:**

Patri, built on image tokenization and autoregressive models, has achieved state-of-the-art performance on multiple benchmarks such as MS-COCO, LN-COCO. Its architecture is very simple, with only encoder-decoder and image tokenzier, making it possible to scale up. Moreover, distributed and parallel mechanism of Pathways further makes it efficient for training. Thus, it demonstrates the scaling law of text-to-image generation models for the first time. It has also provided a comprehensive and systematic analysis of failure modes of text-to-image models with the newly released diverse and multi-dimensional PartriPrompts set which points out numerous future directions. However, due to a lack of controllability and explainability, it may suffer from low security and biases. Although it is not absolutely necessary, it may be interesting to explore or compare text-to-image generation models with diffusion models.

**Broader Impact Concerns:**

The authors have provided sufficient discussions on broader impact concerns of this work.

**Requested Changes:**

No

**Strengths And Weaknesses:**

## Strengths

1. It is based on a simple architecture with only encoder-decoder and image tokenizer, making it easy to scale up. More importantly, the authors indeed make it scale up and demonstrate that scaling up autoregressive models improves the generation capabilities significantly.

2. It demonstrates the scaling law of text-to-image generation models which enables models to generate images with complex compositions and world knowledge.

3. It uses distributed and parallel mechanism of *Pathways,* making it efficient for training.

4. It achieves the SOTA results on multiple common benchmarks as well as the Localized Narratives dataset whose descriptions are four times longer than MS-COCO on average.

5. It has released a new prompt set *PartiPrompts* which is more diverse and multi-dimensional, giving a more comprehensive picture of text-to-image generation models. In virtue of this set, it gives a systematic analysis of Patri model with a detailed breakdown of error types.

## Weaknesses

I don't really notice any weakness for such a paper focusing on scaling up existing tools/concepts and making it work. The authors have already provided in-depth discussions on the limitations.

---

> ### Author Response · Authors · 2022-10-01
> **Authors' Responses to Reviewer Eo75**
>
> Thanks for your time reviewing and providing suggestions (and all the positive comments) for our manuscript! We agreed that “it may be interesting to explore or compare text-to-image generation models with diffusion models” and we are conducting follow-up studies as a future work!

---

### Review · Reviewer_TitR · 2022-09-19

**Summary Of Contributions:**

In general, the paper introduces a scaling recipe for text-to-image generation, which shows impressive image generation results both qualitatively and quantitively. Here is a summary of the contributions:

1. Proposing a simple encoder-decoder Transformer architecture, together with previously introduced ViT-VQGAN (as image tokenizer and detokenizer), for text-to-image generation. The encoder-decoder Transformer can be scaled up to 20B parameters, which achieves impressive results on MS-COCO/Localized Narratives benchmark.

2. Proposing a P2 benchmark, with 1600 prompts covering diverse concepts and challenging aspects. This benchmark can be a useful asset to prompt and stress test future text-to-image generation models.

3. In-depth analyses conducted on the generated images, such as varying model scales, comparing against retrieval-baseline and etc. The growing a cherry tree analysis is especially interesting and may pose as a better visualization tool than simply cherry picking best results from a few independent prompts.

**Broader Impact Concerns:**

The authors have discussed Broader Impacts in Section 8 of the paper.

**Requested Changes:**

- Questions regarding scaling-up recipe
1. As an important contribution of the paper is the scaling-up recipe, I feel there are some missing details regarding scaling law observed during pre-training. For example, Is it always the case across different model scales, the more data the better?  We have seen some studies on the scaling law on vision transformers and generative pre-training for image-to-text generation. I am wondering whether the same scaling law applies here.
2. Another important insight benefiting future studies maybe which model component is more important when scaling Parti model. The paper provides some scaling designs, for example the decoder has much more parameters than encoder, a much bigger image tokenizer decoder is trained, in contrast to the frozen image tokenizer encoder. Some brief reasoning has been provided, but there is no quantitative results supporting these design choices. Further, it is not clear whether scaling up the image tokenizer can further benefit the model performance.
3. It would be better if including model parameter counts of other methods and compare against Parti in Table 5.
4. Missing insights on pre-training data scaling, especially on the multimodal pre-training datasets. I am especially curious about the following questions: (1) For JFT-4B, how important it is to randomly replace the original label with machine generated captions. (2) For FIT400M, what is the filtering rules? what if directly use the full 1.8B data from ALIGN. Would similar filtering on LAION-400M benefit the model training, as both are web-crawled images with alt-text descriptions?

- Questions about retrieval baseline
1. What is the R1/5/10 scores of retrieval baseline on MS-COCO and LN-COCO?

- Questions about P2 benchmark
1. Prompts in P2 benchmark are interesting and useful, but as there is no ground truth image for all these prompts, the only way to evaluate a model is through human evaluation. Human evaluation is generally very expensive, especially if we are comparing more and more models. This should not be considered as a drawback of P2 benchmark. However, I am wondering whether the authors have insights on alternative evaluation methods.

- Additional missing details
1. Which model size of SimVLM (for generating captions on JFT) and CoCa (for re-ranking generated outputs) is used?
2. Regarding the frozen VIT-VQGAN tokenizer. It is not clear how much advantages are from retraining ViT-VQGAN tokenizer on "images on our pre-training data", in comparison to direct adoption of a pre-trained ViT-VQGAN tokenizer from the original paper.
3. For re-ranking, 16 images per text prompt are adopted by default, while DALL-E uses 512 examples. Would more images always benefit Parti performance? Including a curve to illustrate the tradeoff between performance and computation might be better.

- Minor paper edits
1. Duplicated "and" found in the last sentence of "Automatic image-text alignment." paragraph on page 14. "and then the similarity of the input prompt **and and** the generated caption is assessed via BLEU"
2. Section 2.2 mentions "all models use conv-shaped masked sparse attention", which reads like both encoder and decoder across all model scales uses conv-shaped masked sparse attention. It becomes more clear in section 3 that only decoder uses conv-shaped masked sparse attention. Also need to more clearly point out the reference of Appendix B.1 Figure 11 is to the DALL-E paper.

**Strengths And Weaknesses:**

## Strengths

1. The final scaled-up Parti model achieves impressive quantitative results on two datasets, compared to previous methods. In addition, the generated images are in very good quality and the selected visualization examples are quite interesting, demonstrating the strong ability of Parti model to understand abstract and complex concepts.

2. The paper presents in-depth analysis on generated images. I especially like the discussions in Section 6.2 and 6.3. Section 6.2 suggests a better visualization method than just cherry picking best results from a few independent prompts. The limitation of Parti model shown in Section 6.3 point out potential future directions for follow-up studies.

3. The P2 benchmark provided can be a valuable asset for model evaluation in future studies.

## Weaknesses

0. It would be beneficial for the community if the smallest model trained on public dataset (for example, on C4 and LAION400M) can be released.

1. As the paper emphasizes on scaling, there are some missing discussions or results regarding the scaling up recipe. Please refer to the next section for more details.

2. For model pre-training and evaluation, multiple large-scale image-text models are used, for example, SimVLM to generate textual descriptions on JFT, CoCa to rerank the generated outputs, and ALIGN-base to build the retrieval baseline. One concern is that all these model are not publicly released. And the other concern is why not use CoCa for all, as CoCa achieves better results in image captioning and ZS retrieval than SimVLM and ALIGN.

3. There are some missing details in addition to the missing discussions on scaling up recipe, which I listed in the next section.

---

> ### Author Response · Authors · 2022-10-02
> **Authors' Responses to Reviewer TitR**
>
> Thanks for your time reviewing and providing detailed suggestions for our manuscript! We have updated the manuscript according to your suggestions and addressed all concerns below.
>
> 1. It would be beneficial for the community if the smallest model trained on public dataset (for example, on C4 and LAION400M) can be released.
>
> Thanks for openly raising this question! As we also mentioned in our reply to Reviewer QQkZ, we discussed in detail in the Section 8 Broader Impacts, tasks especially involving image generations raise a number of issues that should be considered. The code of this work and the models we trained might be used for malicious purposes without proper licenses and agreements. Due to these reasons, we have firstly released our model card and data card. Meanwhile we are happy to take any technical questions on implementation details that are not clear. To all readers and reviews, please feel free to leave any technical questions on OpenReview.
>
> 2. Discussions or results regarding the scaling up recipe (model, data, quantizer)
>
> We totally agree that studying detailed scaling up laws is of significant importance and interest. In this work, we presented results on a simple scaling recipe by 4 model sizes and demonstrated consistent gains and state-of-the-art results. A detailed study on scaling law (model vs. data; encoder vs. decoder; language model vs. image quantizer; different data compositions) might further help us reduce the computation and improve the quality. We hope our current Parti results could be an encouraging milestone to motivate future work on more detailed scaling laws of text-to-image generation models collaboratively.
>
> 3. Why not use CoCa for all, as CoCa achieves better results in image captioning and ZS retrieval than SimVLM and ALIGN.
>
> Good question! This is due to the practical timeline limitation of research development. The main reason is that the captioning data was prepared and the Parti models were trained before the CoCa model is available. Retraining Parti models is non-trivial in terms of required computation and time. We note that although CoCa model achieves better results than SimVLM on image captioning, the gap (e.g., CIDEr score) isn’t huge. Thus the difference between SimVLM and CoCa won’t affect the results presented in this Parti work.
>
> 4. It would be better if including model parameter counts of other methods and compare against Parti in Table 5.
>
> Thanks for the suggestion! The authors discussed and hesitated to include parameter counts in Table 5 as 1) We focus our comparisons on the Parti family of models (Figure 9), for which we can fairly compare parameters counts and measure performance consistently. 2) It is non-trivial to get parameters counts correct for all models we cited as some work required pretrained and frozen model (like Imagen used frozen T5 model, DALLE-2 used frozen CLIP model / encoder) and the table gets lengthy if we include all the details. Thus we report our own parameter counts and refer readers to individual work for comparison of model sizes and computations with readers' own standards.
>
> 5. R1/5/10 scores of retrieval baseline on MS-COCO and LN-COCO
>
> For the retrieval baseline, we ran on both datasets, and got a CLIP-R-Precision of around 98.6-99%. Using different subsets of 30k examples from mscoco2014 yields roughly the same result. This is expected: the retrieval baseline “generates” images by text-to-image retrieval and in turn has best image-to-text retrieval results, since the CLIP/ALIGN model can embed them in a “uniquely identifying” way. We mainly report retrieval baseline’s captioning results of DALL-Eval in Table 6.
>
> 6. Ground truth of P2 benchmark
>
> Good question! It is true that there is no ground truth image for P2 prompts, and we mainly use human evaluation SxS to evaluate the quality of different text-to-image generation methods. We presented multiple evaluation results including FID, finetuned FID, next-token-prediction loss, DALL-Eval, Human Side-by-Side, retrieval baseline, and hope our model and results presented in this paper could inspire the community towards better, standardized and automated evaluation benchmarks for text-to-image generation.
>
> 7. Additional missing details
>
> We have modified the drafts to include these details for clarification! We use SimVLM-Huge (to generate high-quality captions) and CoCa-base reranker (to reduce latency). For ViT-VQGAN tokenizer (trained on Imagenet vs. trained on current data) we don’t have direct comparison yet as it requires retraining the encoder-decoder. For sampling size vs Parti performance, our empirical finding is that reranking helps the text-to-image generation demo to surface better image examples on the front, improving users experience. We will be conducting some comparison together with ablating tokenizers in our next version.
>
> 8. Minor paper edits
>
> Thanks for pointing out the typos of this work! We have fixed them with revised texts in blue colors.

---

### Review · Reviewer_QQkZ · 2022-09-26

**Summary Of Contributions:**

This paper presents pathways autoregressive text-to-image model (Parti) for high-fidelity photorealistic image generation. Parti is an auto-regressive model similar to DALLE and achieves state-of-the-art performance on image generation (compared to DALLE2 and the latest diffusion models). The authors proposed multiple techniques to improve the image generation quality of Parti:

- Vit-VQGAN instead of VQ-VAE for image tokenizer;
- Encoder-Decoder model instead of Decoder only model (DALLE);
- Pre-training of Encoder with Large NLP datasets;
- Classifier free Guidance for training and inference.
- Scaling of dataset size and model size.

For better evaluation, the authors proposed PartiPrompts, which contains 1600 diverse English prompts to test the limit of test-to-image synthesis models. The authors did a comprehensive evaluation and human study for the proposed model and its ablations (model with smaller parameters). There are also very interesting discussions and "Growing a cherry Tree" to show the abilities and limits of the propose models.

**Broader Impact Concerns:**

No concerns about ethical implications.

**Requested Changes:**

- More information about decoder conv-shaped masked sparse attention and comparison with full attention.
- More details about the prompt used in CF-guidance training and inference.
- Comparison with Stable-diffusion model with similar data and model scale.
- Discussion and thoughts on the usage proposed Parti model other than art, whether Parti can be used as a source of knowledge for other tasks such as GPT3 in the NLP domain.

**Strengths And Weaknesses:**

[Strengths]

Overall, Parti model has shown great potential for the auto-regressive model for image generation. The proposed method is simple, but with many novel techniques to improve the qualities of the generated images. The authors also perform a comprehensive evaluation and human study of Parti model and the limitation of the model is well discussed.

In terms of technical strength:
- The first model shows that the auto-regressive model can achieve similar or better image generation performance compared to the recent diffusion model (DALLE2).
- Compared to DALLE, Parti has taken advantage of recent techniques on image generation model, such as ViT-VQGAN for image tokenizer; Pre-training Encoder with NLP datasets; Classifier free Guidance similar to Make-A-SCENE; Re-Ranking using COCA; super large scale dataset size and model size.
- The proposed 16-stage GSPMD pipelines reduce the ratio of bubbles and enable large-scale training.

In terms of evaluation strength:
-  The authors utilize multiple evaluation metrics for Parti, such as automatic image quality, and image text alignments and achieve the best performance compared to the existing model.
-  The authors also create PartiPrompts, which provide very rich concepts for image synthesis evaluation and comprehensive human study has shown the clear advantage of the proposed model.
- Ablation study with multiple model size is very informative, it shows how parameter size can change the image generation quality.
- The discussion section and limitation section are very informative and shows the problem Parti model.

[Weaknesses]

My major concern with this paper is reproducibility. In section 8, the authors mentioned the reasons not to release the models, code, or data for Parti model and it is definitely understandable. However, the concern of bias, safety, and other limitations should not be a barrier to the research communities. It would be great if the authors can release smaller models or help the academic communities to reproduce the proposed methods. Besides, there are some details not clear in the model.
- In section 2.2, the authors mentioned the decoder use conv-shaped masked sparse attention. Could you provide more information on how the masked is created? Compared to full attention, is the mask mainly benefit the memory or speed or has better image generation qualities?
- In section 2.4, the authors mentioned CF-guidance in Parti, I wonder what is the prompt for unconditioned training G(z).
- In section 4.1, the training datasets contain a mix of open-sourced datasets (Laion-400m) and private datasets (FIT400M), etc. It would be great if the authors can train the proposed model on open-sourced datasets and have a fair comparison with existing diffusion models such as stable diffusion. This is also a very informative ablation study that can show the effect of data size on image generation qualities.

---

> ### Author Response · Authors · 2022-10-02
> **Authors' Responses to Reviewer QQkZ**
>
> Thanks for your time reviewing and providing detailed suggestions for our manuscript! We have updated the manuscript according to your suggestions and addressed all concerns below.
>
> 1. Questions on models, code, or data release of Parti model
>
> Thanks for openly raising this question! As we discussed in detail in the Section 8 Broader Impacts, tasks especially involving image generations raise a number of issues that should be considered, such as possible biases in underlying models and data, especially with respect to capabilities for people with different demographic backgrounds. The code of this work and the models we trained might be used for malicious purposes without proper licenses and agreements. Predicting all possible uses and consequences of infrastructure is difficult if not impossible. Due to these reasons, we have firstly released our model card and data card. Meanwhile we are happy to take any technical questions on implementation details that are not clear. To all readers and reviews, please feel free to leave any technical questions on OpenReview.
>
> 2. Questions on conv-shaped masked sparse attention. Could you provide more information on how the masked is created? Compared to full attention, is the mask mainly benefit the memory or speed or has better image generation qualities?
>
> Good question! We implemented a local sparse attention with row, column and conv-window visibility following DALL-E (detailed information can be found in DALLE’s Appendix B.1 Figure 11). The speed and memory benefits from sparse attention is highly dependent on implementation and hardware. As a data point: our training of Parti models is implemented with masked full-attention (with conv-shape masking) so it doesn’t give us gains on speed or memory. However, our sampling / decoding implementation has benefited from conv-shape masked sparse attention, which gives us about a ~2x speedup in latency for the 20B model. We expanded the sparse attention part in Section 2.2 (in blue texts).
>
> 3. Questions on what is the prompt for unconditioned training G(z).
>
> Thanks for the question! To clarify, for CF guidance training, we didn’t use any specific text prompt for unconditioned inputs. Instead, we simply mask out all text information by setting text ids to zeros and all text paddings to ones and the transformer encoder output will be a fixed learned vector, which will be used as the unconditioned inputs when we apply cf-guidance during sampling. We have added the clarification (in blue texts) in Section 2.4.
>
> 4. Questions on authors can train the proposed model on open-sourced datasets and have a fair comparison with existing diffusion models such as stable diffusion like on LAION data
>
> Thanks for raising this question and we agree training on open-sourced datasets is important and can provide more value to the community. We are actively working on a Parti model to be trained on the LAION data. However we also note that there are RAI and ethical implications of LAION data that the community needs to work together to improve further. We hope together we will be able to analyze different model variants trained on the same data with a more apple-to-apple comparison in future.
>
> 5.  Discussion and thoughts on the usage proposed Parti model other than art, whether Parti can be used as a source of knowledge for other tasks such as GPT3 in the NLP domain.
>
> Good question. We did find Parti model’s encoder is still capable of GLUE benchmarks with comparison to BERT and other multimodal models in Appendix Table 9. We note that Parti is a “language model” approach and we hope our results could be an encouraging milestone to motivate future work on unifying text and multimodal models, as well as building better benchmarks to understand models' capabilities (e.g., to better evaluate visually-grounded text understanding).

---

> > ### Comment · Reviewer_QQkZ · 2022-10-11
> > **Thanks for the Comments**
> >
> > Thanks for the detailed comments. I have one more question about the details of the paper.
> >
> > In page 5, section 2.1, the ViT-VQGAN encoder tokenizer is using a small model, while the decoder uses a large model.
> >
> > `We first train a ViT-VQGAN-Small configuration (8 blocks, 8 heads, model dimension 512, and hidden dimension 2048 as
> > shown in Table 2 of (Yu et al., 2021), with about 30M total parameters), and learn 8192 image token classes
> > for the codebook. We note that the second stage autoregressive encoder-decoder training only relies on the
> > encoder and the codebook of a learned image tokenizer. To further improve visual acuity of the reconstructed
> > images after second-stage encoder-decoder training, we freeze the tokenizer’s encoder and codebook, and
> > finetune a larger-size tokenizer decoder (32 blocks, 16 heads, model dimension 1280, and hidden dimension
> > 5120, with about 600M total parameters). We use 256×256 resolution for the image tokenizer’s input and
> > output.`
> >
> > I wonder what the reason behind this?  Why not use the ViT-VQGAN large for both encoding and decoding? What is the benefit of using the smaller model for the encoder tokenizer? Thanks!

---

> > > ### Author Response · Authors · 2022-10-11
> > > **Authors' Responses to Reviewer QQkZ on ViT-VQGAN tokenizer**
> > >
> > > Thanks for the follow up question!
> > >
> > > 1. Why not use the ViT-VQGAN large for both encoding and decoding? What is the benefit of using the smaller model for the encoder tokenizer?
> > >
> > > Our choice of ViT-VQGAN encoder / decoder sizes are based on a few empirical reasons: (1) in terms of FID (the primary quality metric of image tokenizer), we find it's more beneficial to increase the decoder size than encoder size given compute constraints. As a reference, our base-size encoder and base-size decoder achieves FID of 1.55 on ImageNet validation set with 30k examples, large-size encoder and large-size decoder achieves FID of 1.42, our small-size encoder and huge-size decoder achieves FID of 1.16 (the final setup we used in this work). The sizes of small / base / large / huge of image tokenizer are listed in the table below. (2) The training of Parti's encoder-decoder model only activates the encoder path of an image tokenizer, thus an image tokenizer with smaller encoder speeds up the sequence-to-sequence model training, where random augmentations are applied first to an image, followed by the encoder of image quantizer to obtain the input image tokens. (On the other hand, the inference / sampling of Parti model only actives the decoder path of an image tokenizer, as shown in Figure 3).
> > >
> > >
> > > |  name | num_layers | num_heads | model_dims | mlp_dims |
> > > |:-----:|:----------:|:---------:|:----------:|:--------:|
> > > | small |      8     |     12    |     512    |   2048   |
> > > |  base |     12     |     12    |     768    |   3072   |
> > > | large |     24     |     16    |    1024    |   4096   |
> > > |  huge |     32     |     16    |    1280    |   5120   |
> > >
> > > The sizes of small / base / large / huge of image tokenizer.

---

### Decision · Action_Editors · 2022-10-26

**Recommendation:** Accept as is

**Comment:**

This paper presents Parti for high-fidelity photorealistic image generation. Parti is a simple auto-regressive model similar to DALLE but with an encoder-decoder architecture, and achieves state-of-the-art performance on image generation (compared to DALLE2 and the latest diffusion models). The authors have performed a comprehensive evaluation and human study of Parti model and the limitation of the model is also well discussed. On the other hand, reviewers also showed concerns about open-source plans and experiments regarding scaling laws, which is not easy to address, but we also could not blame too much on this. Overall, the editor thinks that this paper is of great significance, which shows the power of scaling up simple models for achieving impressive results.

**Audience:**

The general TMLR's audience will be interested in this paper.

**Claims And Evidence:**

The claims made in this submission is accurate with clear evidence.

---

> ### Author Response · Authors · 2022-11-05
> **Authors' Updates to the Editor and Reviewers**
>
> We'd love to thank the editor and all reviewers for reviewing and providing improvement comments to this work. We have incorporated the feedbacks into our final camera-ready version. Thanks!